# Mixed-matrix membranes with molecular recognition windows for selective helium extraction from natural gas

Wen He[1], Xiangzeng Wang[2], Jian Guan[1], Quansheng Liang[2], Ji Ma[1], Ying Liu[2], Hongjun Zhang [3], Chunwei Zhang[2] & Jiangtao Liu [1]✉

Mixed-matrix membranes (MMMs) with high chain packing density by incorporating soluble macrocycle compounds represent a promising class of materials for gas separation. However, achieving the ultra-high selectivity (He/CH$_4$ > 1000) for helium extraction from natural gas with ultra-low helium content remains a formidable challenge, especially for Matrimid membranes, which are commercially available but exhibit relatively low permeability and moderate selectivity. Herein, the cyclic Cyclen with specific intra-ring dimensions was incorporated into Matrimid as a pore-structure modifier to enhance the He/CH$_4$ selectivity. The strong hydrogen bonding interactions between Cyclen and Matrimid chains induced a denser chain stacking and modulation of the interchain gap structures, which enables rapid mass transfer of small He gas molecules while hindering the diffusion of large CH$_4$ gas molecules across the membrane, thereby significantly enhanced He/CH$_4$ molecular sieving capacity. Molecular dynamics simulations indicate that the MMMs prepared using Cyclen as a filler exhibited tunable microporous and more efficient He transport channels. Notably, the He/CH$_4$ selectivity reached up to an impressive value of 6788 after physical aging for 110 days, which outperformed almost all reported polymer-based membranes and was even comparable to that of some advanced carbon molecular sieve membranes.

Helium (He) is an extremely scarce, non-renewable gas with a robust safety profile, making it an essential resource for the development of numerous high-tech industries, including aerospace, electronics, medical equipment, and scientific research[1–3]. The rapid expansion of these industries has driven the global consumption of helium to increase at an annual rate of 5–7%[4]. On Earth, He is primarily found in the atmosphere and in natural gas. Although the atmosphere contains a substantial amount of He, its ultra-low concentration (less than 5 ppm) makes extraction from the air extremely difficult[5–7]. By contrast, natural gas serves as the richest and most accessible source, despite the initial concentration of He in most natural gas reservoirs is still relatively low (around 0.3%), which also places a great challenge on the He extraction technology[8–11]. Currently, the most widely used and large-scale industrial He separation and purification methods are based on energy-intensive processes such as cryogenic distillation and pressure swing adsorption[12]. To reduce the overall energy consumption and enhance the separation efficiency, membrane technology has emerged as an effective alternative for He enrichment[13,14]. Membrane-based separation technologies are particularly attractive due to their low energy consumption, high efficiency, and operational simplicity in the field of He recovery from natural gas[15–19].

[1]CAS Key Laboratory of Urban Pollutant Conversion, Department of Environmental Science and Engineering, University of Science and Technology of China, Hefei, China. [2]Shanxi Yanchang Petroleum (Group) Co., Ltd., Xi'an, China. [3]State Key Laboratory of Particle Detection and Electronics, University of Science and Technology of China, Hefei, China. ✉e-mail: jiangtaoliu@ustc.edu.cn

The conventional polymer membranes for gas separation have favorable processability, high stability and low cost[20]. However, their gas transport performance is governed by an inherent trade-off relationship between permeability and selectivity, resulting in most commercially available polymer membranes having low permeability and moderate selectivity[21–23]. To extract helium from natural gas with ultra-low He content, the gas separation membranes must possess He/$CH_4$ selectivity greater than 1000, a critical threshold for industrial applications[6,24]. Unfortunately, few polymer membranes meet this stringent requirement[25]. Membranes based on molecular sieve materials can easily overcome Robeson's upper limit by relying on their ability to distinguish molecules based on size and shape, but these materials often suffer from poor mechanical properties[26]. Mixed matrix membranes (MMMs), which incorporate nanofillers with certain pore sizes into a processable polymer matrix, effectively improve the gas separation efficiency by integrating the advantages of both components[27–29]. Nonetheless, the disparity in physicochemical properties between filler and polymer often leads to poor interfacial compatibility, resulting in the formation of non-selective regions at the interface that ultimately reduce membrane selectivity[30–32]. Enhancing filler-polymer interactions can improve the interfacial compatibility of mixed matrix membranes. Common methods used in current research include filler surface functionalization[30], morphology regulation[33,34], in-situ growth[35], solid-solvent processing approach[36], ligand replacement[37,38], ionic cross-linking[39], covalent cross-linking[40], and intermolecular force regulation[41]. However, most nanofillers show poor dispersibility in organic solvents, and the formation of nanofiller clusters in the polymer matrix will reduce the gas transport properties[42]. Even though the dispersion of nanofillers can be enhanced by surface functionalization[43,44], it is still a great challenge to achieve acceptable selectivity for helium extraction from natural gas.

High-permeability polymers fail to provide the necessary selectivity for high purity He extraction. In this regard, Matrimid was selected as the polymer matrix for its dense stacking of polymer chains. Most of the Matrimid-based mixed matrix membranes used for He separation select porous materials as nanofillers (e.g., Cu-BTC, Cu-BDC, CNFs, C60, etc.)[45–49], which provide special pores that only allow He to pass through, thereby enhancing its diffusion rate. In addition, these nanofillers provide moderate resistance to the mass transfer of larger gas molecules, thereby improving the molecular sieve nature of the membrane. However, the poor compatibility between these fillers and Matrimid induces an undesirable effect on membrane separation performance, making it difficult to meet the required separation capacity for He purification.

Macrocycles with specific nanostructures are organic molecules that can be well dispersed in organic solvents[50]. The functional groups of macrocycles can form strong intermolecular forces with the polymers, thereby improving their interfacial compatibility with the polymer matrix and minimizing their self-interaction tendency in the mixed solution[51,52]. In addition, the strong interactions between the macrocycle and the polymer chains tighten the polymer packing closer together and shrink the *d*-spacing values of polymer chains, which significantly improves the gas separation performance of the membranes[52–54]. Cyclen, an organic macrocyclic compound with flexible structure, has a cavity size of ~3.2–3.4 Å[55,56]. The abundant secondary amine groups in the Cyclen structure provide enough binding sites to modulate the membrane structure through the formation of chemical bonds or strong intermolecular forces, which is commonly used in the construction of molecule transport channels to enhance membrane densification and reduce interfacial defects by regulating the free volume of the membrane[57–59].

In this study, we prepared He separation membranes with ultra-high permselectivity by incorporating a molecule recognition window (Cyclen) into Matrimid. Cyclen connected with Matrimid chains by hydrogen bonding (Fig. S1), which adjusted the submicroporous structure of the MMM membrane, tightened the chain spacing, and increased the transmembrane mass transfer resistance for large gases. At the same time, the Cyclen with a certain cavity size creates a special transport channel for small gases, increasing their permeability and counteracting the negative effect of increased chain stacking density (Fig. 1). In addition, the membrane showed exceptional performance after physical aging, exceeding that of almost all mixed matrix membranes.

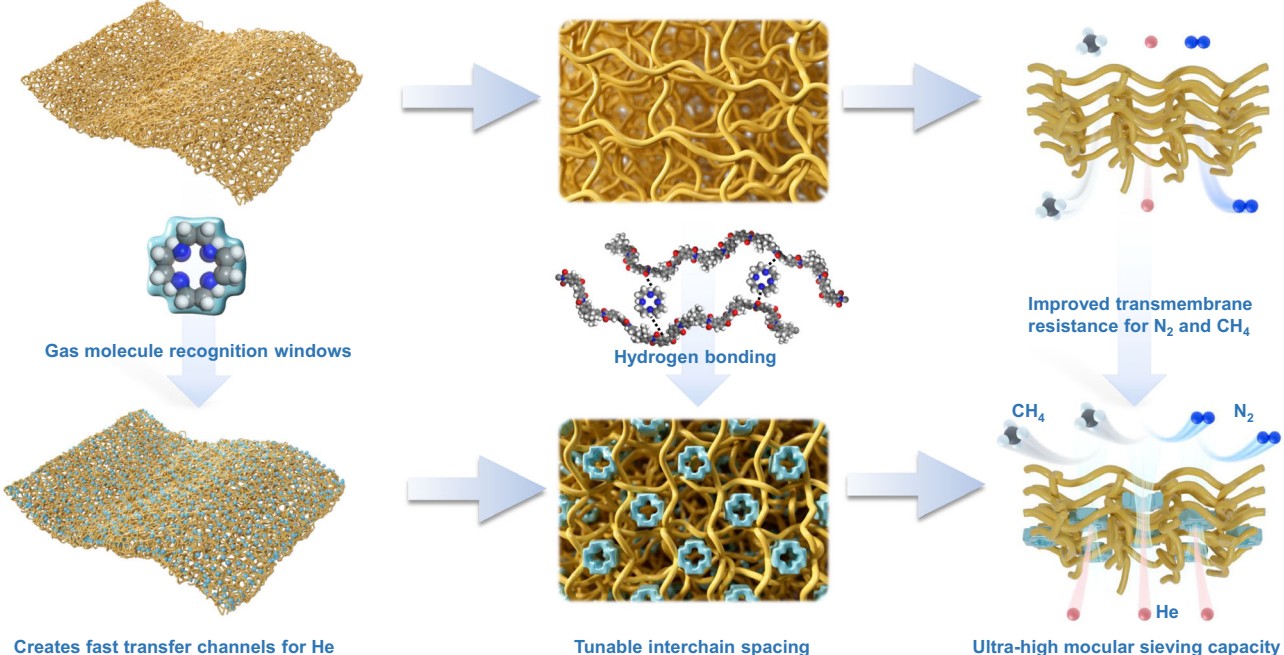

**Gas molecule recognition windows**

**Creates fast transfer channels for He**

**Hydrogen bonding**

**Tunable interchain spacing**

**Improved transmembrane resistance for $N_2$ and $CH_4$**

$CH_4$ $N_2$

He

**Ultra-high mocular sieving capacity**

**Fig. 1 | Schematic of the mixed-matrix membranes (MMMs) for helium extraction from natural gas via gas molecule recognition window.** Including membrane structures, chain stacking modes, and gas transport mechanisms of the Matrimid and Matrimid-Cyclen membranes.

## Results

### Characterizations of Matrimid-Cyclen membranes

The optical photograph showed that all Matrimid-Cyclen membranes are yellow and highly transparent (Fig. 2a). The cross-sectional microstructures of Matrimid-Cyclen membranes by the field emission scanning electron microscopy (FESEM) are shown in Fig. 2b. With the increase of Cyclen content, the vein structures gradually appeared and became dense, which indicates a continuous enhancement of the hydrogen-bonding interactions between the Matrimid chains and Cyclen. At low Cyclen loading (Cyclen loading ≤ 5%), the cross sections of Matrimid-Cyclen membranes demonstrated dense and smooth morphologies. With the gradual increase of Cyclen loading (10–15% Cyclen loading), more hydrogen bonding between Cyclen and Matrimid chains leads to the formation of more prominent vein structures in the membrane. All the Matrimid-Cyclen membranes display homogeneous cross section and surface at 50,000× electron magnification, without any significant aggregation or discrete particulate phases observed (Fig. 2c and Fig. S2), confirming the presence of only a continuous single phase within these membranes. Additionally, energy dispersive X-ray spectroscopy (EDS) mapping of the cross-sections of the matrimid-Cyclen membranes further verifies the uniform distribution of Cyclen as evidenced by the detection of characteristic nitrogen (N) signals (Fig. 2d).

The X-ray diffraction (XRD) patterns of pure Matrimid membrane, MMMs, and Cyclen are shown in Fig. 3a and Fig. S3. The pristine Matrimid membrane exhibits two broad amorphous diffraction peaks

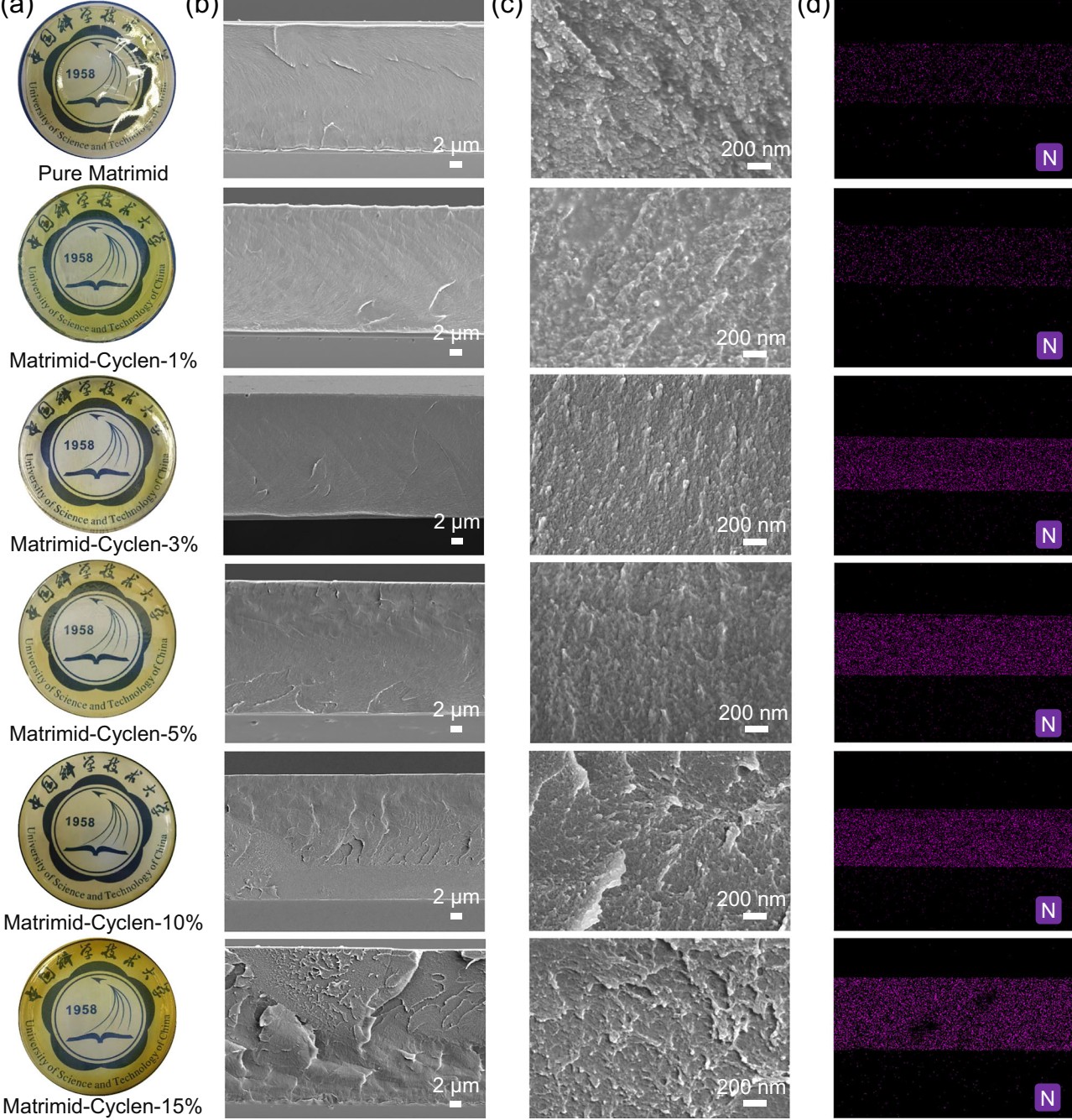

**Fig. 2 | SEM images of the Matrimid-Cyclen membranes. a** photograph, and **b**, **c** cross section of the Matrimid-Cyclen-*n*% membranes (*n*% is the Cyclen mass loading and was varied from 0 to 15 wt%, *n* = 0, 1, 3, 5, 10, 15). **d** Cross-sectional SEM-EDS nitrogen (N) mapping of Matrimid-Cyclen membranes.

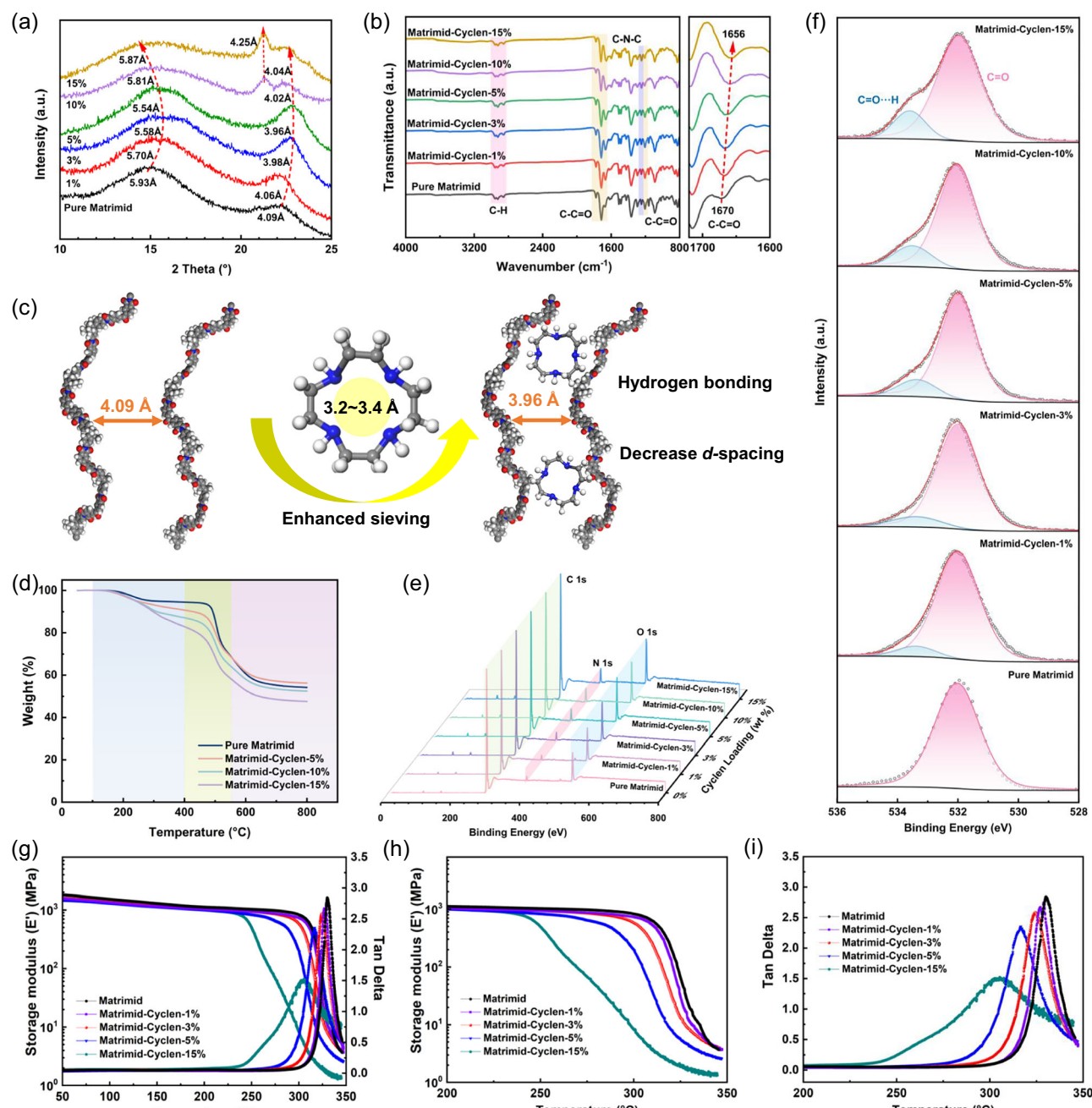

**Fig. 3 | Structure characterization of Matrimid-Cyclen membranes. a** XRD patterns of the Matrimid-Cyclen membranes. **b** ATR-FTIR spectra of the Matrimid-Cyclen membranes. **c** Schematic illustration of the effect of intramembrane hydrogen bonding on polymer chain stacking as the Cyclen content is increased from 0% to 5%. **d** TGA plots of the Matrimid-Cyclen membranes. **e** X-ray photoelectron spectroscopy (XPS) broad scan spectra of the Matrimid-Cyclen membranes. **f** XPS data for the O 1s peaks of the Matrimid-Cyclen membranes. **g** The dynamic mechanical thermal analysis (DMTA) spectra of Matrimid-Cyclen membranes over a temperature range from 50 to 350 °C. **h** The storage modulus (E′) and **i** Tan δ of Matrimid-Cyclen membranes (**h**, **i** are local enlargements of (**g**).

at ~15 and 22°, corresponding to d-spacings of 5.93 Å and 4.09 Å, respectively. These peaks indicate the existence of two major types of interchain gaps in Matrimid membranes. When Cyclen was incorporated at low loading (1–5% Cyclen loading), only two broad amorphous diffraction peaks were observed for the Matrimid-Cyclen membranes, while the crystallinity of Cyclen completely disappeared upon mixing with the polymers, which suggests that homogeneous mixture at the molecular level was achieved between Cyclen and Matrimid. The peak at 15° in Matrimid membrane corresponding to a d-spacing value of 5.93 Å and the broad peak at 22° corresponding to a d-spacing of 4.09 Å. At low loadings (1–5% Cyclen loading), these peaks shift to lower angles in the MMMs. Specifically, the d-spacing value of the

broad peaks at 22° and 15° decreases from 4.09 to 3.96 Å and 5.93 to 5.54 Å, respectively. This shift is attributed to the hydrogen bonding interactions between Cyclen and the regularly arranged Matrimid chains reduces interchain spacing and densifies the overall packing, which is essential for improving the sieving ability of the membrane (Fig. 3c). However, two bigger interchain regions tend to undergo distinct expansion with further increasing Cyclen content (10–15% Cyclen mass loading), probably because an excessive number of Cyclen molecules are accommodated within the microporous region. In addition, a new diffraction peak attributed to Cyclen was observed, the peaks of Cyclen are evident in those of the MMMs without any peak shifts, indicating that the crystalline structure of Cyclen was

maintained inside Matrimid, and that penetration of the Matrimid chains could be neglected.

The strong interaction between the filler and polymer often leads to peculiar changes in the Fourier-transform infrared-attenuated total reflectance (ATR-FTIR) profile, thus ATR-FTIR analysis of the Matrimid membrane, Matrimid·Cyclen membranes, and Cyclen were conducted. As shown in Fig. 3b and Fig. S4, the characteristic absorbance bands at 1775 cm$^{-1}$ and 1717 cm$^{-1}$ correspond to the symmetric and asymmetric stretching vibrations of the 5-membered imide ring carbonyl groups, whereas the band at 1670 cm$^{-1}$ corresponds to the symmetric stretching vibration of benzophenone carbonyl. Compared with the Matrimid membrane, the characteristic C-C=O peak at 1670 cm$^{-1}$ of all Matrimid·Cyclen membranes shows a red shift as the Cyclen loading increased, which was attributed to the formation of hydrogen bonding interaction between Matrimid and Cyclen. It is speculated that the preferred binding site for the -NH of Cyclen is the benzophenone carbonyl on the Matrimid backbone. After the formation of MMMs, the characteristic peaks attributed to the imine N-H stretching vibrations in the 3080–3340 cm$^{-1}$ range of the Cyclen spectra completely disappeared (Fig. 3b and Fig. S4), which confirmed the formation of dense hydrogen bonding in the Matrimid·Cyclen membranes. The dense hydrogen bonding between Cyclen and Matrimid enhances their interfacial compatibility, reduces non-selective interfacial gaps and significantly improves the high gas separation performance of the MMMs[21,32].

Figure 3d presents the thermogravimetric analysis (TGA) of Matrimid and Matrimid·Cyclen membranes. Cyclen begins to degrade at temperatures slightly above 100 °C (Fig. S5). The initial weight loss observed from 100 to 400 °C may be attributed to the evaporation of the residual organic solvent and the degradation of Cyclen within the MMMs. The weight loss gradually increases with increasing Cyclen mass loading. The second weight loss occurred between 400 and 550 °C, at which point the degradation of lateral methyl groups from Matrimid begins. The third stage is caused by the decomposition of the polymer main chains, which begins from ~550 °C. In general, the interactions between the polymer matrix and the filler restrict the thermal movement of the polymer chains, resulting in an increase in the energy required for chain movement and decomposition, thereby enhancing the thermal stability of these membranes. The FTIR analysis indicates that hydrogen bonding between Matrimid and Cyclen increased with the increase of Cyclen mass loading, the thermal stability of Matrimid·Cyclen membranes should be improved. However, this is not the case, as the decomposition temperature of Cyclen is significantly lower than that of Matrimid (Fig. S5). The decomposition of Cyclen creates micropores within the Matrimid·Cyclen membranes, which provide additional space for polymer chain movement and reduce the energy required for the decomposition of the polymer chains, ultimately decreasing the overall thermal stability of the membrane.

To further investigate the intermolecular forces in the membrane, the chemical composition and elemental states were analyzed using X-ray photoelectron spectroscopy (XPS). The full XPS survey spectra (Fig. 3e) reveal that the nitrogen (N) signal intensity progressively increases with Cyclen loading. The appearing C=O···H and N-H···O signals in the O 1*s* and N 1*s* XPS spectra of the Matrimid·Cyclen membrane as compared to the pristine Matrimid membrane provided evidence for the formation of hydrogen bonds (Fig. 3f and Fig. S6). In addition, the extent of hydrogen bonding reactions between secondary amine groups in Cyclen and the carbonyl in Matrimid backbone quantified by XPS results is found to increase from 5.96 to 15.37% as the Cyclen mass loading increases from 1 to 15% (Table S1), which further corroborates the existence of dense hydrogen bonding between Cyclen molecule and Matrimid chains. These findings are consistent with previous characterization results and the hydrogen bonding interactions between the N-H groups of

Cyclen and the carbonyl (C=O) groups of Matrimid play a pivotal role in governing polymer chain packing. Dynamic mechanical thermal analysis (DMTA) was conducted for Matrimid·Cyclen samples to evaluate their mechanical property within a temperature range of 50–350 °C. As shown in Fig. 3g–i, the storage modulus (E′) (1800 MPa) of these membranes almost completely maintained before reaching the corresponding glass transition temperature (Tg). Actually, the higher Cyclen loadings resulted in more brittle membranes. The effect of Cyclen content on the peak intensity of damping (Tan δ) is shown in Fig. 3i. The peak intensity of Tan δ (damping value) decreases as Cyclen content increases. These results demonstrated that the Matrimid·Cyclen membranes exhibit both good mechanical properties and thermal stability, making them promising candidates for gas separation applications.

## Gas separation performance of Matrimid·Cyclen membranes

To investigate the gas transport properties of Matrimid and Matrimid-Cyclen membranes, gas permeation tests for He (2.60 Å), H$_2$ (2.89 Å), CO$_2$ (3.3 Å), N$_2$ (3.64 Å) and CH$_4$ (3.80 Å) were performed using a constant-volume, variable-pressure system at 25 °C under a feed pressure of 1.0 bar (Fig. S7). The gas permeability and selectivity data were summarized in Table S2. As shown in Fig. 4a, the gas permeability was found to decrease with increasing gas kinetic diameter, and a distinct cut-off effect of the gas permeability was observed between CO$_2$ and N$_2$, demonstrating the presence of distinct gas molecular sieving characteristics in the Matrimid·Cyclen membranes. The incorporation of Cyclen enhances the molecular sieving effect, showing significant separation potential for He, H$_2$, and CO$_2$. The uniform pore size (3.2–3.4 Å) of Cyclen provides dedicated transport channel for small molecule gases (i.e., He, H$_2$ and CO$_2$), facilitating their selective transport across the membrane, while the large molecule gases (N$_2$ and CH$_4$) exhibit decreased gas permeability due to the tightening of polymer chain spacing.

The gas separation performance of the prepared Matrimid·Cyclen membranes was evaluated for several important gas pairs (He/N$_2$, He/CH$_4$, H$_2$/N$_2$, H$_2$/CH$_4$, CO$_2$/N$_2$ and CO$_2$/CH$_4$) as a function of filler concentration (Fig. 4c–e). With increasing Cyclen loading from 0 to 5 wt%, the He permeability increased from 34 to 66 Barrer, the H$_2$ permeability increased from 27 to 62 Barrer, and the CO$_2$ permeability increased from 15 to 20 Barrer, corresponding to increases of ~98%, 127% and 34%, respectively. In contrast, the permeability of larger gas molecules (e.g., N$_2$ and CH$_4$) steadily decreased. The inconsistent permeability changes of these gases make the membranes exhibit attractive ultra-high gas selectivity, the selectivities of He/N$_2$ and He/CH$_4$ are 8 and 19 times that of the pure membranes, respectively. The improved separation performance of Matrimid·Cyclen membranes indicated a reduction in interchain gaps with a significantly enhanced size-sieving effect, which is consistent with the XRD findings. XRD analysis shows that the increasing hydrogen bonding interactions between Cyclen and Matrimid will reduce the polymer chain spacing as the Cyclen content increases from 0% to 5%, which might typically result in a decrease in permeability for all gases. In fact, the permeabilities of He, H$_2$ and CO$_2$ continue to increase as the Cyclen content increases. This can be attributed to the molecular sieve effect of Cyclen, which selectively facilitates the passage of gases with molecular diameters smaller than the Cyclen pore size while hindering the transport of larger molecules such as N$_2$ and CH$_4$ to pass through more devious diffusion transport pathways. Therefore, higher Cyclen content leads to the formation of more efficient molecular sieving channels, enhancing the permeability of small gases (e.g., He, H$_2$ and CO$_2$). The strong hydrogen bonding between Cyclen and Matrimid tightens the *d*-spacing of Matrimid polymer chains and reduces the chance of defects in the membrane, resulting in longer and more tortuous transport pathways for larger gas molecules (e.g., N$_2$ and CH$_4$), thereby decreasing their permeability (Fig. 4b). This also increases the

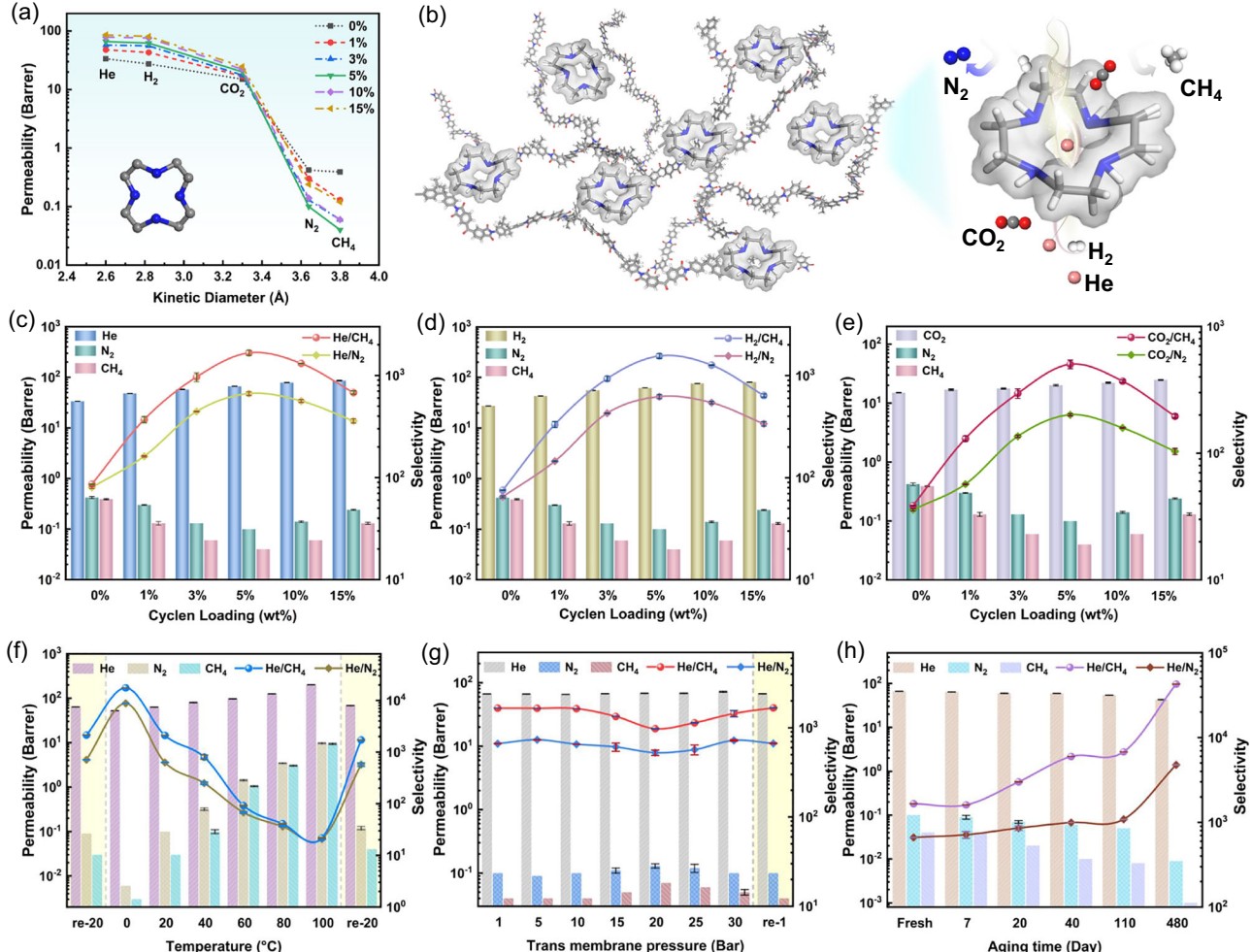

**Fig. 4 | Gas separation performance of Matrimid-Cyclen membranes.**
**a** Correlations between the permeabilities and molecular kinetic diameters of various gases. **b** Schematic illustration of gas transport behaviors of Matrimid-Cyclen membrane. **c** He, $N_2$ and $CH_4$ permeability and $He/N_2$ and $He/CH_4$ selectivity with different Cyclen mass loading. **d** $H_2$, $N_2$ and $CH_4$ permeability and $H_2/N_2$ and $H_2/CH_4$ selectivity with different Cyclen mass loading. **e** $CO_2$, $N_2$ and $CH_4$ permeability and $CO_2/N_2$ and $CO_2/CH_4$ selectivity with different Cyclen mass loading. **f** Effect of operating temperature on the separation performance of Matrimid-

Cyclen-5% membrane. **g** Effect of feed gas pressures on the separation performance of Matrimid-Cyclen-5% membrane. **h** Long-term stability of the Matrimid-Cyclen-5% membrane (membrane was stored in an ambient environment without the introduction of any protective gas at room temperature, small pieces of membrane were extracted from the complete membrane at different aging times for a gas separation performance test). All data are presented as mean values ± s.d. (standard deviation), $n \geq 3$.

selectivity of multiple important gas pairs. The Matrimid-Cyclen-5% membrane exhibits a He permeability of 66 Barrer, and the separation selectivities of $He/N_2$ and $He/CH_4$ were 664 and 1660, respectively. These results are very attractive for industrial applications, as improving selectivity is more critical than enhancing permeability for obtaining high-purity He. With Cyclen content increased from 5 to 15%, all the gas permeabilities increased, the $He/CH_4$ selectivity decreased from 1660 to 661, and the $He/N_2$ separation selectivity decreased from 664 to 358. This decline in selectivity is primarily due to the formation of non-selective regions at high filler loading. Although more Cyclen molecules can form more selective gas transport channels during the process of membrane formation, excessive Cyclen concentration uniformly dispersed in the polymer chains can also cause an increase in polymer chain spacing, leading to the formation of non-selective regions larger than the kinetic diameters of $N_2$ and $CH_4$, which diminishes the molecular sieving capacity, reduces the overall selectivity of the membrane. At the same time, the formation of non-selective regions with larger sizes increases the permeability of all gases.

## Effects of operating temperatures, feed pressures and aging behavior

As shown in Fig. 4f and Fig. S8, the effect of operating temperature on the separation performance of the Matrimid-Cyclen-5% membrane at 1 bar for $He/N_2$ and $He/CH_4$ was investigated (Table S3). As the operating temperature increased from 0 °C to 100 °C, the permeability of He, $N_2$ and $CH_4$ increased, while the selectivity of $He/N_2$ and $He/CH_4$ decreased. This could be attributed to the intense thermal motion movement of polymer chains at higher temperatures, which increased the percentages of larger size transient gaps in the membrane. $N_2$ and $CH_4$ molecules benefit more from increased percentage of larger size transient gaps than He molecules with smaller kinetic diameters, and the increased permeability of $N_2$ and $CH_4$ molecules is more dramatic, weakening the membrane's selectivity for gas pairs. The selectivity of $He/N_2$ and $He/CH_4$ decreased rapidly as the operating temperature increased from 40 to 60 °C. This can be explained by the accelerated movement frequency of polymer chains at high temperature, rendering fast transport of gas molecules through the matrix, the permeability of larger molecules ($N_2$ and $CH_4$) increases more than that of He,

weakening the molecular sieving effect of the membrane. After cooling from 20 to 0 °C, the He permeability decreased from 63 to 53 Barrer, while the He/$N_2$ selectivity increased from 631 to 7534, and the He/$CH_4$ selectivity increased from 2104 to 17580, corresponding to 12-fold and 9-fold increases, respectively. The ultra-high selectivity of the Matrimid-Cyclen-5% membrane at 0 °C for He/$N_2$ and He/$CH_4$ facilitates the extraction of high-purity He from natural gas at low temperatures. In industrial applications, improving gas purity is often more challenging than enhancing transport efficiency, especially for He, which is critical for industries such as aerospace, electronics, medical equipment and scientific research. Notably, the gas separation performance of the membranes fully recovered as the temperature changed from 100 °C and 0 °C to 20 °C, indicating that the observed temperature-dependent performance changes are related to the characteristics of the membrane rather than any structure damage.

To explore the practical applicability of the Matrimid-Cyclen membranes at different feed pressures, the gas separation performance of the Matrimid-Cyclen-5% membrane was evaluated at pressures up to 30 bar, and the data are summarized in Table S4. As presented in Fig. 4g, as the pressure increases from 1 bar to 10 bar, the permeability of He decreases slightly, and the pressure dependence of $N_2$ and $CH_4$ was almost unrecognizable, with only small fluctuations in the selectivity for He/$N_2$ and He/$CH_4$. These observations suggest that the membranes remain structurally stable under moderate pressure increases. This may be attributed to the presence of Cyclen molecules, which increase the packing density of the polymer chains, thereby reducing their mobility, increasing chain stiffness and improving the high-pressure resistance of the membranes. As the operating pressure increases from 10 bar to 20 bar, the permeability of all gases rises, and the selectivity for He/$N_2$ and He/$CH_4$ decreases. This behavior is typical of glassy polymers, such as Matrimid, under plasticization pressure, where increased feed pressure leads to higher polymer segment mobility and larger chain spacing, resulting in higher permeability and lower selectivity. This phenomenon aligns with the dual-mode sorption theory. As the pressure increases from 20 bar to 30 bar, the permeability of He remains almost constant, while the permeability of $N_2$ and $CH_4$ gradually decreases, and the selectivity of He/$N_2$ and He/$CH_4$ tends to increase, which is ascribed to the existence of unrelaxed volume within the polymer chain structure. An increase in feed pressure results in greater compactness of the polymer matrix, which reduces the gas permeability in the membrane by reducing fractional free volume.

The membrane was stored at room temperature without any protective gases, and its performance was periodically tested to evaluate the physical aging performance (Table S5). As shown in Fig. 4h, after 110 days of aging, the He permeability of the Matrimid-Cyclen-5% membrane decreased from 66 Barrer to 54 Barrer, a decrease of 18%. However, the He/$N_2$ selectivity increased from 664 to 1086, and the He/$CH_4$ selectivity increased from 1660 to 6788, corresponding to increases of 64% and 310%, respectively. After physical aging for 480 days, the He permeability of Matrimid-Cyclen-5% membrane decreased from 66 Barrer to 43 Barrer, a decrease of 36%. The selectivity showed an order of magnitude improvement due to the fact that the polymer chain segments in the non-equilibrium state within the membrane gradually adjust to an equilibrium state by optimizing the chain stacking during the physical aging process. The shrunk fractional free volume reduces the gas permeability to a certain extent, while greatly improves the molecular sieving capacity of the membrane. The XRD pattern of Matrimid-Cyclen-5% membrane showed a significant decrease in chain spacing after aging 480 days, further demonstrating that the adjustment of chain stacking during the aging process has improved the molecular sieving capacity of the membrane (Fig. S9).

Figure 5 compares the He/$N_2$, He/$CH_4$, $H_2/N_2$, $H_2/CH_4$, $CO_2/N_2$ and $CO_2/CH_4$ separation performance of the Matrimid-Cyclen membranes with those of the Matrimid-based membranes and other membranes,

and the data are presented in Tables S6–S8. It is observed that the Matrimid-Cyclen-5% membranes exhibited optimized comprehensive separation performance, with He permeability of 66 Barrer, He/$N_2$ and He/$CH_4$ selectivities up to 664 and 1660, respectively (Fig. 5a, b). The Matrimid-Cyclen membranes exhibit ultra-high He selectivity compared to the other membranes reported in the literature, which breaks the 2008 upper bound limit. After physical aging for 110 days, the separation selectivities of He/$N_2$ and He/$CH_4$ reach up to 1086 and 6788, an increase of 18-fold and 110-fold over pure Matrimid membrane, respectively. The selectivities of Matrimid-Cyclen membrane even exceeded those of some carbon molecular sieve (CMS) membranes, which is one of the highest values reported for He gas separation membrane (Fig. S10). Owing to the strong hydrogen bonding between Cyclen and Matrimid, the Matrimid polymer chains aligned more tightly, thus achieving significantly higher He/$N_2$ and He/$CH_4$ selectivities. Other notable gas pairs including $H_2/N_2$, $H_2/CH_4$, $CO_2/N_2$ and $CO_2/CH_4$ also show considerably high gas selectivities at an optimum Cyclen loading of 5% (Fig. 5c–f). The gas separation performance far exceeds the latest gas separation upper bound line. Figure 5g–h show the improvement in both He permeability and He/$CH_4$ selectivity of MMMs formed in Matrimid with four different fillers, the data are summarized in Table S9. The increase in He/$CH_4$ selectivity of MMMs prepared with Cyclen and Cyclodecane, which have the same intra-ring dimensions as fillers is 427% and -18%, respectively, as compared to the pristine Matrimid membrane. This demonstrates that tightening the polymer chain stacking by hydrogen bonding can effectively enhance the separation performance of MMMs. Compared to Piperazine without an annular cavity, the He permeability of MMMs relative to the Matrimid membrane increased by 143 and 108%, respectively. This indicates that the piperazine without macrocyclic cavities also can enhance He/$CH_4$ selectivity by narrowing the polymer interchain spacing via hydrogen bonding. Compared to Hexacyclen with a larger cavity size, the increase in He/$CH_4$ selectivity of MMMs over the Matrimid membrane was 428% and 210%, respectively. This indicates that the filler with suitable cavity size can improve the gas transport efficiency and separation performance of MMMs. The three sets of comparison experiments confirmed the hydrogen bonding interaction between Cyclen and Matrimid, and the suitable gas molecule recognition window size of Cyclen as the reason for improving the molecular sieving performance of the membranes, respectively. The possible gas-transport mechanisms of Matrimid-Cyclen membranes are presented in Fig. 5i. The strong hydrogen bonding between Cyclen and Matrimid chains reduces the $d$-spacing of the chains, and the synergistic interaction with Cyclen with a suitable inner cavity size (3.2–3.4 Å) increases the transport resistance of $N_2$ and $CH_4$, which significantly improves the selectivity of Matrimid-Cyclen membranes.

Molecular dynamic simulations were performed to determine the mechanism of gas separation in the membranes. The membrane structure model is generated from MD simulation based on PACKMOL software (Fig. 6a). A comparison of molecular modeling for the pristine Matrimid membrane and Matrimid-Cyclen membranes reveals efficient chain packing with increasing of Cyclen loading. This hydrogen bond-induced tightening effect aligns with the simulation for free volume and density (Table S10). He and $N_2$ were used as the probe gases for the membrane model to analyze the gas accessible volume, the average gas transmembrane path and the inter-connectivity of void space. Figure 6b, c shows the accessible volumes of He and $N_2$ within Matrimid-Cyclen-$n$% membranes ($n$ = 0, 5, 10). It was found that some channels allow He transport but block $N_2$. As the Cyclen content increased in the membrane, the difference in accessible volume for He and $N_2$ significantly increased and then slightly decreased (Fig. 6e). This suggests that the presence of Cyclen enables the membrane to accommodate more He and less $N_2$ than the pristine Matrimid membrane, thus greatly improving the molecular sieving ability of the membrane. Figure 6d and Fig. S11 present the connectivity of He with

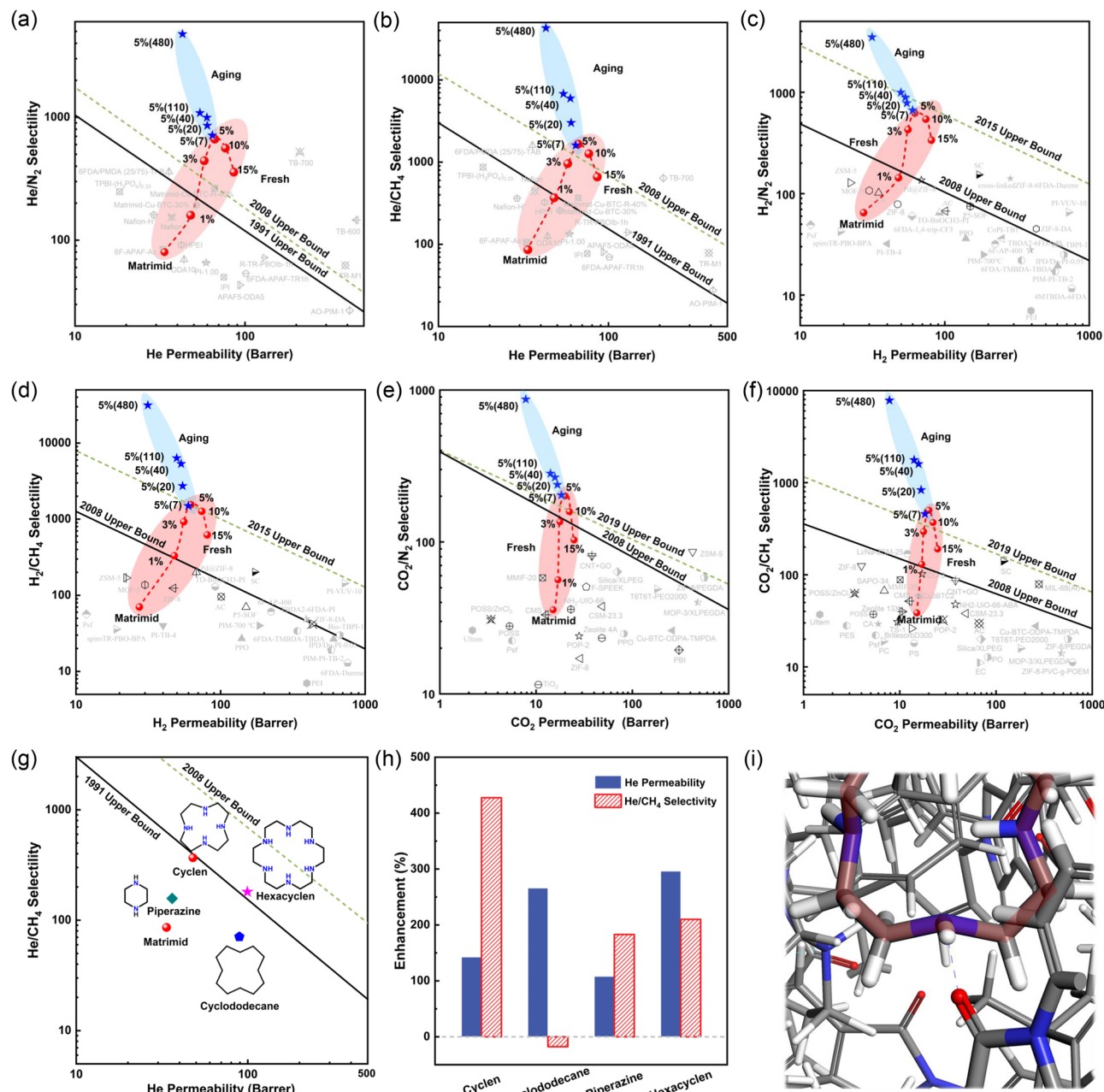

**Fig. 5 | Gas separation performance of Matrimid-Cyclen membranes compared with other polymeric membranes.** Plots of **a** He/N₂, **b** He/CH₄, **c** H₂/N₂, **d** H₂/CH₄, **e** CO₂/N₂ and **f** CO₂/CH₄ separation performance. (The red symbols are gas data of fresh membranes, while the blue ones are gas data of membranes after physical aging, the black symbols are Matrimid-based membranes. Lines are drawn to guide eyes.) **g**, **h** Gas separation performance of mixed matrix membranes (Matrimid-Cyclen-1% membrane, Matrimid-Cyclodecane-1% membrane, Matrimid-Piperazine-1% membrane and Matrimid-Hexacyclen-1% membrane) prepared with different fillers (Cyclen, Cyclodecane, Piperazine and Hexacyclen) at 1% loading. **i** Gas-transport mechanism in Matrimid-Cyclen membranes.

$N_2$-passable void space within Matrimid-Cyclen-$n$% membranes ($n = 0$, 5, 10). The increase in Cyclen content within the membranes resulted in a continuous enhancement in the connectivity of the He passable pore space within the membranes, and the average He transmembrane path calculated on the basis of the He-passable pore space increased from 635 Å to 922 Å (Fig. 6f), thus significantly improving the He permeability. In addition, the average transmembrane path length difference between He and $N_2$ increased from 23 Å to 331 Å and then decreased to 322 Å, indicating that the presence of Cyclen could create a special transport channel for He, increasing its transport paths and transmission path length in the membrane, effectively enhancing the transport capacity of He and the separation of gas pairs in the membrane. To gain further insight into microstructures, the pore size

distribution of membranes was characterized using $H_2$ adsorption experiments combined with molecular dynamics simulation. As shown in Fig. 6g and Fig. S12, the micropore sizes in Matrimid membrane is mainly located at 6.9 Å, accompanied by a small proportion of ultra-micropores with a size of 3.9 Å. In contrast, the Matrimid-Cyclen membranes show smaller micropore sizes at around 6.3 and 6.5 Å, respectively. The Matrimid-Cyclen membranes exhibited significantly reduced micropore sizes, indicating that hydrogen bonding interaction between Cyclen and Matrimid effectively modulate the membrane pore structure. This pore size modulation enhances the molecular sieving properties of Matrimid-Cyclen membranes. The content of ultramicropores with a size of about 3.9 Å in Matrimid-Cyclen membranes gradually increased with increasing Cyclen content. Many

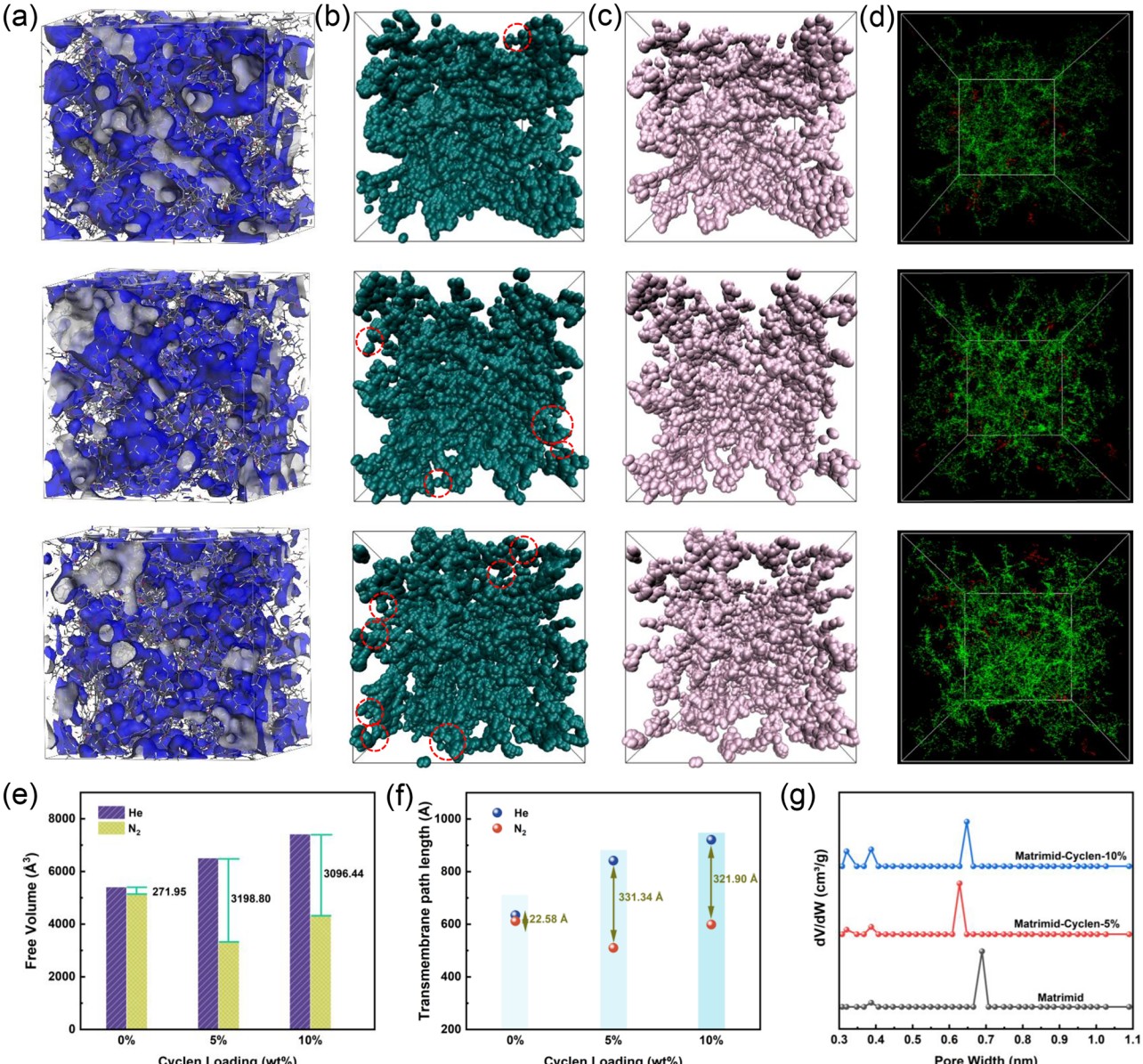

**Fig. 6 | Structural analysis of Matrimid-Cyclen membrane models. a** 3D view of Matrimid-Cyclen-*n*% membrane models (n% is the Cyclen mass loading, *n* = 0, 5, 10). The gray and blue shades indicate the free volume detected by a probe with a radius of 2.6 Å. **b**, **c** Accessible void spaces for He (left) and $N_2$ (right) in Matrimid-Cyclen-*n*% membranes models (*n*% is the Cyclen mass loading, *n* = 0, 5, 10); The red circles in the left picture indicate additional accessible void spaces for He, compared with that of $N_2$. **d** Interconnected (green) and disconnected (red) voids in Matrimid-Cyclen-*n*% membrane models (*n*% is the Cyclen mass loading, *n* = 0, 5, 10) with respect to a probe of 1.3 Å radius. **e** The accessible volume of He and $N_2$ as a percentage of the membrane pore volume and the simulated density of the Matrimid-Cyclen membranes. **f** The average He and $N_2$ transmembrane path was calculated based on accessible void space of the Matrimid-Cyclen membranes. **g** Pore size distribution of the Matrimid-Cyclen membranes. The pore size distribution was calculated from the corresponding $H_2$ adsorption isotherm using the DFT method.

studies have demonstrated that an increase in the content of ultra-microporous pores in the membrane often significantly enhances the membrane's gas molecular sieving ability. Therefore, as the Cyclen content increases, the membrane's selectivity for $He/CH_4$ significantly enhanced. Furthermore, the ultra-microporous pores create selective diffusion pathways that preferentially promote the transport of small gas molecules through the membrane. Consequently, the permeability of small gases gradually increases with increasing Cyclen content. The addition of Cyclen resulted in the appearance of new pores in the membrane with a size of about 3.2 Å, which agrees well with the theoretical value of the molecular window of Cyclen[55,56]. This indicates that Cyclen creates ultra-micropores with the ability to recognize gas molecules, further enhancing molecular sieving properties of the

membranes. Their presence creates more and longer transport channels for He, which significantly improves the transport capacity of He. This gives the membrane satisfactory sieving properties while obtaining good permeability.

Thin-film composite (TFC) membranes, composed of a highly selective thin layer supported by a mechanically robust porous support, have been employed for practical gas separation[60,61]. Matrimid-Cyclen TFC membranes were fabricated on AAO supports using a scalable bar-coating method[62]. Scanning electron microscopy (SEM) images (Fig. 7a, b) showed no significant aggregation or particulate formation. As the membrane thickness increased, the membrane structure evolved to become denser and smoother (Fig. 7a). For the calculation of gas permeance, the thickness of the Matrimid-Cyclen

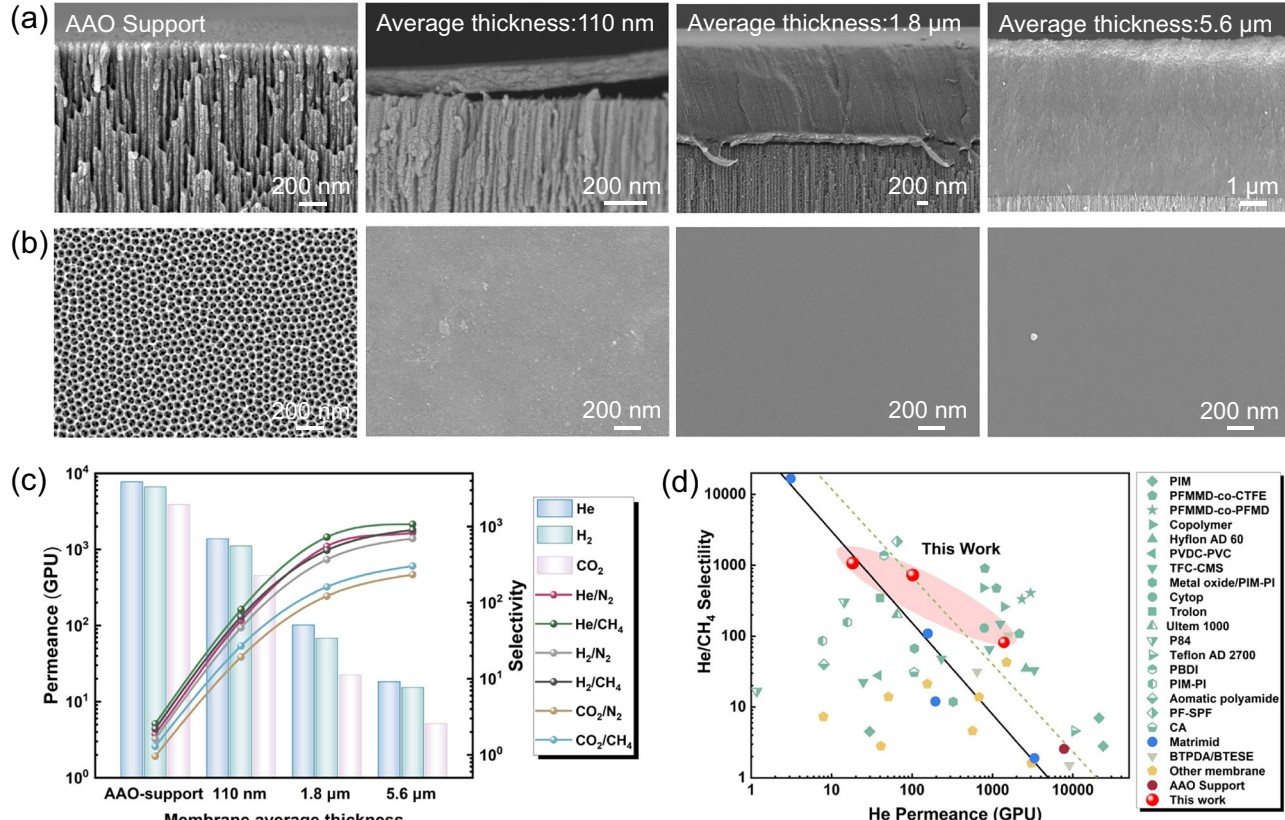

**Fig. 7 | SEM images and gas separation performance of Matrimid-Cyclen-5% TFC membrane. a** Cross section and **b** surface SEM images of the Matrimid-Cyclen TFC membrane with different thickness. **c** Gas separation performances of the Matrimid-Cyclen TFC membrane with different average thicknesses. **d** Comparison of $He/CH_4$ separation performance of Matrimid-Cyclen TFC membranes with available literature data.

TFC membrane's selective layer was determined from cross-sectional SEM images, with values of ~110 nm, 1.8 μm, and 5.6 μm. Figure 7c illustrates the gas permeance and selectivity of the Matrimid-Cyclen TFC membranes with average thicknesses ranging from 110 nm to 5.6 μm (Table S11). The He permeance decreased and the $He/CH_4$ selectivity increased as the membrane thickness increased. Reproducibility was confirmed by testing six randomly selected Matrimid-Cyclen TFC membranes with an average thickness of 1.8 μm, which exhibited consistent He permeance and $He/CH_4$ selectivity (Fig. S13). The average He permeance was $102 \pm 6$ GPU, with a corresponding $He/CH_4$ selectivity of $728 \pm 59$. Figure 7d illustrates the $He/CH_4$ selectivity and He permeance for Matrimid-Cyclen TFC membranes and other types of membranes reported in the literature (please see the detailed comparison in Table S12). Although the membrane's gas permeance is not as high as that of some polymer membranes, Matrimid-Cyclen-1.8 μm exhibits higher $He/CH_4$ selectivity than most reported membranes. Moreover, its performance still exceeds the 2008 Robeson upper bound. In addition, the gas permeance of the AAO supports, pure Matrimid TFC membranes (without filler), and Matrimid-Cyclen TFC membranes (with filler) is summarized in Table S13. Notably, the Matrimid-Cyclen TFC membrane exhibits significantly higher $He/CH_4$ selectivity than the pure Matrimid TFC membrane, the critical role of the cyclen filler in enhancing gas selectivity.

## Discussion

In summary, Cyclen was incorporated as nanofillers into the Matrimid polymer matrix, and the fortified hydrogen bonding interactions improved the interfacial compatibility between Cyclen and Matrimid polymer chains. The pore size of Cyclen (3.2–3.4 Å) creates a significant permeation cut-off between quick gases (He: 2.60 Å, $H_2$:

2.89 Å, $CO_2$: 3.30 Å) and slow gases ($N_2$: 3.64 Å and $CH_4$: 3.80 Å). The strong hydrogen bonding between the N-H groups from Cyclen macrocycle structure and the C=O groups from Matrimid chain effectively tunes the interchain spacing and blocks the transport ways for larger $N_2$ and $CH_4$ gas molecules. The Matrimid-Cyclen membranes achieve ultra-high $He/CH_4$ selectivity of 1660 at an optimum Cyclen loading of only 5%. Strikingly, all the separation performance for $H_2/CH_4$, $CO_2/N_2$ and $CO_2/CH_4$ of Matrimid-Cyclen-5% membrane considerably exceed the 2008 Robeson upper bound line. Both $He/N_2$ and $He/CH_4$ separation performance far exceed the upper bound line even after physical aging for 110 days, which is on par with some CMS membranes and makes it one of the best He gas separation membranes. Therefore, the Matrimid-Cyclen membranes with gas molecule recognition windows possess great potential for high purity He acquisition, $H_2$ purification and $CO_2$ capture from natural gas.

## Methods
### Materials
1, 4, 7, 10-Tetraazacyclododecane (Cyclen, >98%, C8H20N4, 172.27 g/mol), Cyclododecane (Cyclodecane, >99%, C12H24, 168.32 g/mol), and 1, 4, 7, 10, 13, 16-Hexaazacyclooctadecane (Hexacyclen, >98%, C12H30N6, 258.41 g/mol) from Shanghai BeiDi Pharmaceutical Technology Co. were supplied as fine solid powder. Matrimid was supplied by Huntsman Advanced Materials. ReagentPlus®-grade N, N-dimethylformamide (DMF) solvent (≥99.5%) purchased from China National Pharmaceutical Group was used directly for making polymer solutions without further purification. Matrimid 5218 was dried under vacuum at 120 °C for 12 h each time before use. Methanol solvent (≥99.5%) from China National Pharmaceutical Group was directly utilized as received. Purified He gas (99.9%) was supplied by Air Liquide

Shanghai Co., $H_2$ gas (99.999%) and $CH_4$ gas (99.9%) were supplied in compressed gas cylinders from Nanjing Nanyuan Gas company with limited liability, $CO_2$ gas (99.9%) was supplied by Nanjing Shangyuan Industrial Gas Plant, and $N_2$ gas (99.999%) was supplied by Nanjing Special Gas Plant.

## Fabrication of pristine Matrimid membranes

A raw Matrimid dope solution was prepared by dissolving pure Matrimid powder (0.1 g) in DMF (4.0 g) solvent, stirred for 10 h and sonicated for 2 h. The clear Matrimid dope solution was obtained by filtering the raw Matrimid dope solution through a 1.2 μm PTFE syringe. The clear dope solution was then directly poured into a clean Petri-dish by a syringe and placed in a vacuum oven to heat from room temperature to 120 °C for 8 h to evaporate the solvent. The original membrane, which still contained a small amount of DMF solvent, was peeled off, and a 14 mm diameter circular membrane was cut from the original membrane, which was submerged in methanol for 12 h to remove the residual DMF from the membrane by solvent exchange. Then, the membranes were placed in a Petri-dish and dried in a vacuum oven at 120 °C for 8 h to obtain membrane samples that could be tested directly with a thickness of 20 ± 5 μm.

## Fabrication of the Matrimid-Cyclen mixed matrix membranes

A raw Matrimid dope solution was prepared by dissolving 0.1 g pure Matrimid powder in 4.0 g DMF solvent, stirred for 10 h and sonicated for 2 h. The clear Matrimid dope solution was obtained by filtering the raw Matrimid dope solution through a 1.2 μm PTFE syringe. Cyclen powder (1.0, 3.0, 5.0, 10.0 or 15.0 mg) were added to 6.0 g DMF solvent, and the mixture was stirred for 10 h and sonicated for 2 h to obtain clear and transparent solutions.

The clear Matrimid dope solution was added to the Cyclen/DMF solution, stirred for 24 h and sonicated for 1 h to obtain homogeneous Matrimid/Cyclen solutions, which were then injected into a clean Petri-dish by a syringe and then transferred into a vacuum oven at 120 °C for 8 h solvent evaporation (Fig. S14). The residual DMF was similarly removed from the Matrimid-Cyclen membranes by solvent exchange with methanol. The as-fabricated samples were named as Matrimid-Cyclen-n% (n% is the Cyclen mass loading and was varied from 1 to 15 wt%). These Cyclen-incorporated mixed matrix membranes had an average thickness of 25 ± 5 μm.

## Fabrication of the Matrimid-Hexacyclen membrane, Matrimid-Piperazine membrane and Matrimid-Cyclodecane membrane

A raw Matrimid dope solution was prepared by dissolving 0.1 g pure Matrimid powder in 4.0 g DMF solvent, stirred for 10 h and sonicated for 2 h. The clear Matrimid dope solution was obtained by filtering the raw Matrimid dope solution through a 1.2 μm PTFE syringe. Hexacyclen/Piperazine/Cyclodecane powder (1.0 mg) were added to 6.0 g DMF solvent, and the mixture was stirred for 10 h and sonicated for 2 h to obtain clear and transparent solutions. The clear Matrimid dope solution was added to the Hexacyclen/Piperazine/Cyclodecane solution, stirred for 24 h and sonicated for 1 h to obtain homogeneous Matrimid/Hexacyclen solutions, which was then injected into a clean Petri-dish by a syringe and was then transferred into the 120 °C vacuum oven at ambient temperature for 8-h solvent evaporation. The residual DMF was similarly removed from the Matrimid-Hexacyclen/Matrimid-Piperazine/Matrimid-Cyclodecane membranes by solvent exchange with methanol. The as-fabricated samples were named as Matrimid-Hexacyclen-1% membrane, Matrimid-Piperazine-1% membrane and Matrimid-Cyclodecane-1% membrane.

## Fabrication of TFC membranes

A raw Matrimid dope solution was prepared by dissolving 0.4 g pure Matrimid powder in 16.0 g DMF solvent stirred for 10 h and sonicated for 2 h. The clear Matrimid dope solution was obtained by filtering the raw Matrimid dope solution through a 1.2 μm PTFE syringe. Cyclen powder (20.0 mg) were added to 24.0 g DMF solvent, and the mixture was stirred for 10 h and sonicated for 2 h to obtain clear and transparent solutions. The clear Matrimid dope solution was added to the Cyclen/DMF solution, stirred for 24 h and sonicated for 1 h to obtain homogeneous Matrimid/Cyclen solutions. The solution was then cast onto the anodic aluminum oxide (AAO) support using a bar-coater with a uniform speed and then put into a vacuum oven at 120 °C for 2 h solvent evaporation. The as-fabricated samples were named as Matrimid-Cyclen-n (where n represents the membrane thickness). The membrane thickness was measured using cross-sectional SEM imaging.

## Characterizations

The surface and cross-sectional morphologies of the pristine Matrimid and Matrimid-Cyclen-n% membranes were observed by scanning electron microscope (SEM, GeminiSEM 450). The chain packing structure and d-spacing of the Matrimid and Matrimid-Cyclen-n% membranes were examined by a Bruker D8 Advance X-ray diffractometer (XRD) with the X-ray radiation source being Cu K- (1.54 Å) and the diffraction angles ranged from 10˚ to 25˚ with a scan speed of 5°/min. The average d-spacing was evaluated based on the Bragg's Law ($n\lambda = 2dsin\theta$), where $n$ is an integer (1, 2, 3), $\lambda$ represents the wavelength of the X-ray radiation source, $d$ means the average interchain/intersegmental distance between adjacent polymer chains in semi-crystalline regions (d-spacing) and $\theta$ denotes the X-ray diffraction angle. Fourier transform infrared spectroscopy (FTIR) (Bruker TENSOR II) was used to analyze the chemical structures of pristine Matrimid and Matrimid-Cyclen-n% membranes through attenuated total reflection (ATR) mode with the spectra over the wavenumber range of 800–4000 cm$^{-1}$. Thermal stability of the pristine Matrimid and Matrimid-Cyclen-n% membranes was measured by a thermogravimetric analyzer (TGA, Q5000IR) in the temperature range of 25–800 °C with a heating rate of 10 °C/min under a flowing $N_2$ atmosphere (50 mL min$^{-1}$). X-ray photoelectron spectroscopy (XPS, ESCALAB 250Xi) equipped with a monochromatic Al Kα X-ray source (hν = 1486.6 eV) of 12 kV and 150 W was used to detect the binding signals of key functionalities and interactions in the membrane samples. The dynamic mechanical thermal analysis (DMTA) was conducted for Matrimid-Cyclen samples (20 mm long, and 5 mm wide) on a Mettler Toledo DMTA at a heating rate of 3 °C/min from 30 to 350 °C with a load frequency of 1 Hz in air. The Brunauer-Emmett-Teller (Quantachrome Autosorb IQ) technique was used to measure the pore structure of the prepared membrane. The membrane has been freeze-dried for 10 h before degassing. Membrane samples were degassed under a high vacuum at 120 °C for 8 h before analysis and Cyclen was degassed under a high vacuum at 80 °C for 8 h before analysis.

## Molecular dynamics (MD) simulation

MD simulation was performed using the Large-scale Atomic/Molecular Massively Parallel Simulator (LAMMPS 23) software together with the pysimm program[63]. The amorphous 3D membrane models were constructed using the PACKMOL package[64]. The models were used with a periodic boundary condition and the cut-off radius was set to 12 Å, and the Ewald method was used to describe the long-range interaction force[65]. Before the MD simulation, the NPT simulation was performed to form a stable system. In detail, the system was first annealed under the NPT (P = 101 KPa) ensemble for 20 ps at circulating temperatures ranging from 300 to 500 K. The MD simulation of the system was performed under the NVT ensemble (T = 300 K) for 500 ps to obtain the final structures. All initial models were firstly energy-minimized by using steepest descent algorithm, and the temperature and pressure were controlled by Nose-Hoover thermostat and Parrinello-Rahman barostat, respectively. Visualization and image rendering were performed using OVITO software[66].

## Gas permeation tests of membranes

Gas permeation tests of the membranes were conducted on a variable-pressure constant-volume gas permeation cell. The dense membrane was mounted onto the permeation cell and vacuumed at 25 °C for 12 h before the gas permeation test was carried out. Gas permeability was tested following the order of He, $H_2$, $CO_2$, $N_2$ and $CH_4$ corresponding with the order of gas kinetic diameter (He: 2.60 Å, $H_2$: 2.89 Å, $CO_2$: 3.30 Å, $N_2$: 3.64 Å and $CH_4$: 3.80 Å) from smallest to largest where the pressure was 1 bar. The cell temperature was kept constant at 25 °C. The gas permeability coefficient through the membrane was calculated according to the steady state pressure increment ($dp/dt$) via the following Eq. (1):

$$P = \frac{273 \times 10^{10}}{760} \frac{V\,l}{A\,T\left(p_2 \times \frac{76}{14.7}\right)} \frac{dp}{dt} \quad (1)$$

where $P$ denotes the gas permeability coefficient in Barrer (1 Barrer = $1 \times 10^{-10}$ cm³ (STP)·cm·cm$^{-2}$·s$^{-1}$·cmHg$^{-1}$), $V$ refers to the volume of the downstream reservoir (cm³), $A$ is the effective membrane area (cm²), $l$ represents the membrane thickness (cm), $T$ is the testing temperature (K) and $p_2$ is defined as the upstream pressure of the system (atm). The permeability tests were repeated at least three times and the average deviation obtained was less than 5%.

The ideal selectivity between two different gases across a polymeric membrane is the ratio of their single gas permeability coefficient as described in the following Eq. (2):

$$\alpha_{A/B} = \frac{P_A}{P_B} \quad (2)$$

where $P_A$ and $P_B$ refer to the permeability coefficients of gases $A$ and $B$, respectively.

For TFC membrane, the gas separation parameters, including gas permeance ($J$) and separation selectivity, were calculated according to the following Eq. (3):

$$J = \frac{P}{l} \quad (3)$$

Where $J$ denotes the gas permeance of the TFC membrane in GPU (1 GPU = $1 \times 10^{-6}$ cm³ (STP)•cm$^{-2}$·s$^{-1}$•cmHg$^{-1}$), $P$ denotes the gas permeability coefficient in Barrer (1 Barrer = $1 \times 10^{-10}$ cm³ (STP) •cm•cm$^{-2}$·s$^{-1}$•cmHg$^{-1}$), $l$ represents the thickness of membrane selective-layer (μm).

The gas permeation ideal selectivity ($\alpha_{A/B}$) is obtained by the following Eq. (4):

$$\alpha_{A/B} = \frac{J_A}{J_B} \quad (4)$$

where $J_A$ and $J_B$ refer to the permeance of gases $A$ and $B$, respectively.

## Data availability

The authors declare that all the data supporting the findings of this study are available within the article (and Supplementary Information files). Additional data are available from the corresponding author upon request. Source data are provided with this paper.

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

## Acknowledgements

The authors sincerely acknowledge the financial support provided by the National Key Research and Development Program of China

(2025YFF0522700 [J.L.]), the National Natural Science Foundation of China (22478371 [J.L.]), and the Students' Innovation and Entrepreneurship Foundation of USTC (XY2024C008 [W.H.], CY2025C002B [W.H.]). This work was partially carried out at the Instrument Center for Physical Science, University of Science and Technology of China.

## Author contributions

Wen He: Designed the research and conceived the idea, carried out laboratory research, wrote draft of manuscript. Xiangzeng Wang: Contributed to writing of theory section of manuscript. Jian Guan: Conducted the XPS measurements. Quansheng Liang: Helped with the graph drawing. Ji Ma: Conducted the SEM measurements. Ying Liu: Commented on the manuscript. Hongjun Zhang: Discussed the results and commented on the manuscript. Chunwei Zhang: Discussed the results. Jiangtao Liu: Optimized the research and conceived the idea, led the analysis, discussed the results and commented on the manuscript. All authors analyzed and discussed the results.

## Competing interests

The authors declare no competing interests.
