## [Transparent Peer Review file · Nature Communications]

Mixed-matrix membranes with molecular recognition windows for selective helium extraction from natural gas

Corresponding Author: Professor Jiangtao Liu

Version 0:

Reviewer comments:

Reviewer #1

(Remarks to the Author)

The manuscript reports on the combination of cyclen and Matrimid into a mixed matrix membrane that has high selectivity for gas separation. Both cyclen and Matrimid have been studied separately, and there exists a lot of literature on both materials. Hence, the novelty being presented is the combination of the two into a membrane. The resulting selectivities are very high, which gives me concern, as generally for mixed matrix membrane systems, selectivity is aligned with either of the membrane or additive, and not an enhancement. I can understand the argument for the strong binding between cyclen and Matrimid, but that does not necessarily translate into the high selectivities reported. There is no reported error for the experimental results and no discussion on reproducibility, so this raises questions about the provided data.

I am also concerned about the high cyclen loading in the membrane, up to 15%. There is no discussion about mechanical properties in the manuscript, but the literature on Matrimid is very clear that high levels of additives result in a very brittle membrane. I see no reason why this is not the case here. This has issues with the ability to translate the performance into a viable membrane.

The operating temperature, feed pressure and aging behaviors follows the expected trend of mixed matrix membranes, with the presented discussion valid.

The application for helium extraction is important, and it is good that the authors considered both separation from methane and nitrogen.

I recommend that the manuscript be rejected, and the authors submit to a more appropriate journal.

Reviewer #2

(Remarks to the Author)

• What are the noteworthy results?

The work seems to be looking at high selectivity membranes for He/CH₄ separation, by altering the properties of a commonly used polyimide in gas separation, Matrimid, adding different concentrations of Cycle, which is presented as a macrocycle (particle, molecule? Please clarify) with a wide opening providing H-bonding with the Matrimid polymer chains. Both the porosity of this filler and the crosslinking thereof seems to be having a significant influence in increasing the selectivity of Matrimid above the state-of-the-art of such membrane materials without decreasing the permeability of the smaller gas molecules.

All statements in the scientific paper should be justified, either by results presented in the work, or by references to previous reports in state-of-the-art literature. See for instance lines 193-196, "The ultrahigh gas separation performance of mixed matrix membrane attributed to enhanced interfacial compatibility due to dense hydrogen bonding between Cyclen and Matrimid chains", before showing any gas separation results.

The authors often do not comment on the hypotheses until they are about to introduce the results (see lines 307-311 introducing the effect of operating temperature, pressure and "aging").

Already in the Highlights, why do the authors talk of cyclen as a "window" and not an amine molecule with such and such properties that give the potential to provide the Matrimid polyimide matrix with the desired transport/separation properties in the He/CH₄ separation? Please, be more precise.

- Will the work be of significance to the field and related fields? How does it compare to the established literature? If the work is not original, please provide relevant references.

Given the low quantity of He in the feed stream it is reasonable that the authors look at Matrimid because of its low permeability and high selectivity properties and try this hybridization to increase the pore size and therefore the permeability of the Matrimid polymer. The state-of-the-art does very vaguely support this justification, and only reading between lines with prior knowledge of mixed matrix membrane technology and challenges allows the reader to withdraw the novelty of the work. This is probably due to the fact that the bibliographic review of the authors is not mentioned until Figure 5 where the data on Matrimid mixed matrix membranes collected in the supplementary information is used to build the Robeson upper-bound trade-off figures for the different gas pairs considered in this work. Please revise.

- Does the work support the conclusions and claims, or is additional evidence needed?

Besides, but no other information on the morphology and porosity characteristics justifying the authors' selection as filler in Matrimid polyimide are presented, so the reader cannot judge for himself the reasons why the crosslinking only appear at loadings above 15%, or the presence or absence of crosslinking with the polymer matrix that leads to higher selectivity and usually decreases permeability, or the "torn lamellar structures" formation, as well as the evidence of the strong H-bonding mentioned in the Results section.

Have authors envisaged the measuring of BET surface area of Cyclen and the MMM for verification? I guess they have done, and the Cyclen is a sacrificial filler given the statement in page This could visualize the statement of the authors " in lines 214 – 216, "the decomposition of Cyclen leaves many micropores in the Matrimid-Cyclen membranes", but this porosity has not been shown in the paper, or is not revealed in the cross-sectional images in SEM pictures. Please clarify, this is important to show evidence on the trade-off between crosslinking and additional porosity effects on the mixed matrix membrane performance observed in the results section.

- Is the methodology sound? Does the work meet the expected standards in your field?

The membrane materials were characterized by the common range of analytical techniques in this kind of work: SEM (surface and cross section), XRD, ATR-FTIR, NMR, TGA, XPS, as well as single gas permeation at room temperature by time-lag experiments at 25°C. These are adequate in membrane materials characterization. A but lays in the fact that the authors claim to evaluate the effect of temperature and temperature in permeation in the range 25-45°C and 1-6 bar. These are very narrow in gas permeation membranes. Although it is shown an impact in selectivity in this narrow temperature range, the value of 25°C cannot be considered as "low temperature" standard (line 327). Likewise, 6 bar is the regular operating pressure in gas separation. Please consult previous works such as that by R. Baker, studying the optimal pressure for gas separation as 5 bar. The pressure of 6 bar in a strong polyimide as Matrimid is not usually sufficient enough to see any observable packing density of the polymer chains (line 336). The authors should give evidence of that, or appropriately check literature on polyimide membrane materials characterization. The same can be said of the effect of aging, which in this work is attributed to the measurements performed with the membranes stored away from the gas permeation plant up to 110 days after the synthesis. Have the membranes been taken out of the module in this intervals? The damage caused by the joints may be more significant that any physical or chemistry change due to a slight variation of the polymer packing. Please revise.

- Are there any flaws in the data analysis, interpretation and conclusions? Do these prohibit publication or require revision?

The commentary, though, seems to be mingling these characterization results since the authors speak of diffraction peaks regarding the absorption bands in the ATR -FTIR spectra (line 191). Please correct. Also, I think for amorphous polymers such as Matrimid, it is more correct to talk about diffraction ands as well in the XRD, since the polymer diffracts in broad bands whose width is given information regarding the morphology and opening and crosslinking of the polymer chains that influence the membrane performance afterwards. Please revise.

Besides, other comments, according to the TGA curves presented, Cyclen does not provide thermal stability to the Matrimid membrane materials, since it is degraded at a low temperature. Therefore, the statement where the second stage of the TGA of MMM at 300-450°C attributed to Cyclen degradation should be revised for clarification.

Lines 257, please provide evidence to such statements.

Line 267, please specify to which membrane composition corresponds to these three % values.

Line 292, again, it is not clear whether is selectivity enhancement the property the authors are seeking for or permeability?

Why is a macrocycle with additional porosity chosen as fillers then, and Matrimid, a polyimide known for its selectivity and low gas permeability? Please clarify.

Please revise the experimental gas permeation values given in Tables S2-S5, since the values for permeation at 25°C and 1 atm in Tables S2 and S3, and tables S4 and S5, do not agree. Have the authors measured the same membrane materials several times? The information on reproducibility is very important. Please comment.

Line 299, the authors attribute now that too many Cyclen molecules tend to agglomerate at high loading, but no such agglomeration was shown in the SEM images above. Please provide evidence to these conclusions.

Line 353, the plots are the 2008 Robeson upper-bounds for the target gas pair separations. Please be specific.

Line 382, the authors compare suddenly the Cyclen -Matrimid membranes of this work with additional Cyclodecane-hybrid membranes, whose synthesis and characterization has not been elsewhere described in this manuscript. Please revise and include, either experimental evidence or reported reference.

Lines 384-394, again these statements need additional evidence to verify that the supposedly hypothesis that Cyclen hybridization of Matrimid is providing porosity and crosslinking thus inducing remarkable increase in selectivity in Matrimid towards He gas separation, is fulfilled or not.

English revised by professional native translator recommended. There are several wrong verb tenses and other grammar or semantic mistakes that could easily be avoided. For example, consider the use of the adjective "residual" instead of "resident" in line 201, comparisons wrongly used, such as "with a higher filler content increased" (line 293), "shrined" (line 348) instead

of “shrunked”, and so on.

Small notations to be corrected, for instance: Figure 4c, the permeability changes are not shown in figure 4c, but the single gas permeability in Barrer.

Line 249, the gas permeability measurement is a time-lag variable pressure- constant volume setup, I guess, since the starting pressure is 1 bar, what is the vacuum pressure?

The selectivity in Table S2 is the intrinsic selectivity in the table below are obtained as the ratio between two single gas permeabilities from the table above.

• Is there enough detail provided in the methods for the work to be reproduced?

I have included my comments through the review according to the journal requests. The aim of the paper and the gas permeation improvement of Matrimid membranes are significant and worthwhile but the manuscript must be improved.

Reviewer #3

(Remarks to the Author)

See file attached.

[Editorial Note: See end of file]

Version 1:

Reviewer comments:

Reviewer #1

(Remarks to the Author)

The authors have not adequately addressed the reviewers comments and have not improved the quality of their manuscript. This manuscript should not be accepted, as it does not meet the standards of the journal.

Reviewer #4

(Remarks to the Author)

[Note from the Editor: reviewer #4 was asked to assess the response given to reviewer #2 who was not able to look over the revision.]

This study reports the preparation of mixed matrix membranes by incorporating Cyclen molecules into Matrimid polyimide, and explores their application in gas separation. The approach of membrane development is certainly innovative. I have carefully reviewed the revised manuscript and the authors' responses to the previous reviewers' comments. The authors have provided a comprehensive set of characterization data and presented a plausible explanation for the polymer-filler interactions—primarily through hydrogen bonding—which may play a critical role in enhancing gas selectivity. Most of the previous concerns have been addressed appropriately. In my view, the revised manuscript could be considered for publication in Nature Communications, but the authors must address the remaining concerns regarding data accuracy and reproducibility, especially the membranes showed record-high gas separation selectivity.

1. A key concern raised by previous reviewers pertains to the unusually record-high gas separation selectivity of the Matrimid-Cyclen mixed matrix membranes. I share Reviewer 1's skepticism about the record-high selectivities reported. I strongly recommend that the authors re-evaluate their results and, if possible, validate the membrane performance through collaboration with other membrane research groups, either in China or internationally, to rule out any inadvertent errors.

2. The authors should carefully re-examine their raw data and the time-lag profiles. Based on my understanding, Matrimid typically exhibits low gas permeability for CH₄ and N₂ (<0.5 Barrer). In this study, the reported CH₄ permeability is below 0.1 Barrer, for instance, the Matrimid-Cyclen-5% membrane shows a CH₄ permeability of only 0.04 Barrer, which further declines to 0.001 Barrer after aging. At such low permeability values, the pressure increase on the permeate side would be extremely slow. In my experience, accurate measurement in this regime is highly sensitive to experimental conditions, particularly potential gas leakage. Notably, the reported vacuum pressure of 0.006579 Torr (equivalent to ~0.877 Pa) is relatively high for reliable time-lag measurements. The addition of nanofillers can increase gas sorption, prolonging the time needed to reach steady-state permeation. This can lead to underestimation of the slope and thus an artificially low gas permeability. Furthermore, leakage from the atmosphere could distort measurements. Did the authors account for and subtract gas leakage? This could be a critical source of error. The authors should revisit their raw data and provide representative time-lag profiles, including slope fittings, for key samples in the Supplementary Information. The authors should also provide more experimental details of their time-lag rig, including the accuracy of the pressure sensor.

3. Given the reported high performance, I strongly recommend that the authors fabricate thin-film composite (TFC) membranes and measure gas permeance and selectivity for both single gases and gas mixtures, ideally with gas

chromatography. The Matrimid-Cyclen solution should be suitable for casting onto porous polymeric or AAO supports. With selective layer thicknesses below 1 micron, or ideally down to ~100 nm, the gas permeance should be significantly higher, and the influence of leakage on time-lag measurements substantially reduced. Such tests would help confirm whether the TFC membranes maintain high gas permeance and selectivity (e.g., for He/CH₄ or CO₂/CH₄ mixtures). This is crucial, as the low permeability of thick films may introduce significant measurement error.

4. Building on the points above, the authors are encouraged to test their approach using other polymer matrices with higher intrinsic gas permeability. This would help determine whether similar enhancements can be achieved and reduce the susceptibility to measurement error.

5. A central hypothesis is that the Cyclen macrocycle facilitates gas transport through its cavity and narrows the polymer interchain spacing via hydrogen bonding. The authors performed control experiments with Cyclododecane and Hexacyclen. However, it would strengthen their case to include additional controls using amine-containing molecules without macrocyclic cavities, such as piperazine.

6. The three-dimensional plots in Figure 4 are difficult to interpret. A clearer presentation would be to convert them to two-dimensional plots.

Reviewer #5

(Remarks to the Author)

[Note from the Editor: reviewer #5 was asked to assess the response given to reviewer #3 who was not able to look over the revision.]

The authors have carefully addressed the comments raised from the previous review round - the manuscript now has met the standards of Nature Communications and it can be considered for publication.

Version 2:

Reviewer comments:

Reviewer #4

(Remarks to the Author)

The authors have made commendable efforts and have addressed most of the previous comments satisfactorily. However, some results still require deeper analysis. In particular, the thin-film composite (TFC) membrane data are of high significance and should be presented both in the main manuscript and in the Supporting Information. The experimental design could also be strengthened by preparing additional control samples, such as a TFC membrane made from Matrimid without the filler. The membranes should be carefully analyzed, as the TFC results are critical in determining whether the proposed membrane fabrication strategy can be scaled up for industrial applications.

Many studies on gas separation membranes report gas permeability and selectivity; however, they rarely fabricate membranes in the form of thin-film composites, and even fewer evaluate mixed-gas separation performance. In my opinion, this represents a key bottleneck that has limited the progress and industrial relevance of gas separation membranes. Since the authors have now developed highly selective materials, they should prepare TFC versions and thoroughly assess their performance.

1. The authors prepared TFC membranes by coating a Matrimid/Cyclen-5% solution onto PAN and PES supports. These membranes show very high gas permeance (>10,000 GPU) (e.g. 30,000 GPU on PAN) and high He/CH₄ selectivity (≈100). However, these values appear too good to be true, as the reported permeance is unusually high for this type of system. The authors should carefully verify these measurements and rule out possible experimental or data interpretation errors. Furthermore, they should evaluate the permeance and selectivity for other gases (e.g., H₂, CO₂, N₂) to provide a more comprehensive understanding of the membrane's gas separation performance.

2. Such high permeance values are typically observed for ultrafiltration supports coated with a PDMS gutter layer. Did the authors apply a PDMS coating on the polymeric supports? This may allow the authors to prepare good quality TFC membranes.

3. The gas permeance of the bare supports, as well as that of the TFC membranes made from pure Matrimid (without filler) and from Matrimid/Cyclen (with filler), should be reported for comparison.

4. The authors should clarify why the membranes on polymeric supports exhibit a selectivity of approximately 100, whereas the membrane coated on AAO-2 shows much lower permeance but much higher selectivity (≈689). How many samples were tested to obtain these results, and what are the associated uncertainties? Although the precise permeance values are not given, they appear from the plots to be around 600 GPU, with a selective layer thickness of about 1 μm, which is relatively thick. If the other TFC membranes have similar selective-layer thicknesses, their corresponding gas permeabilities would exceed 10,000 Barrer while maintaining a selectivity of 100. This seems inconsistent with the low pure-gas permeability values (He permeability <100 Barrer) of the dense films, and therefore the authors should double-check the

results.

5. If the selective-layer thickness were reduced to around 100 nm, would it be possible to achieve higher permeance while maintaining high selectivity?

6. The authors are also encouraged to evaluate the mixed-gas performance of the TFC membranes, which would provide more realistic insight into their industrial potential.

7. More detailed characterization of the TFC membranes is strongly recommended, including both surface and cross-sectional imaging, along with accurate measurement of the selective-layer thickness. These data should be used to calculate the gas permeability, enabling direct comparison between permeance and permeability. The corresponding experimental methods and analysis procedures should also be clearly described.

Version 3:

Reviewer comments:

Reviewer #4

(Remarks to the Author)

The authors have made considerable efforts to address the reviewer's comments, and most of the concerns have been satisfactorily resolved. I am particularly pleased to see that the authors prepared thin-film composite membranes with varied thickness and systematically measured gas permeance and selectivity. The results are reasonable. These additional experiments significantly strengthen the manuscript and greatly improve the overall quality of the work.

The new results confirm that my previous concerns were valid and, importantly, they have been clearly and convincingly explained in the revised manuscript. Overall, the revisions have substantially enhanced the rigor, clarity, and impact of the study. The manuscript is now much improved and suitable for publication.

Minor comment: The title of the manuscript could be further improved for clarity and specificity. For example, "Mixed-matrix membranes with molecular recognition windows for selective helium extraction from natural gas" would more accurately reflect the content and focus of the work.

Response to the Reviewers' Comments:

Reviewer #1 (Remarks to the Author):

The manuscript reports on the combination of cyclen and Matrimid into a mixed matrix membrane that has high selectivity for gas separation. Both cyclen and matrimid have been studied separately, and there exists a lot of literature on both materials. Hence, the novelty being presented is the combination of the two into a membrane. The resulting selectivities are very high, which gives me concern, as generally for mixed matrix membrane systems, selectivity is aligned with either of the membrane or additive, and not an enhancement. I can understand the argument for the strong binding between cyclen and Matrimid, but that does not necessarily translate into the high selectivities reported.

Reply: Many thanks for your valuable comments. While it is true that both Cyclen and Matrimid have been extensively studied separately, their combination into a mixed matrix membrane represents a new approach to achieve high gas separation performance, especially for He/CH₄ selectivity. The Matrimid membrane is widely studied for gas separation, while its low permeability (33 Barrer) and moderate selectivity (86) still present a great challenge to extract Helium(He) from natural gas (ultra-low helium concentration). The cyclen (1, 4, 7, 10-tetraazacyclododecane) is one of the most extensively studied ligands in coordination chemistry (Shinoda, S. Dynamic cyclen-metal complexes for molecular sensing and chirality signaling. Chem. Soc. Rev. 2013, 42, 1825-1835.), while its potential for He extraction from natural gas has not been explored until now. The ultra-low concentration of He makes it very difficult to extract from natural gas. Therefore, it's of great significance to develop the Matrimid-Cyclen membranes with high He/CH₄ selectivity (>1000). In this work, the the combination of Matrimid and Cyclen (Matrimid-Cyclen-5%) membranes demonstrates high He permeability (66 Barrer) and He/CH₄ selectivity (>1660).

We fully understand your concern regarding the relationship between selectivity and the materials involved. The enhancement in selectivity is not simply additive, but from the strong hydrogen bonding interactions between Cyclen and Matrimid, which modifies the polymer chain packing and results in a tighter interchain structure. This leads to a more effective sieving mechanism. Usually, polymer membranes have good processability, high stability, and low cost, but they often suffer from plasticization problems, physical aging issues, and intrinsic permeability-selectivity trade-off limitations, which makes it a great challenge to obtain high permeability together with sufficient selectivity ^[1-4]. Molecular sieve materials have attractive transport properties well above the polymer upper bound, but they tend to have poor membrane-forming and mechanical property ^[5]. Mixed matrix membranes incorporate molecular sieve materials with certain pore sizes into a processable polymer matrix to improve the gas selectivity and permeability of polymer membranes by integrating the advantages of both ^[6-8], which is the significance of preparing hybrid matrix membranes. Therefore, the mixed matrix membranes (Matrimid-Cyclen) combine the advantages of easy

processibility from Matrimid polymer and high gas permeability/selectivity from molecular sieve material (Cyclen).

Table 1 summarizes the He separation performance of Matrimid-based membranes. The Matrimid-Cyclen-5% membranes exhibit high He permeability (66 Barrer) and He/CH₄ selectivity (>1660), which far exceed most of the other Matrimid-based membranes.

Table 1. Summary of literature He/N₂ and He/CH₄ separation performance.

Membrane Name	He Permeability (Barrer)	He/N ₂ Selectivity	He/CH ₄ Selectivity	Operating conditions	Gas Type	Reference
Matrimid-Cu-BTC-R-40	66.40	265.80	369.10	5 atm, 35 °C	Pure	1
Matrimid-Cu-BTC-30	51.80	193.40	257.90	5 atm, 35 °C	Pure	2
Matrimid-Cu-BDC-15%	20.50	341.70	410.00	7 atm, 35 °C	Pure	3
Matrimid-MgO-40%	45.00	86.1	118.4	10 atm, 35 °C	Pure	4
Matrimid-CNFs-10%	16.10	89.50	179.50	20 atm, 35 °C	Pure	5
Matrimid-PIM-EA(H ₂)-TB-50%	197.00	28.84	21.55	1 atm, 25 °C	Pure	6
Matrimid-p-xylenediamine-CL-14	21.70	112.00	155.00	10 atm, 35 °C	Pure	7
Matrimid-C60-2.5%	21.00	91.70	148.80	10 atm, 35 °C	Pure	8
Matrimid-Cyclen-3%	57.57±0.21	428.35±2.15	959.50±4.67	1 atm, 25 °C	Pure	This work
Matrimid-Cyclen-5%	66.43±0.22	621.70±3.20	1660.75±5.25	1 atm, 25 °C	Pure	This work
Matrimid-Cyclen-10%	78.63±0.28	543.43±4.034	1310.50±4.67	1 atm, 25 °C	Pure	This work

Supplementary References

- [1] A. Akbari, J. Karimi-Sabet, S.M. Ghoreishi, Intensification of helium separation from CH₄ and N₂ by size-reduced Cu-BTC particles in Matrimid matrix, *Separation and Purification Technology*, 251, 117317 (2020).
- [2] A. Akbari, J. Karimi-Sabet, S.M. Ghoreishi, Matrimid® 5218 based mixed matrix membranes containing metal organic frameworks (MOFs) for helium separation, *Chemical Engineering & Processing: Process Intensification*, 148, 107804 (2020).
- [3] A. Ali, J. Karimi-Sabet, S.M. Ghoreishi, Polyimide based mixed matrix membranes incorporating Cu-BDC nanosheets for impressive helium separation, *Separation and Purification Technology*, 253, 117430 (2020).

- [4] S.S. Hosseini, Y. Li, T.S. Chung, Y. Liu, Enhanced gas separation performance of nanocomposite membranes using MgO nanoparticles, *Journal of Membrane Science*, 302 (1-2), 207-217 (2007).
- [5] M. Dohade, Incorporation of carbon nanofibers into a Matrimid polymer matrix: Effects on the gas permeability and selectivity properties, *Journal of Applied Polymer Science*, 135 (12) 46019 (2018).
- [6] E. Esposito, I. Mazzei, M. Monteleone, A. Fuoco, M. Carta, N.B. Makeown, R. Malpass-Evans, J.C. Jansen, Highly permeable Matrimid®/PIM-EA (H₂)-TB blend membrane for gas separation, *Polymers*, 11 (1), 46 (2018).
- [7] P.S. Tin, T.S. Chung, Y. Liu, R. Wang, S.L. Liu, K.P. Pramoda, Effects of cross-linking modification on gas separation performance of Matrimid membranes, *Journal of Membrane Science*, 225(1-2), 77-90 (2003).
- [8] T.S. Chung, S.S. Chan, R. Wang, Z.H. Lu, C.B. He, Characterization of permeability and sorption in Matrimid/C60 mixed matrix membranes, *Journal of Membrane Science*, 211 (1), 91-99 (2003).

There is no reported error for the experimental results and no discussion on reproducibility, so this raises questions about the provided data.

Reply: Many thanks for your comments. Repeated experiments were conducted and the error bars have been included in the revised manuscript, The gas separation performance and fabrication of mixed matrix membrane with gas molecule recognition window (cyclen) was found to be reproducible. The specific data has been listed as follows.

Table 1. Summary of Gas transport properties of the Matrimid and Matrimid-Cyclen-n% membranes at 1.0 atm and 25 °C (n = 1-15% indicated the degree of Cyclen mass loading).

Membrane	Gas permeability (Barrer)				
	He	H ₂	CO ₂	N ₂	CH ₄
Matrimid	33.59±0.41	27.45±0.08	15.03±0.06	0.42±0.03	0.39±0.03
Matrimid-Cyclen-1%	47.87±0.24	43.04±0.16	16.90±0.06	0.30±0.02	0.13±0.01
Matrimid-Cyclen-3%	57.57±0.21	55.69±0.23	17.66±0.12	0.13±0.00	0.06±0.00
Matrimid-Cyclen-5%	66.43±0.22	62.17±0.32	20.07±0.08	0.10±0.00	0.04±0.00
Matrimid-Cyclen-10%	78.63±0.28	76.08±0.12	22.17±0.15	0.14±0.01	0.06±0.00
Matrimid-Cyclen-15%	85.93±0.80	80.85±0.30	24.76±0.07	0.24±0.02	0.13±0.01

Membrane	Idea Selectivity					
	He/N ₂	He/CH ₄	H ₂ /N ₂	H ₂ /CH ₄	CO ₂ /N ₂	CO ₂ /CH ₄

Matrimid	79.97±6.05	86.13±2.25	65.00±4.82	70.38±1.98	35.79±2.88	38.54±0.99
Matrimid-Cyclen-1%	145.06±1.029	368.23±2.8.10	143.47±9.71	331.08±26.34	56.33±3.81	130.00±1.033
Matrimid-Cyclen-3%	575.70±2.15	959.50±4.67	428.35±2.15	928.17±4.67	135.85±0.92	294.33±2.00
Matrimid-Cyclen-5%	664.30±2.10	1660.75±5.25	621.70±3.20	1554.25±8.00	200.70±0.80	501.75±3.75
Matrimid-Cyclen-10%	561.64±4.136	1310.50±4.67	543.43±40.34	1268.00±3.17	158.36±11.10	369.50±2.33
Matrimid-Cyclen-15%	358.04±1.228	661.00±4.8.42	336.88±11.59	621.92±49.33	103.17±3.64	190.46±1.471

Table 2. Summary of gas transport permeability and selectivity of the Matrimid-Cyclen-5% membrane at 1.0 atm and different temperatures.

Temperature (°C)	Gas permeability (Barrer)			Selectivity	
	He	N ₂	CH ₄	He/N ₂	He/CH ₄
Return to 20 °C	63.97±0.46	0.09±0.00	0.03±0.00	710.78±2.33	2132.33±7.00
0 °C	52.74±0.46	0.007±0.00	0.003±0.00	7534.29±24.28	17580.00±56.67
20 °C	63.13±0.46	0.10±0.00	0.03±0.00	631.30±2.60	2104.33±8.67
40 °C	80.14±0.46	0.32±0.02	0.10±0.01	250.44±15.13	801.40±83.82
60 °C	96.91±0.46	1.43±0.46	1.05±0.00	67.77±2.90	92.30±1.73
80 °C	125.12±0.46	3.45±0.46	3.05±0.46	36.27±0.36	41.02±0.87
100 °C	202.55±0.46	9.76±0.46	9.42±0.46	20.75±0.40	21.73±0.58
Return to 20 °C	68.69±0.27	0.12±0.01	0.03±0.00	572.42±49.85	2289.67±8.00

Table 3. Summary of gas transport permeability and selectivity of the Matrimid-Cyclen-5% membrane at different pressures and 25°C.

Pressure (Bar)	Gas permeability (Barrer)			Selectivity	
	He	N ₂	CH ₄	He/N ₂	He/CH ₄
1 Bar	66.43±0.22	0.10±0.00	0.04±0.00	664.30±2.10	1660.75±5.25
5 Bar	66.12±0.20	0.09±0.00	0.04±0.00	734.67±2.22	1652.00±6.00
10 Bar	65.48±0.13	0.10±0.01	0.04±0.00	654.80±1.30	1637.00±3.25

15 Bar	67.08±0.32	0.11±0.01	0.05±0.00	609.82±132.85	1341.60±6.40
20 Bar	68.34±0.57	0.13±0.01	0.07±0.00	525.69±39.06	976.29±8.15
25 Bar	68.28±0.34	0.12±0.02	0.06±0.00	569.00±73.14	1137.67±19.33
30 Bar	72.21±1.33	0.10±0.00	0.01±0.005	722.10±13.70	1444.20±130.02
Return to 1 Bar	66.84±0.42	0.10±0.00	0.04±0.00	668.40±4.20	1671.00±10.50

Table 4. Summary of gas transport properties of the Matrimid and Matrimid-Cyclen-5% membrane at 1.0 atm and 25 °C during the aging process.

Membrane	Gas permeability (Barrer)				
	He	H ₂	CO ₂	N ₂	CH ₄
Fresh	66.43±0.22	62.17±0.32	20.07±0.08	0.10±0.00	0.04±0.00
Aging for 7 days	64.22±.34	59.90±0.23	18.30±0.07	0.09±0.01	0.04±0.00
Aging for 20 days	59.96±0.53	54.68±0.14	16.67±0.17	0.07±0.005	0.02±0.00
Aging for 40 days	59.73±0.09	53.39±0.33	15.96±0.09	0.06±0.00	0.01±0.00
Aging for 110 days	54.31±0.10	50.76±0.10	14.12±0.03	0.05±0.00	0.008±0.00
Aging for 480 days	42.87±0.21	31.43±0.13	7.82±0.04	0.009±0.00	0.001±0.00

Membrane	Selectivity					
	He/N ₂	He/CH ₄	H ₂ /N ₂	H ₂ /CH ₄	CO ₂ /N ₂	CO ₂ /CH ₄
Fresh	664.30±2.1	1660.75±	621.70±	1554.25±	200.70	501.75±3
	0	5.25	3.20	8.00	±0.80	.75
Aging for 7 days	713.56±67.	1605.50±	665.56±	1497.50±	203.33	457.50±1
	960	8.50	68.04	21.22	±21.79	.42
Aging for 20 days	856.57±57.	2998.00±	781.14±	2734.00±	238.14	833.50±9
	74	26.50	54.20	6.76	±18.02	.66
Aging for 40 days	995.50±1.5	5973.00±	889.83±	5339.00±	266.00	1596.00±
	0	9.00	5.07	30.45	±1.33	7.94
Aging for 110 days	1086.20±2.	6788.80±	1015.20	6345.00±	282.40	1765.00±
	00	12.45	±1.74	10.90	±0.53	3.31
Aging for 480 days	4763.33±2	42870.00	3496.22	31430.00	868.89	7820.00±
	3.34	±200.00	±12.62	±113.58	±4.01	36.06

Figure 1 shows the preparation process of the Matrimid-Cyclen membrane, the dispersion of cyclen in Matrimid solutions is homogeneous. There is no Tyndall phenomenon in the clear and transparent Matrimid/Cyclen solutions, indicating that Cyclen is uniformly dispersed in the mixed solution. The reproducibility of a mixed

matrix membrane depends on the dispersibility of the filler, the higher the solubility of the filler and the better the dispersion in the mixed solution, the smaller the error in the performance of the membrane material.

Figure 1. Preparation process of the Matrimid-Cyclen membranes. (a) Cyclen powder (1.0, 3.0, 5.0, 10.0 or 15.0 mg) were added to 6.0 g DMF solvent to obtain clear and transparent solutions. (b) The clean Matrimid dope solution was added to the Cyclen/DMF solution to obtain homogeneous Matrimid/Cyclen solutions. (c) The Matrimid/Cyclen solutions was injected into clean petri dishes and transferred to a vacuum oven at an ambient temperature of 120 °C for 8-hour solvent evaporation to obtain Matrimid-Cyclen membranes.

I am also concerned about the high cyclen loading in the membrane, up to 15%. There is no discussion about mechanical properties in the manuscript, but the literature on Matrimid is very clear that high levels of additives result in a very brittle membrane. I see no reason why this is not the case here. This has issues with the ability to translate the performance into a viable membrane.

Reply: Thanks for your comments. We understand your concern about the potential brittleness of the Matrimid mixed matrix membranes with higher Cyclen loadings. To address this, the dynamic mechanical thermal analysis (DMTA) was conducted on Matrimid-Cyclen samples to assess the mechanical properties across a temperature range of 50 to 350 °C. As shown in Figure 3 (g-i), the storage modulus (E') of the Matrimid-Cyclen membranes (1800 MPa) was almost completely maintained before reaching the corresponding T_g . This indicated that the membrane retains its mechanical integrity even with higher Cyclen content. Furthermore, we observed that higher Cyclen loadings did increase in the brittleness of the membranes, as seen in the decreasing peak intensity of damping ($\text{Tan } \delta$) with increasing Cyclen content (Figure 3i). These data demonstrated that this kind of gas separation membrane was endowed with good mechanical property and thermal stability, even with the higher Cyclen loading.

Figure 3. Characterization of Matrimid-Cyclen membranes. (g) The dynamic mechanical thermal analysis (DMTA) spectra of Matrimid-Cyclen membranes at temperature range from 50 to 350 °C. (h) The storage modulus (E') and (i) Tan δ of Matrimid-Cyclen membranes.

The operating temperature, feed pressure and aging behaviors follows the expected trend of mixed matrix membranes, with the presented discussion valid. The application for helium extraction is important, and it is good that the authors considered both separation from methane and nitrogen. I recommend that the manuscript be rejected, and the authors submit to a more appropriate journal.

Reply: Thank you very much for your professional comments and suggestions. We sincerely apologize for some inadequate in data analysis and descriptions in the original manuscript. As you pointed out, several issues required further attention, and we have made substantial revisions to address the concerns. Below is a detailed list of our responses to your comments. In our work, we designed the mixed matrix membranes based on several key considerations derived from numerous studies.

Firstly, Cyclen was selected as filler due to its size-sieving effect, which is beneficial for He separation. Based on the molecular dynamic diameters of He (2.60 Å), H₂ (2.89 Å), CO₂ (3.30 Å), N₂ (3.64 Å) and CH₄ (3.80 Å), the Cyclen with cavity size (3.2-3.4 Å) between the molecular dynamic diameters of He and N₂ was selected, which makes it an ideal candidate for He separation, such as He/N₂ and He/CH₄. Furthermore, the Cyclen with a certain cavity size creates a special transport channel for small gases, like He, which increases their transport paths in the membrane, counteracting the negative effect of the increased chain stacking density and enhancing the permeability of small gas molecules.

Secondly, the chosen filler must have good solubility and dispersion in solvents. To ensure uniform dispersion of the filler in the membrane and to minimize the formation of non-selective voids caused by filler agglomeration, we selected Cyclen for its high solubility and good dispersion in organic solvents. Cyclen was fully dissolved in DMF, and then the clean Matrimid dope solution was added to the Cyclen/DMF solution. The resulting homogeneous Matrimid/Cyclen solution, which was stirred for 24 hours and sonicated for 1 hour, remained clear and transparent, even with up to 15% Cyclen content, as shown in Figure 1.

Figure 1. (a) Cyclen powder (1.0, 3.0, 5.0, 10.0 or 15.0 mg) were added to 6.0 g DMF solvent to obtain clear and transparent solutions. (b) Matrimid dope solution. (c) The clean Matrimid dope solution was added to the Cyclen/DMF solution to obtain clear and transparent mixed Matrimid/Cyclen solutions.

Thirdly, compatibility between filler and polymer matrix. It is crucial to create mixed matrix membranes with fair matching between filler and polymer matrix to gain attractive performance. This is why we chose Cyclen as a filler. Hydrogen bonding between Cyclen and the Matrimid polymer chains immobilizes Cyclen molecules in the interchain gaps of the Matrimid membrane, resulting in homogeneously mixed matrix membranes. The SEM image (Figure 2) shows the surface and interface of the membrane, confirming the good interfacial compatibility between Cyclen and Matrimid.

Figure 2. SEM images of the cross section of the Matrimid-Cyclen-n% membranes up to $\times 50,000$ magnification (n% is the Cyclen mass loading and was varied from 1 to 15 wt%, n = 0, 1, 3, 5, 10, 15).

Fourthly, Helium (He) purification requirements. Helium is an extremely scarce, non-renewable gas with a strong safety profile that is an essential resource for the development of many high-tech industries, such as the aerospace industry, electronics, medical equipment, and scientific research. On Earth, He occurs in very low concentrations, mainly in the atmosphere and in natural gas. Helium is commonly extracted industrially from high-pressure natural gas. However, the initial concentration of He in most natural gas reservoirs is still as low as 0.3%, and to extract helium from natural gas with ultra-low He content, the gas separation membranes must possess He/CH₄ selectivity greater than 1000. Highly permeable polymers rarely provide sufficient selectivity for high-purity He extraction, and Matrimid was chosen as the polymer matrix due to its dense stacking of polymer chains. XRD characterization revealed that the strong hydrogen bonding between Cyclen and Matrimid modulated the chain stacking of the mixed matrix membranes, which reduce the pore size, narrow the pore size distribution and enhances the selectivity of the membrane by modulating the chain spacing.

We have verified the success of the hybrid matrix membrane design with ultra-high helium selectivity to achieve excellent separation performance.

Reviewer #2 (Remarks to the Author):

- What are the noteworthy results?

The work seems to be looking at high selectivity membranes for He/CH₄ separation, by altering the properties of a commonly used polyimide in gas separation, Matrimid, adding different concentrations of Cyclen, which is presented as a molecule (particle, molecule? Please clarify) with a wide opening providing H-bonding with the Matrimid polymer chains. Both the porosity of this filler and the crosslinking thereof seems to be having a significant influence in increasing the selectivity of Matrimid above the state-of-the-art of such membrane materials without decreasing the permeability of the smaller gas molecules.

Reply: Thanks for your recognition and comments. In this work, the cyclic Cyclen with specific intra-ring dimensions was incorporated into Matrimid as a pore structure modifier to improve the He/CH₄ selectivity. We explored both inorganic or organic fillers for fabricating the MMMs to enhance the gas separation performance. While, the inorganic fillers (e.g., MOFs, ZIFs, and Mxene) often suffer from poor compatibility with polymer matrix, and the agglomeration results in defects in the membrane. Notably, Cyclen, a purely organic additive has better compatibility with Matrimid polymer matrices than inorganic fillers.

As shown in Figure 1, even with Cyclen loading increased from 1% to 15%, the tyndall effect never appeared in the Cyclen solution and the Cyclen/Matrimid mixture, indicating that Cyclen was bound to Matrimid at the molecular level.

Figure 1. (a) Cyclen powder (1.0, 3.0, 5.0, 10.0 or 15.0 mg) were added to 6.0 g DMF solvent to obtain clear and transparent solutions. (b) Matrimid dope solution. (c) The Matrimid dope solution was added to the Cyclen/DMF solution to obtain clear and transparent mixed Matrimid/Cyclen solutions.

This molecular dispersion of Cyclen within the membrane was further confirmed by SEM and EDS analysis. The cross-section morphology of Matrimid-Cyclen-n% (n = 1, 3, 5, 10, 15) membranes shown in Figure 2, at different magnifications, revealed an increase in torn lamellar structures as Cyclen loading increases. This morphology is attributed to the enhanced interaction between the polymer chains and filler, which results in interfacial stress during the fracturing of MMMs in liquid nitrogen for SEM analysis. This stress can deform the polymer segments, leading to increased plastic deformation. The Matrimid-Cyclen-15% membrane material remained homogeneous at 50,000x magnification, without any significant aggregation or discrete particulate phases observed. The elemental mapping (Figure 3) further supports the homogeneous distribution of Cyclen within the membrane, showing a gradual increase in nitrogen (N) elemental distribution as Cyclen content rises, without any noticeable aggregation. This is further evidence that Cyclen is dispersed as a molecule in polymer membrane.

Figure 2. SEM images of the cross section of the Matrimid-Cyclen-n% membranes up to ×50,000 magnification (n% is the Cyclen mass loading and was varied from 1 to 15 wt%, n = 0, 1, 3, 5, 10, 15).

Figure 3. SEM images and EDS with nitrogen (N) mapping of planar scans of the cross section of the Matrimid-Cyclen-n% membranes (n% is the Cyclen mass loading and was varied from 1 to 15 wt%, n = 0, 1, 3, 5, 10, 15).

In this work, Cyclen was selected as the filler for several key reasons to prepare the Matrimid matrix, which can lead to ultra-high He permeability and ultra-high He/CH₄ separation selectivity for several main reasons:

Firstly, Cyclen is an organic macrocyclic compound with a flexible structure and its cavity size is approximately 3.2-3.4 Å. This cavity size lies between the molecular dynamic diameters of He (2.60 Å) and N₂ (3.64 Å), producing a strong molecular sieving effect, particularly for He/N₂ and He/CH₄.

Secondly, the strong hydrogen bonds between Cyclen and Matrimid modulate the chain spacing increasing the chain stacking density of the mixed matrix membranes. This modification enhances the resistance to mass transfer for larger gases, such as N₂ and CH₄, improving the overall selectivity of the membrane.

Thirdly, Cyclen with a certain cavity size creates a special transport channel for small gases, like He, which increases their transport paths in the membrane, counteracting the negative effect of the increased chain stacking density and enhancing the permeability of small gas molecules.

Herein MMMs with ultra-high He/CH₄ separation selectivity, exceeding the performance of most existing membrane, were prepared using a simple solution casting method. The He/CH₄ selectivity reached up to an impressive value of 6788 after physical aging for 110 days, which outperformed almost all reported polymer-based membranes and was even comparable to that of some advanced carbon molecular sieve (CMS) membranes. In addition, compared with CMS membrane, MOF membrane and COF membrane with high He/CH₄ selectivity, the MMMs have good solution processability and stable performance, which is easy to achieve industrial production.

All statements in the scientific paper should be justified, either by results presented in the work, or by references to previous reports in state-of-the-art literature. See for instance lines 193-196, “The ultrahigh gas separation performance of mixed matrix membrane attributed to enhanced interfacial compatibility due to dense hydrogen bonding between Cyclen and Matrimid chains”, before showing any gas separation results.

Reply: Many thanks for your valuable comments. We have revised the statements according to your suggestion by adding the references after the expression. Please refer to the revised manuscript for the updated text. The sentence has been revised as “*The dense hydrogen bonding between Cyclen and Matrimid enhances their interfacial compatibility, reduces non-selective interfacial gaps and significantly improves the high gas separation performance of the mixed matrix membranes* ^[21, 32]”.

The authors often do not comment on the hypotheses until they are about to introduce the results (see lines 307-311 introducing the effect of operating temperature, pressure and “aging”).

Reply: Thanks for your comments. The hypotheses were deleted according to your suggestion. Please refer to the revised manuscript.

Already in the Highlights, why do the authors talk of Cyclen as a “window” and not an amine molecule with such and such properties that give the potential to provide the Matrimid polyimide matrix with the desired transport/separation properties in the He/CH₄ separation? Please, be more precise.

Reply: Thanks for your comments. We agree with your description that the Cyclen is an amine molecule from the perspective of its chemical structure. We would like to clarify our use of the term “window” in the context of its role in the membrane.

Firstly, as an amine molecule, Cyclen forms the strong hydrogen bonding with the Matrimid polymer, which tightens the *d*-spacing of Matrimid polymer chains and minimizes defects within the membrane. This arrangement leads to longer and more tortuous transport pathways for the kinetically larger diameters of N₂ and CH₄ in the membrane.

Secondly, and more importantly, Cyclen acts as pore structure modifier with a cavity size of approximately 3.2-3.4 Å, which allows Cyclen to create dedicated transport channels for He molecules providing a “window” for smaller gas molecules, such as He, to pass through more easily. The size of this “window” induces a strong molecular sieving effect, reducing the permeability of N₂ and CH₄ and increasing the selectivity of multiple important gas pairs.

Therefore, the presence of Cyclen plays a dual role in the MMM, i.e., as a “window” and an amine molecule to construct the molecular sieve membrane for He/N₂ and He/CH₄ separation.

• Will the work be of significance to the field and related fields? How does it compare to the established literature? If the work is not original, please provide relevant references.

Reply: Thank you for recognizing the significance of this work in He purification from natural gas and related fields.

The Matrimid membrane has been widely studied for He separation, while its low He permeability (33 Barrer) and He/CH₄ selectivity (86) still meet a great challenge to extract He from natural gas (ultra-low helium concentration). The Cyclen (1, 4, 7, 10-tetraazacyclododecane) is one of the most extensively studied ligands in coordination chemistry (Shinoda, S. Dynamic Cyclen-metal complexes for molecular sensing and chirality signaling. *Chem. Soc. Rev.* 2013, 42, 1825–1835.), but it has not been explored for He extraction from natural gas till now. The ultra-low concentration of He makes it very difficult to be extracted from natural gas, highlighting the significance of developing Matrimid-Cyclen membranes with high He/CH₄ selectivity (>1000). In this work, the combination of Matrimid and Cyclen (Matrimid-Cyclen-5%) membranes demonstrated high He permeability (66 Barrer) and He/CH₄ selectivity (>1660).

The Matrimid-based membranes and related works for He separation were summarized in Table 1. Compared with the existing literature, Matrimid-Cyclen membranes offer moderate He permeability and ultra-high He/CH₄ selectivity, surpassing the performance of all other Matrimid-based membranes (Figure 1). Notably, this is the first report in where the cyclic Cyclen molecules are incorporated into Matrimid as molecular windows to prepare hybrid matrix membranes for He purification.

Figure 1. Plots of (a) He/N₂, (b) He/CH₄ separation performance of Matrimid-Cyclen membranes compared with other Matrimid-based membranes.

Table 1. Summary of literature He/N₂ and He/CH₄ separation performance.

Membrane Name	He Permeability (Barrer)	He/N ₂ Selectivity	He/CH ₄ Selectivity	Operating conditions	Gas Type	Reference
Matrimid-Cu-BTC-R-40	66.40	265.80	369.10	5 atm, 35 °C	Pure	1
Matrimid-Cu-BTC-30	51.80	193.40	257.90	5 atm, 35 °C	Pure	2
Matrimid-Cu-BDC-15%	20.50	341.70	410.00	7 atm, 35 °C	Pure	3

Matrimid-MgO-40%	45.00	86.1	118.4	10 atm, 35 °C	Pure	4
Matrimid-CNFs-10%	16.10	89.50	179.50	20 atm, 35 °C	Pure	5
Matrimid-PIM-EA(H ₂)- TB-50%	197.00	28.84	21.55	1 atm, 25 °C	Pure	6
Matrimid-p-xylenediami ne-CL-14	21.70	112.00	155.00	10 atm, 35 °C	Pure	7
Matrimid-C60-2.5%	21.00	91.70	148.80	10 atm, 35 °C	Pure	8
Matrimid-Cyclen-3%	57.57±0.2 1	428.35±2. 15	959.50±4. 67	1 atm, 25 °C	Pure	This work
Matrimid-Cyclen-5%	66.43±0.2 2	621.70±3. 20	1660.75± 5.25	1 atm, 25 °C	Pure	This work
Matrimid-Cyclen-10%	78.63±0.2 8	543.43±4 0.34	1310.50± 4.67	1 atm, 25 °C	Pure	This work

Supplementary References

- [1] A. Akbari, J. Karimi-Sabet, S.M. Ghoreishi, Intensification of helium separation from CH₄ and N₂ by size-reduced Cu-BTC particles in Matrimid matrix, *Separation and Purification Technology*, 251, 117317 (2020).
- [2] A. Akbari, J. Karimi-Sabet, S.M. Ghoreishi, Matrimid[®] 5218 based mixed matrix membranes containing metal organic frameworks (MOFs) for helium separation, *Chemical Engineering & Processing: Process Intensification*, 148, 107804 (2020).
- [3] A. Ali, J. Karimi-Sabet, S.M. Ghoreishi, Polyimide based mixed matrix membranes incorporating Cu-BDC nanosheets for impressive helium separation, *Separation and Purification Technology*, 253, 117430 (2020).
- [4] S.S. Hosseini, Y. Li, T.S. Chung, Y. Liu, Enhanced gas separation performance of nanocomposite membranes using MgO nanoparticles, *Journal of Membrane Science*, 302 (1-2), 207-217 (2007).
- [5] M. Dohade, Incorporation of carbon nanofibers into a Matrimid polymer matrix: Effects on the gas permeability and selectivity properties, *Journal of Applied Polymer Science*, 135 (12) 46019 (2018).
- [6] E. Esposito, I. Mazzei, M. Monteleone, A. Fuoco, M. Carta, N.B. Makeown, R. Malpass-Evans, J.C. Jansen, Highly permeable Matrimid[®]/PIM-EA (H₂)-TB blend membrane for gas separation, *Polymers*, 11 (1), 46 (2018).
- [7] P.S. Tin, T.S. Chung, Y. Liu, R. Wang, S.L. Liu, K.P. Pramoda, Effects of cross-linking modification on gas separation performance of Matrimid membranes, *Journal of Membrane Science*, 225(1-2), 77-90 (2003).
- [8] T.S. Chung, S.S. Chan, R. Wang, Z.H. Lu, C.B. He, Characterization of permeability and sorption in Matrimid/C60 mixed matrix membranes, *Journal of Membrane Science*, 211 (1), 91-99 (2003).

Given the low quantity of He in the feed stream it is reasonable that the authors look at Matrimid because of its low permeability and high selectivity properties and try this

hybridization to increase the pore size and therefore the permeability of the Matrimid polymer. The state-of-the-art does very vaguely support this justification, and only reading between lines with prior knowledge of mixed matrix membrane technology and challenges allows the reader to withdraw the novelty of the work. This is probably due to the fact that the bibliographic review of the authors is not mentioned until Figure 5 where the data on Matrimid mixed matrix membranes collected in the supplementary information is used to build the Robeson upper-bound trade-off figures for the different gas pairs considered in this work. Please revise.

Reply: Many thanks for your comments. Matrimid and Matrimid-based mixed matrix membranes have been extensively studied for gas separation and pervaporation processes due to the combination of relatively high gas permeability coefficients and separation factors as well as excellent mechanical properties, high solubility in non-hazard organic solvents, and high glass transition temperature in comparison to polycarbonate, polysulfone, and other materials. Due to the excellent properties of the shown by Matrimid membrane, which is desirable for the gas separation process, it is worthwhile to explore modifications to further enhance their performance for He separation.. In response to your suggestion, we have added the rationale for choosing Matrimid as the polymer matrix in the “Introduction” section of the manuscript. We have also summarized the relevant studies on Matrimid-based membrane materials for He separations, discussed the challenges associated with mixed-matrix membrane technology, and clarified the idea and novelty and purpose of our design. These revisions aim to better explain the basis and significance of our work. Please refer to the introduction in the revised manuscript.

Does the work support the conclusions and claims, or is additional evidence needed? Besides, but no other information on the morphology and porosity characteristics justifying the authors’ selection as filler in Matrimid polyimide are presented, so the reader cannot judge for himself the reasons why the crosslinking only appear at loadings above 15%, or the presence or absence of crosslinking with the polymer matrix that leads to higher selectivity and usually decreases permeability, or the “torn lamellar structures” formation, as well as the evidence of the strong H-bonding mentioned in the Results section.

Reply: We apologize for any inadequate data analysis and descriptions in the manuscript, which may have made some of this evidence unclear. We have tried our best to resolve the concerns and made a proper revision of the manuscript. A detailed list of responses to the comments is provided below.

Additional experiments and evidence related to morphology and porosity characteristics have been supplemented. The SEM images in Figure 1 shows that the cross-section morphology of the MMMs is dense and homogeneous, demonstrating excellent interfacial compatibility between Cyclen and Matrimid.

As Cyclen loading increases, the torn lamellar structures in the membranes also increase. This morphology is attributed to the enhanced interaction between the Matrimid and Cyclen, which results in the formation of interfacial stress during the fracturing of MMMs in liquid nitrogen. This stress causes deformation of the polymer

segments, resulting in elongated polymer segments with increased plastic deformation. There is no significant aggregation or discrete particulate phases observed in the Matrimid-Cyclen-15% membrane from 1,000x to 50,000x electron microscopy. The improved compatibility between organic fillers and polymer matrices is conducive to obtain simultaneous improvements in gas permeability and selectivity.

Figure 1. SEM images of the cross section of the Matrimid-Cyclen-n% membranes up to $\times 50,000$ magnification (n% is the Cyclen mass loading and was varied from 1 to 15 wt%, $n = 0, 1, 3, 5, 10, 15$).

The pore size distribution of the Matrimid-Cyclen membranes in Figure 2 shows that the addition of Cyclen significantly reduced the micropore size of the Matrimid-Cyclen membrane. Additionally, the content of ultramicropores with a size of about 3.9 \AA , gradually increased compared to the original Matrimid membrane. The introduction of Cyclen also resulted in the formation of new pores with a size of about 3.2 \AA . This modification of the pore structure is attributed to the hydrogen bonding interaction between Cyclen and Matrimid which enhances the molecular sieving properties of Matrimid-Cyclen membranes by narrowing the pore size distribution. Cyclen creates additional and longer transport channels for He, which significantly improves the transport capacity of He.

Figure 2. Pore size distribution of the Matrimid-Cyclen membranes. The pore size distribution was calculated from the corresponding H₂ adsorption isotherm using the DFT method.

The presence of strong hydrogen bonding interactions was further confirmed by FTIR spectra, which shows that the hydrogen bonding between Cyclen and Matrimid polymer chains immobilizes Cyclen molecules within the interchain gaps of the Matrimid matrix, resulting in homogeneous mixed matrix membranes.

Have authors envisaged the measuring of BET surface area of Cyclen and the MMM for verification? I guess they have done, and the Cyclen is a sacrificial filler given the statement in page This could visualize the statement of the authors “ in lines 214 – 216, “the decomposition of Cyclen leaves many micropores in the Matrimid-Cyclen membranes”, but this porosity has not been shown in the paper, or is not revealed in the cross-sectional images in SEM pictures. Please clarify, this is important to show evidence on the trade-off between crosslinking and additional porosity effects on the mixed matrix membrane performance observed in the results section.

Reply: Thanks for your comments. The pore size distribution of the Matrimid-Cyclen membranes in Figure 1 shows that the addition of Cyclen significantly reduced the micropore size of the Matrimid-Cyclen membrane. The content of ultramicropores with a size of about 3.9 Å, gradually increased compared to the original Matrimid membrane. This modification of the pore structure is attributed to the hydrogen bonding interaction between Cyclen and Matrimid which enhances the molecular sieving properties of Matrimid-Cyclen membranes by narrowing the pore size distribution. Additionally, Cyclen with pore size (3.2 Å) creates additional and longer transport channels for He in the membranes, which significantly improves the transport capacity of He. Carbon molecule sieving (CMS) membrane after thermal treatment will be evaluated in the future.

Figure 1. BET surface area and pore size analysis for Cyclen and Matrimid-Cyclen membranes. (a) H₂ sorption isotherm curves collected at 77K and (b) Corresponding pore size distribution profiles. The pore size distribution was calculated from the corresponding H₂ adsorption isotherm using the DFT method.

Is the methodology sound? Does the work meet the expected standards in your field? The membrane materials were characterized by the common range of analytical techniques in this kind of work: SEM (surface and cross section), XRD, ATR-FTIR, NMR, TGA, XPS, as well as single gas permeation at room temperature by time-lag experiments at 25°C. This are adequate in membrane materials characterization. A but lays in the fact that the authors claim to evaluate the effect of temperature and temperature in permeation in the range 25-45°C and 1-6 bar. These are very narrow in gas permeation membranes. Although it is shown an impact in selectivity in this narrow temperature range, the value of 25°C cannot be considered as “low temperature” standard (line 327). Likewise, 6 bar is the regular operating pressure in gas separation. Please consult previous works such as that by R. Baker, studying the optimal pressure for gas separation as 5 bar. The pressure of 6 bar in a strong polyimide as Matrimid is not usually sufficient enough to see any observable packing density of the polymer chains (line 336). The authors should give evidence of that, or appropriately check literature on polyimide membrane materials characterization. The same can be said of the effect of aging, which in this work is attributed to the measurements performed with the membranes stored away from the gas permeation plant up to 110 days after the synthesis. Have the membranes been taken out of the module in this intervals? The damage caused by the joints may be more significant that any physical or chemistry change due to a slight variation of the polymer packing. Please revise.

Reply: Thanks for your comments. Following your suggestion, the extended gas separation test of the Matrimid-Cyclen membranes was carried out including a wider range from 0 °C to 100 °C to better assess the impact of operating temperature on the gas separation performance. The results of the gas separation performance of the Matrimid-Cyclen-5% membrane are shown in Figure 1, with the data summarized in Table 1.

As the operating temperature increased from 0 °C to 100 °C, the permeability of He, N₂ and CH₄ increased, while the selectivity of He/N₂ and He/CH₄ decreased. This behavior

could be attributed to the increased gas fugacity at elevated temperatures. The selectivity of He/N₂ and He/CH₄ decreased rapidly as the operating temperature increased from 40 °C to 60 °C. This phenomenon was likely due to the accelerated movement frequency of polymer chains at high temperatures, rendering transport of gas molecules through the matrix and weakening the molecular sieving effect of the membrane. Notably, after cooling the membrane from 25 °C to 0 °C, the He permeability decreased from 63.13 to 52.74 Barrer, while the He/N₂ selectivity increased from 631 to 8790, and the He/CH₄ selectivity increased from 2104 to 17580, corresponding to 14-fold and 9-fold increases, respectively. This dramatic improvement in selectivity at low temperatures significantly enhances the potential for high-purity He extraction from natural gas.

The ultra-high selectivity of the Matrimid-Cyclen-5% membrane at 0 °C for He/N₂ and He/CH₄ facilitates the extraction of high-purity He from natural gas at low temperatures. Although the He permeability decreased by 16% at low temperatures, the increase in He selectivity indicates that improving gas purity is more challenging than enhancing transport efficiency. High purity is particularly important for He, given its critical applications in the aerospace industry, electronics, medical equipment, and scientific research. Furthermore, the gas separation performance of the membranes fully when the temperature was returned to 20 °C, both from 100 °C and from 0 °C, indicating that the changes in membrane performance with temperature are inherent to the membrane properties and do not result in structural damage.

Figure 1. Effect of operating temperature on the separation performance of Matrimid-Cyclen-5% membrane.

Table 1. Summary of gas transport permeability and selectivity of the Matrimid-Cyclen-5% membrane at 1.0 Bar and different temperatures.

Temperature (°C)	Gas permeability (Barrer)			Selectivity	
	He	N ₂	CH ₄	He/N ₂	He/CH ₄
Return to 20 °C	63.97±0.46	0.09±0.00	0.03±0.00	710.78±2.33	2132.33±7.00

0 °C	52.74±0.46	0.007±0.00	0.003±0.00	7534.29±24.28	17580.00±56.67
20 °C	63.13±0.46	0.10±0.00	0.03±0.00	631.30±2.60	2104.33±8.67
40 °C	80.14±0.46	0.32±0.02	0.10±0.01	250.44±15.13	801.40±83.82
60 °C	96.91±0.46	1.43±0.46	1.05±0.00	67.77±2.90	92.30±1.73
80 °C	125.12±0.46	3.45±0.46	3.05±0.46	36.27±0.36	41.02±0.87
100 °C	202.55±0.46	9.76±0.46	9.42±0.46	20.75±0.40	21.73±0.58
Return to 20 °C	68.69±0.27	0.12±0.01	0.03±0.00	572.42±49.85	2289.67±8.00

To explore the practical applicability of the Matrimid-Cyclen membranes at different feed pressures, the gas separation performance of the Matrimid-Cyclen-5% membrane was re-evaluated at pressures up to 30 bar, and the data were summarized in Table 2. As presented in Figure 2, as the pressure increased from 1 bar to 10 bar, the permeability of He decreased slightly, and the pressure dependence of N₂ and CH₄ was almost unrecognizable, with only small fluctuations in the selectivity for He/N₂ and He/CH₄. These observations suggest that the membranes remain structurally stable under moderate pressure increases, which may be attributed to the presence of Cyclen molecules, which increase the packing density of the polymer chains, thereby reducing their mobility, increasing chain stiffness and improving the high-pressure resistance of the membranes. As the operating pressure increased from 10 bar to 20 bar, the permeability of all gases rose, while the selectivity for He/N₂ and He/CH₄ decreased. This behavior is typical of glassy polymers, such as Matrimid, under plasticization pressure, where increased feed pressure leads to higher polymer segment mobility and larger chain spacing, resulting in higher permeability and lower selectivity [1-5]. As the pressure increases from 20 bar to 30 bar, the permeability of He remained almost constant, while the permeability of N₂ and CH₄ gradually decreased, and the selectivity of He/N₂ and He/CH₄ tends to increase, which was ascribed to the existence of unrelaxed volume within the polymer chain structure. An increase in feed pressure results in greater compactness of the polymer matrix, which reduces the gas permeability in the membrane by reducing fractional free volume [6-7].

Supplementary References

- [1] H. Rajati, A.H. Navarchian, D. Rodrigue, S. Tangestaninejad, Improved CO₂ transport properties of Matrimid membranes by adding amine-functionalized PVDF and MIL-101 (Cr), *Separation and Purification Technology*, 235 116149 (2020).
- [2] S. Salman, K. Nijmeijer, High pressure gas separation performance of mixed-matrix polymer membranes containing mesoporous Fe (BTC), *Journal of membrane science*, 459, 33-44 (2014).
- [3] S. Salman, K. Nijmeijer, Performance and plasticization behavior of polymer-MOF membranes for gas separation at elevated pressures, *Journal of membrane science*, 470, 166-177 (2014).

[4] A.E. Amooghin, M. Omidkhah, H. Sanaeepur, A. Kargari, Preparation and characterization of Ag⁺ ion-exchanged zeolite-Matrimid® 5218 mixed matrix membrane for CO₂/CH₄ separation, Journal of Energy Chemistry, 25 (3) 450-462 (2016).

[5] H. Asghar, A. Ilyas, Z. Tahir, X.F. Li, A.L. Khan, Fluorinated and sulfonated poly (ether ether ketone) and Matrimid blend membranes for CO₂ separation, Separation and Purification Technology, 203, 233-241 (2018).

[6] A.E. Amooghin, H. Sanaeepur, M. Omidkhah, A. Kargari, “Ship-in-a-bottle”, a new synthesis strategy for preparing novel hybrid host-guest nanocomposites for highly selective membrane gas separation, Journal of Materials Chemistry A, 6 (4) 1751-1771 (2018).

[7] A. Mirzaei, A.H. Navarchian, S. Tangestaninejad, Mixed matrix membranes on the basis of Matrimid and palladium-zeolitic imidazolate framework for hydrogen separation, Iranian Polymer Journal, 29 (6), 479-491 (2020).

Figure 2. Effect of feed gas pressures on the separation performance of Matrimid-Cyclen-5% membrane.

Table 2. Summary of gas transport permeability and selectivity of the Matrimid-Cyclen-5% membrane at different pressures and 25°C.

Pressure (Bar)	Gas permeability (Barrer)			Selectivity	
	He	N ₂	CH ₄	He/N ₂	He/CH ₄
1 Bar	66.43±0.22	0.10±0.00	0.04±0.00	664.30±2.10	1660.75±5.25
5 Bar	66.12±0.20	0.09±0.00	0.04±0.00	734.67±2.22	1652.00±6.00
10 Bar	65.48±0.13	0.10±0.01	0.04±0.00	654.80±1.30	1637.00±3.25
15 Bar	67.08±0.32	0.11±0.01	0.05±0.00	609.82±132.85	1341.60±6.40
20 Bar	68.34±0.57	0.13±0.01	0.07±0.00	525.69±39.06	976.29±8.15

25 Bar	68.28±0.34	0.12±0.02	0.06±0.00	569.00±73.14	1137.67±19.33
30 Bar	72.21±1.33	0.10±0.00	0.01±0.005	722.10±13.70	1444.20±130.02
Return to 1 Bar	66.84±0.42	0.10±0.00	0.04±0.00	668.40±4.20	1671.00±10.50

As shown in the following Figure 3, the aging test was performed by taking samples (labeled A_n, B_n, C_n, D_n and E_n, where n is the number of samples taken) from the same batch of the Matrimid-Cyclen-5% membrane at different aging times (0, 7, 20, 40, 110, 480 days). No damage was caused by the joints since the membrane samples removed at each interval will not be used for retesting.

Figure 3. Multiple small membranes are removed from the intact membrane to be used for testing.

Regarding the aging performance, we continued to test the gas separation performance of the membrane material after physical aging for 480 days, the data are summarized in Table 3. As shown in Figure 4, compared to the fresh membranes, the He permeability of the Matrimid-Cyclen-5% membrane decreased from 66.34 Barrer to 42.87 Barrer after 480 days of physical aging, a decrease of 35%. Meanwhile, the He/N₂ selectivity increased from 663.4 to 4763.3, and the He/CH₄ selectivity increased from 1660.75 to 42870.0, corresponding to increases of 718% and 2581%, respectively.

Figure 4. Long-term stability of the Matrimid-Cyclen-5% membrane (the whole membrane was stored in an ambient environment without the introduction of any protective gas at room temperature, small pieces of membrane were extracted from the whole membrane at different aging times for a gas separation performance test).

Table 3. Summary of gas transport properties of the Matrimid and Matrimid-Cyclen-5% membrane at 1.0 atm and 25 °C during the aging process.

Membrane Aging (Days)	Gas permeability (Barrer)				
	He	H ₂	CO ₂	N ₂	CH ₄
0	66.43±0.22	62.17±0.32	20.07±0.08	0.10±0.00	0.04±0.00
7	64.22±.34	59.90±0.23	18.30±0.07	0.09±0.01	0.04±0.00
20	59.96±0.53	54.68±0.14	16.67±0.17	0.07±0.005	0.02±0.00
40	59.73±0.09	53.39±0.33	15.96±0.09	0.06±0.00	0.01±0.00
110	54.31±0.10	50.76±0.10	14.12±0.03	0.05±0.00	0.008±0.00
480	42.87±0.21	31.43±0.13	7.82±0.04	0.009±0.00	0.001±0.00

Membrane Aging (Days)	Selectivity					
	He/N ₂	He/CH ₄	H ₂ /N ₂	H ₂ /CH ₄	CO ₂ /N ₂	CO ₂ /CH ₄
0	664.30±2.	1660.75±	621.70±3.	1554.25±	200.70±	501.75±
	10	5.25	20	8.00	0.80	3.75
7	713.56±67	1605.50±	665.56±6	1497.50±	203.33±	457.50±
	.960	8.50	8.04	21.22	21.79	1.42
20	856.57±57	2998.00±	781.14±5	2734.00±	238.14±	833.50±
	.74	26.50	4.20	6.76	18.02	9.66
40	995.50±1.	5973.00±	889.83±5.	5339.00±	266.00±	1596.00
	50	9.00	07	30.45	1.33	±7.94

110	1086.20±2 .00	6788.80± 12.45	1015.20± 1.74	6345.00± 10.90	282.40± 0.53	1765.00 ±3.31
480	4763.33±2 3.34	42870.00 ±200.00	3496.22± 12.62	31430.00 ±113.58	868.89± 4.01	7820.00 ±36.06

Are there any flaws in the data analysis, interpretation and conclusions? Do these prohibit publication or require revision?

Reply: Thanks for your comments. Generally speaking, the permeability and selectivity should be increased with higher Cyclen loading. However, it was found that the CO₂/N₂ selectivity of the Matrimid-Cyclen-15% membrane decreased when the Cyclen loading increased from 10% to 15%. This phenomenon may be attributed to the fact that the excess Cyclen is uniformly distributed between the Matrimid chains, and as a result, the Cyclen increases the chain spacing by its own volume, leading to an increase in interchain distance. Importantly, SEM and EDS analyses confirm that Cyclen is uniformly dispersed in the membrane, ruling out the possibility of agglomerating to form non-selective defects. This behavior suggests a balance between Cyclen concentration and membrane performance that requires further optimization.

The commentary, though, seems to be mingling these characterization results since the authors speak of diffraction peaks regarding the absorption bands in the ATR -FTIR spectra (line 191). Please correct. Also, I think for amorphous polymers such as Matrimid, it is more correct to talk about diffraction bands as well in the XRD, since the polymer diffracts in broad bands whose width is given information regarding the morphology and opening and crosslinking of the polymer chains that influence the membrane performance afterwards. Please revise.

Reply: Thanks for your comments. We have corrected this error according to your suggestion, changed “diffraction peaks” to “characteristic peaks” for FTIR and changed “diffraction bands” to “diffraction peaks” for XRD. Please refer to the revised manuscript.

Besides, other comments, according to the TGA curves presented, Cyclen does not provide thermal stability to the Matrimid membrane materials, since it is degraded at a low temperature. Therefore, the statement where the second stage of the TGA of MMM at 300-450°C attributed to Cyclen degradation should be revised for clarification.

Reply: Thanks for your comments. The TGA curves were revised for clarification in response to your suggestion.

In the first stage of the TGA, which occurs between 100-400 °C, the weight loss is attributed to the evaporation of residual organic solvent and the Cyclen degradation in the MMMs. The weight loss gradually increases with increasing Cyclen mass loading, reflecting the higher content of Cyclen in the membrane. The second weight loss, occurring in the range of 400 to 550 °C, is associated with the degradation of lateral methyl of polymer. The third stage is caused by the decomposition of the polymer main chains, which begins at approximately 550 °C.

Lines 257, please provide evidence to such statements.

Reply: Thanks for your comments. The evidence for our findings is supported by BET characterization of the membranes.

The pore size distribution of the Matrimid-Cyclen membranes shows that the addition of Cyclen resulted in the formation of new pores in the membrane, approximately 3.2 Å in size (Figure 1), which agrees well with the theoretical value of the molecular window of Cyclen. This indicates that Cyclen creates ultra-micropores capable of selectively recognizing gas molecules. The presence of these ultra-micropores creates more and longer transport channels for He, which significantly improves the transport capacity of He.

Figure 1. Pore size distribution of the Matrimid-Cyclen membranes. The pore size distribution was calculated from the corresponding H₂ adsorption isotherm using the DFT method.

Line 267, please specify to which membrane composition corresponds to these three % values.

Reply: Thanks for your comments. Compared with the Matrimid membrane, the He, H₂ and CO₂ permeability of the Matrimid-Cyclen-5% membrane increased by approximately 108%, 99% and 20%, respectively. Please refer to the revised manuscript.

Line 292, again, it is not clear whether is selectivity enhancement the property the authors are seeking for or permeability? Why is a macrocycle with additional porosity chosen as fillers then, and Matrimid, a polyimide known for its selectivity and low gas permeability? Please clarify.

Reply: Thanks for your comments. What we are aiming for is a simultaneous improvement in both permeability and selectivity, with priority given to improving selectivity. To achieve this, the macrocycle with additional porosity, specifically Cyclen, was chosen as filler for the following reasons:

Firstly, Cyclen was selected as filler due to its size-sieving effect, which is beneficial for He separation. Based on the molecular dynamic diameters of He (2.60 Å), H₂ (2.89

Å), CO₂ (3.30 Å), N₂ (3.64 Å) and CH₄ (3.80 Å), the Cyclen with cavity size (3.2-3.4 Å) between the molecular dynamic diameters of He and N₂ was selected, which makes it an ideal candidate for He separation, such as He/N₂ and He/CH₄. Furthermore, the Cyclen with a certain cavity size creates a special transport channel for small gases, like He, which increases their transport paths in the membrane, counteracting the negative effect of the increased chain stacking density and enhancing the permeability of small gas molecules.

Secondly, the chosen filler must have good solubility and dispersion in solvents. To ensure uniform dispersion of the filler in the membrane and to minimize the formation of non-selective voids caused by filler agglomeration, we selected Cyclen for its high solubility and good dispersion in organic solvents. Cyclen was fully dissolved in DMF, and then the clean Matrimid dope solution was added to the Cyclen/DMF solution. The resulting homogeneous Matrimid/Cyclen solution, which was stirred for 24 hours and sonicated for 1 hour, remained clear and transparent, even with up to 15% Cyclen content, as shown in Figure 1.

Figure 1. (a-b) Cyclen powder (1.0, 3.0, 5.0, 10.0 or 15.0 mg) were added to 6.0 g DMF solvent to obtain clear and transparent solutions. (c-d) The clean Matrimid dope solution was added to the Cyclen/DMF solution to obtain clear and transparent mixed Matrimid/Cyclen solutions.

Thirdly, compatibility between filler and polymer matrix. It is crucial to create mixed matrix membranes with fair matching between filler and polymer matrix to gain attractive performance. This is why we chose Cyclen, as a filler. Hydrogen bonding between Cyclen and the Matrimid polymer chains immobilizes Cyclen molecules in the interchain gaps of the Matrimid membrane, resulting in homogeneously mixed matrix membranes. The SEM image (Figure 2) shows the surface and interface of the membrane, confirming the good interfacial compatibility between Cyclen and Matrimid.

Figure 2. SEM images of the cross section of the Matrimid-Cyclen-n% membranes up to $\times 50,000$ magnification (n% is the Cyclen mass loading and was varied from 1 to 15 wt%, n = 0, 1, 3, 5, 10, 15).

Fourthly, Helium (He) purification requirements. Helium is an extremely scarce, non-renewable gas with a strong safety profile that is an essential resource for the development of many high-tech industries, such as the aerospace industry, electronics, medical equipment, and scientific research. On Earth, He occurs in very low concentrations, mainly in the atmosphere and in natural gas. Helium is commonly extracted industrially from high-pressure natural gas. However, the initial concentration of He in most natural gas reservoirs is still as low as 0.3%, and to extract helium from natural gas with ultra-low He content, the gas separation membranes must possess He/CH₄ selectivity greater than 1000. Highly permeable polymers rarely provide sufficient selectivity for high-purity He extraction, and Matrimid was chosen as the polymer matrix due to its dense stacking of polymer chains. XRD characterization revealed that the strong hydrogen bonding between Cyclen and Matrimid modulated the chain stacking of the mixed matrix membranes, which reduced the pore size, narrowed the pore size distribution and enhances the selectivity of the membrane by modulating the chain spacing..

In summary, it's a great significance to develop the Matrimid-Cyclen membrane with enhanced He permeability (66 Barrer) and high He/CH₄ selectivity (>1660).

Please revise the experimental gas permeation values given in Tables S2-S5, since the values for permeation at 25°C and 1 atm in Tables S2 and S3, and tables S4 and S5, do not agree. Have the authors measured the same membrane materials several times? The information on reproducibility is very important. Please comment.

Reply: Thanks for your comments. According to your suggestion, the membranes were re-prepared to test the performance under different temperature and pressure conditions. The updated gas separation performance is summarized in S3-S4, and the data in Tables S2-S5 is consistent now. Please refer to the revised manuscript.

The discrepancy in the permeation values at 25°C and 1 atm between Tables S2 and S3, Tables S4 and S5, because the aging data for the Matrimid-Cyclen-5% membranes in Table S2 were averaged values, as summarized in Table S5. Additionally, the changes in gas separation performance for the measured membranes as a function of temperature and pressure after 40 days of aging are summarized in Tables S3 and S4, respectively. Multiple measurements were not performed on the same piece of tested membrane, as the experimental procedure involved sampling different locations from different batches of Matrimid-Cyclen-5% membranes.

Sampling Procedure: Three small membrane sheets (named A1, B1 and C1) from five different preparation batches of Matrimid-Cyclen-5% membranes were taken for gas separation performance testing. Subsequently, these different preparation batches of Matrimid-Cyclen-5% membranes were re-sampled for different aging times, operating pressures, and operating temperature conditions for performance testing. The gas transport properties of the membrane materials were averaged over these different conditions.

Line 299, the authors attribute now that too many Cyclen molecules tend to agglomerate at high loading, but no such agglomeration was shown in the SEM images above. Please provide evidence to these conclusions.

Reply: Thanks for your comments. We have conducted additional SEM (higher magnification) and EDS analysis of the membrane materials to confirm whether the Cyclen molecules in the highly loaded Matrimid-Cyclen-n% membrane were agglomerated. The results showed that all the Matrimid-Cyclen membranes display homogeneous cross-sections and surfaces at 50,000x electron magnification, without any significant aggregation or discrete particulate phases observed (Figure 1 and Figure 2), confirming the presence of only a continuous single phase within these membranes. Additionally, energy dispersive X-ray spectroscopy (EDS) mapping of the cross-sections of the Matrimid-Cyclen membranes further verifies the uniform distribution of Cyclen as evidenced by the detection of characteristic nitrogen (N) signals (Figure 3).

Figure 1. SEM images of the surface of Matrimid-Cyclen-n% membranes up to ×50,000 magnification (n% is the Cyclen mass loading and was varied from 1 to 15 wt%, n = 0, 1, 3, 5, 10, 15).

Figure 2. SEM images of the cross section of the Matrimid-Cyclen-n% membranes up to $\times 50,000$ magnification (n% is the Cyclen mass loading and was varied from 1 to 15 wt%, n = 0, 1, 3, 5, 10, 15).

Figure 3. EDS cross-section with nitrogen (N), carbon(C) and oxygen (O) mapping of planar scans of Matrimid-Cyclen membranes (n% is the Cyclen mass loading and was varied from 1 to 15 wt%, n = 0, 1, 3, 5, 10, 15).

The XRD characterization showed a gradual increase in polymer chain spacing after $\geq 10\%$. We speculate that this may be due to the fact that the excess Cyclen is uniformly distributed between the Matrimid chains, and the Cyclen increases the chain spacing by its volume, rather than agglomerating to form non-selective defects between the polymer chains. This hypothesis is supported by SEM and EDS analyses, which show that Cyclen is uniformly dispersed in the membrane.

We have corrected these questions according to your suggestion. Please refer to the revised manuscript.

Line 382, the authors compare suddenly the Cyclen-Matrimid membranes of this work with additional Cyclodecane-hybrid membranes, whose synthesis and characterization has not been elsewhere described in this manuscript. Please revise and include, either experimental evidence or reported reference.

Reply: Thanks for your comments. The Cyclen, Hexacyclen, and Cyclodecane are derivatives of the organic macrocyclic. We have added the preparation procedure of the Matrimid-Hexacyclen membrane and Matrimid-Cyclodecane membrane in the Methods section according to your suggestion. Please refer to the revised manuscript.

Lines 384-394, again these statements need additional evidence to verify that the supposedly hypothesis that Cyclen hybridization of Matrimid is providing porosity and crosslinking thus inducing remarkable increase in selectivity in Matrimid towards He gas separation, is fulfilled or not.

Reply: Thanks for your comments. As shown in Figure 1, the micropore sizes in Matrimid, Matrimid-Cyclen-1% and Matrimid-Cyclodecane-1% membranes were around 6.9Å, 6.3Å and 7.1 Å, respectively. This suggests that the hydrogen bonding interaction between Cyclen and Matrimid indeed modulates the pore structure in the membranes to enhance the molecular sieving properties of Matrimid-Cyclen membranes by narrowing pore sizes. Compared with Matrimid-Cyclen-1% membranes, the Matrimid-Hexacyclen-1% membranes prepared with fillers with larger window sizes had larger micropore sizes, and their separation of He is weaker, demonstrating that the filler with suitable pore size is more effective in improving the separation performance of MMMs.

Figure 1. Pore size distribution of the mixed matrix membranes (Matrimid-Cyclen-1% membrane, Matrimid-Hexacyclen-1% membrane and Matrimid-Cyclodecane-1% membrane). The pore size distribution was calculated from the corresponding H₂ adsorption isotherm using the DFT method.

Here, the possible gas transport mechanism of Matrimid-Cyclen membranes is derived from two sets of comparison experiments. Then, molecular dynamics simulations and BET characterization of the membranes were used to determine the gas separation mechanism of the membranes.

The membrane structure model generated from MD simulation is based on PACKMOL software (Figure 6a). He and N₂ were used as the probe gases for the membrane model to analyze the gas accessible volume, the average gas transmembrane path and the interconnectivity of void space. Figure 6b-c shows that some channels allow He transport but block N₂. The increase in Cyclen content within the membranes resulted in a continuous enhancement in the connectivity of the He passable pore space within the membranes (Figure 6d), and the average He transmembrane path calculated on the basis of the passable pore space increased from 635 Å to 922 Å (Figure 6f), thus significantly improving the He permeability. In addition, the average transmembrane path length difference between He and N₂ increased from 22.58 Å to 331.34 Å and then decreased to 321.90 Å, indicating that the presence of Cyclen could create a special transport channel for He, increasing its transport paths and transmission path length in the membrane, effectively enhancing the transport capacity of He and the separation of gas pairs in the membrane.

The pore size distribution of the Matrimid-Cyclen membranes shows that the micropore size of the Matrimid-Cyclen membrane is significantly reduced, and the content of ultra-micropores with a size of about 3.9 Å is gradually increased compared to the original Matrimid membrane (Figure 6g). In addition, the addition of Cyclen resulted in the appearance of new pores in the membrane with a size of about 3.2 Å, which agrees well with the theoretical value of the molecular window of Cyclen.

Figure 6. Structural analysis of Matrimid-Cyclen membrane models. (a) 3D view of Matrimid-Cyclen-n% membrane models (n% is the Cyclen mass loading, n = 0, 5, 10). The gray and blue shades indicate the free volume detected by a probe with a radius of 2.6 Å. (b-c) Accessible void spaces for He (left) and N₂ (right) in Matrimid-Cyclen-n% membranes models (n% is the Cyclen mass loading, n = 0, 5, 10); The red circles in the left picture indicate additional accessible void spaces for He, compared with that of N₂. (d) Interconnected (green) and disconnected (red) voids in Matrimid-Cyclen-n% membrane models (n% is the Cyclen mass loading, n = 0,5,10) with respect to a probe of 1.3 Å radius. (e) The accessible volume of He and N₂ as a percentage of the membrane pore volume and the simulated density of the Matrimid-Cyclen membranes. (f) The average He and N₂ transmembrane path calculated based on accessible void space. (g) Pore size distribution of the Matrimid-Cyclen membranes. The pore size distribution was calculated from the corresponding H₂ adsorption isotherm using the DFT method.

English revised by professional native translator recommended. There are several wrong verb tenses and other grammar or semantic mistakes that could easily be avoided. For example, consider the use of the adjective “residual” instead of “resident in line 201, comparisons wrongly used, such as “with a higher filler content increased” (line 293), “shrined” (line 348) instead of “shrunked”, and so on.

Reply: Many thanks for your comments. The English was carefully revised by professional native specialist.

Small notations to be corrected, for instance: Figure 4c, the permeability changes are not shown in figure 4c, but the single gas permeability in Barrer.

Reply: Thanks for your comments. We have corrected this question according to your suggestion. The legend for Figure 4 (c-e) has been revised to the following:

Figure 4. Gas separation performance of Matrimid-Cyclen membranes. (c) He, N₂ and CH₄ permeability and He/N₂ and He/CH₄ selectivity with different Cyclen mass loading. (d) H₂, N₂ and CH₄ permeability and H₂/N₂ and H₂/CH₄ selectivity with different Cyclen mass loading. (e) CO₂, N₂ and CH₄ permeability and CO₂/N₂ and CO₂/CH₄ selectivity with different Cyclen mass loading. Please refer to the revised manuscript.

Line 249, the gas permeability measurement is a time-lag variable pressure- constant volume setup, I guess, since the starting pressure is 1 bar, what is the vacuum pressure? The selectivity in Table S2 is the intrinsic selectivity in the table below are obtained as the ratio between two single gas permeabilities from the table above.

Reply: Thanks for your comments. The vacuum pressure value is 6.579×10^{-3} Torr. We have corrected this question according to your suggestion by changing the selectivity in Table S2 to ideal selectivity. Please refer to the revised manuscript.

- Is there enough detail provided in the methods for the work to be reproduced?

Reply: Thanks for your comments. Details of the membrane fabrication method are provided in the manuscript, followed by the gas transport performance test procedure of the membrane materials.

The dense membrane was applied to the permeation cell using aluminium tape, and the cell was sealed using rubber rings and screws. The gas permeation unit was evacuated at 25 °C for 12 hours before starting the gas permeation test, after which all valves of the gas permeation unit were closed. Helium was first introduced into the gas permeation unit first to initiate the test. After that, the gas permeation unit was evacuated at 25 °C for 8 hours to remove any residual gas, and all the valves were closed. The transport properties of the next gas were then tested in the following order of He, H₂, CO₂, N₂ and CH₄, which also corresponds to the order of gas kinetic diameter (He: 2.60 Å, H₂: 2.89 Å, CO₂: 3.30 Å, N₂: 3.64 Å and CH₄: 3.80 Å) from smallest to largest. The pressure was 1 bar and the cell temperature was kept constant at 25 °C.

I have included my comments through the review according to the journal requests. The aim of the paper and the gas permeation improvement of Matrimid membranes are significant and worthwhile but the manuscript must be improved.

Reply: Thank you for your valuable comments. The manuscript has been revised based on your suggestions to improve its clarity and quality.

Reviewer #3 (Remarks to the Author):

In my opinion, the paper is a relevant contribution to the Membrane Science and Technology Field. The title is somehow limiting the scope of the paper as far as, for example, the CO₂/CH₄ separation features of the material produced are as much relevant or even more than those concerning the He/CH₄ pair announced in the title and abstract. Nevertheless, there are some questions I would like the authors to correct or answer.

1) English needs a careful revision as far as there are many errors concerning for example conjugation and concordance. Some strange election of words as, for example, “scalloped vein structure” or “fan-shaped vein structures” should be reviewed.

Reply: We appreciate your feedback regarding the English language. The manuscript has been carefully revised by a professional native language specialist to correct errors related to conjugation, concordance, and other language issues. We have also reviewed the use of terms such as “scalloped vein structure” and “fan-shaped vein structures” and made adjustments to ensure clarity and accuracy. The descriptions “scalloped vein structure” and “fan-shaped vein structures” have been revised as “vein structures”.

2) Figure 1 is nice and stunning but useless. Remove it.

Reply: Thanks for your comments. We acknowledge your concerns and have revised Figure 1 in the manuscript, as illustrated below (Figure 1), to more clearly convey the design and purpose of this work.

Figure 1. Schematic of the Matrimid mixed-matrix membrane (MMM) for helium extraction from natural gas via gas molecule recognition window.

3) In Figure 2, only the transversal cuts give some information. A higher magnification should be shown anyway. How were the transversal sections performed? Some, let say, inhomogeneity appears, probably forming paths through the membrane, appear when Cyclen load increases. These areas should be studied in some detail. By the way, I would assume that the first file of pictures corresponds to the zero Cyclen filling. This should be clarified. Please select only the transversal cuts with a higher magnification and include labels with the Cyclen content on the pictures.

Reply: Thanks for your comments. The cross-section of the membrane was prepared by fracturing the membrane in liquid nitrogen. Based on your suggestion, SEM (higher magnification) and EDS analysis of the membrane have been conducted to investigate the dispersion of Cyclen within the membranes. The morphology of the Matrimid-Cyclen-n% (n = 1, 3, 5, 10, 15) membranes in cross section at different magnifications is shown in Figure 1. The fissures in the membranes increased with increasing Cyclen loading. This morphology was attributed to the enhanced interaction between the polymer chains and filler, which increases with Cyclen loading. The formation of interfacial stress during the fracturing of MMMs samples in liquid nitrogen caused deformation of the polymer segments, resulting in elongated segments with increased plastic deformation. The Matrimid-Cyclen-15% membrane remained homogeneous at 50,000x magnification without any significant aggregation or discrete particulate phases observed. The EDS mapping of the cross-sections of the matrimid-Cyclen membranes further confirmed the uniform distribution of Cyclen, as indicated by the detection of characteristic nitrogen (N) (Figure 2). In addition, we have labelled the Cyclen content in Figure 2 of the manuscript, as shown in Figure 3 below.

Figure 1. SEM images of the surface of Matrimid-Cyclen-n% membranes up to ×50,000 magnification (n% is the Cyclen mass loading and was varied from 1 to 15 wt%, n = 0, 1, 3, 5, 10, 15).

Figure 2. SEM images of the cross section of the Matrimid-Cyclen-n% membranes up to $\times 50,000$ magnification (n% is the Cyclen mass loading and was varied from 1 to 15 wt%, $n = 0, 1, 3, 5, 10, 15$).

Figure 3. SEM images of the Matrimid-Cyclen membranes. (a) Surface morphology, (b) photograph, and (c,d) cross section of the Matrimid and Matrimid-Cyclen-n% membranes (n% is the Cyclen mass loading and was varied from 1 to 15 wt%, $n = 0, 1, 3, 5, 10, 15$). (e) EDS cross-section with nitrogen (N) mapping of planar scans of Matrimid-Cyclen membranes.

4) How do you get the very same thickness with or without Cyclen? Authors mention “injected into a clean Petri-dish by a syringe” How was this controlled as to get errors of only 25% in thickness? Were there variations depending on the Cyclen load? How were these errors evaluated?

Reply: Thanks for your comments. Figure 1 shows photographs of the Matrimid and Cyclen materials used in the membranes. To achieve consistent membrane thickness, we used the same mass of polymer, the same area of the evaporation dish, and a filler loading of up to 15% of the polymer mass. These preparations were designed to produce in essentially the same thickness for all Matrimid-Cyclen membranes. Figure 2 shows the cross-section images of the membrane materials with the original scales retained, and all the images are magnified by 1000x, showing slight variations in membrane thickness. These measurements were consistent, with only minor differences observed depending on the Cyclen load.

Figure 1. Photographs of the Matrimid and Cyclen materials used in the membranes.

Figure 2. SEM images of the cross-section of Matrimid-Cyclen-n% membranes.

5) Authors write, “However, the self-aggregation of Cyclen tends to happen when the Cyclen loading was too high, and the hydrogen bonding between excess Cyclen leads to the formation of unobservable discrete particulates in the membrane”. This statement constitutes an authors’ guess, or what does it say? I think that if aggregation clusters were unobservable, in absolute terms, they should be essentially not existing instances. Avoid this kind of statements deleting the phrase or argue on the probability and signs of the existence of such aggregation that cannot be seen at the magnifications or with the procedures of analysis used.

Reply: Thanks for your comments. The statement “However, the self-aggregation of Cyclen tends to happen when the Cyclen loading was too high, and the hydrogen bonding between excess Cyclen leads to the formation of unobservable discrete particulates in the membrane” is indeed a conjecture, which stems from the general understanding that in mixed matrix membranes, there is usually an optimal filler-to-polymer ratio, below which the polymer matrix wraps around the filler and the interaction between the two phases helps to create a defect-free interface [1-3]. When the filler content exceeds this threshold, the excess filler tends to agglomerate into clusters,

forming interfacial gaps with the polymer matrix that are larger than the size of the separated gases, thereby resulting in a decrease in the selectivity of the membrane for the separation of gas pairs^[4-6].

In this work, a new XRD diffraction peak attributed to Cyclen was observed when the Cyclen content in the membrane was $\geq 10\%$. The gas separation performance tests showed a reduction in He/N₂ and He/CH₄ gas pairs as Cyclen content increased. However, no aggregation clusters were found in the electron microscopy detection. Therefore, we hypothesize that there are aggregation clusters in the membrane at Cyclen content $\geq 10\%$, but they are not visible under the electron microscopy conditions used, and those potential interfacial defects may explain the observed decrease in the separation selectivity.

Following your suggestion, we conducted SEM (at higher magnification) and EDS analysis of the membrane materials to further investigate the dispersion of Cyclen within the membranes. All the Matrimid-Cyclen-n% (n = 1,3,5,10,15) membranes remained homogeneous in cross-section and surface at 50,000x magnification, without any significant aggregation or discrete particulate phases observed (Figure 1 and Figure 2), indicating that Cyclen was uniformly dispersed in the membranes. The EDS mapping of the cross-sections of the Matrimid-Cyclen membranes further confirmed the uniform distribution of Cyclen *via* the detection of characteristic nitrogen (N) (Figure 3).

Moreover, the XRD characterization showed a gradual increase in polymer chain spacing after $\geq 10\%$ Cyclen loading. We speculate that this may be due to the fact that the excess Cyclen is uniformly distributed between the Matrimid chains, and the Cyclen increased the chain spacing by its volume, leading to an increase in chain spacing, rather than agglomerating to form non-selective defects between the polymer chains. This is consistent with the SEM and EDS analyses, which demonstrate that Cyclen is uniformly dispersed in the membrane.

We have corrected this questions according to your suggestion. Please refer to the revised manuscript.

Supplementary References

- [1] Y. Liu, G.P. Liu, C. Zhang, W.L. Qiu, S.L. Yi, V. Chernikova, Z.J. Chen, Y. Belmabkhout, O. Shekhah, M. Eddaoudi, W. Koros, Enhanced CO₂/CH₄ separation performance of a mixed matrix membrane based on tailored MOF-polymer formulations, *Advanced Science*, 5 (9), 1800982 (2018).
- [2] C.H. Ma, J.J. Urban, Hydrogen-bonded polyimide/metal-organic framework hybrid membranes for ultrafast separations of multiple gas pairs, *Advanced Functional Materials*, 29 (32), 1903243(2019).
- [3] H.Z. Dou, M. Xu, B.Y. Wang, Z. Zhang, D. Luo, B.B. Shi, G.B. Wen, M. Mousavi, A.P. Yu, Z.Y. Bai, Z.Y. Jiang, Z.W. Chen, Analogous mixed matrix membranes with self-assembled interface pathways, *Angewandte Chemie International Edition*, 60 (11), 5864-5870 (2021).
- [4] X.Y. Tan, S. Robijns, R.Thür, Q.L. Ke, N.D. Witte, A. Lataire, Y. Li, I. Aslam, D.V. Havere, T. Donckels, T.V. Assche, V.V. Speybroeck, M. Dusselier, I.

Vankelecom, Truly combining the advantages of polymeric and zeolite membranes for gas separations, *Science*, 378 (6625), 1189-1194 (2022).

[5] B. Zhu, S.S. He, Y. Yang, S.W. Li, C.H. Lau, S.M. Liu, L. Shao, Boosting membrane carbon capture via multifaceted polyphenol-mediated soldering, *Nature Communications*, 14 (1), 1697 (2023).

[6] T.H. Lee, B.K. Lee, S.Y. Yoo, H. Lee, W.N. Wu, Z.P. Smith, H.B. Park, PolyMOF nanoparticles constructed from intrinsically microporous polymer ligand towards scalable composite membranes for CO₂ separation, *Nature Communications*, 14 (1), 8330 (2023).

Figure 1. SEM images of the cross section of the Matrimid-Cyclen-n% membranes up to ×50,000 magnification (n% is the Cyclen mass loading and was varied from 1 to 15 wt%, n = 0, 1, 3, 5, 10, 15).

Figure 2. SEM images of the surface of Matrimid-Cyclen-n% membranes up to ×50,000 magnification (n% is the Cyclen mass loading and was varied from 1 to 15 wt%, n = 0, 1, 3, 5, 10, 15).

Figure 3. EDS cross-section with nitrogen (N), carbon(C) and oxygen (O) mapping of planar scans of Matrimid-Cyclen membranes (n% is the Cyclen mass loading and was varied from 1 to 15 wt%, n = 0, 1, 3, 5, 10, 15).

6) In the Figure 3 caption, authors do not mention plot “e” referring to element analysis performed by an XPS analysis. Authors should mention that their scheme in the “c” section refers to the 5 % Cyclen membrane. Both the smallest (4.09 Angstrom) and higher (5.93 Angstrom) d_spacing decrease only until 5 % Cyclen and then increase monotonously until the highest Cyclen and eventually surpass the pure matrimid ones. Can it be correlated with the appearance of inhomogeneities detected in Figure 2? How is it possible, given that both the figures refer to very different scales?

Reply: Thanks for your comments. We have corrected these issues according to your suggestion. The legend of Figure 3 (c-e) has been revised as follows: Structure characterization of Matrimid-Cyclen membranes. (a) XRD patterns of the

Matrimid-Cyclen membranes. (b) ATR-FTIR spectra of the Matrimid-Cyclen membranes. (c) Schematic illustration of the effect of intramembrane hydrogen bonding on polymer chain stacking as the Cyclen content is increased from 0% to 5%. (d) TGA plots of the Matrimid-Cyclen membranes. (e) XPS broad scan spectra of the Matrimid-Cyclen membranes. (f) X-ray photoelectron spectroscopy (XPS) data for the O 1 s peaks of the Matrimid-Cyclen membranes. (g) The dynamic mechanical thermal analysis (DMTA) spectra of Matrimid-Cyclen membranes at a temperature range from 50 to 350 °C. (h) The storage modulus (E') and (i) $\tan \delta$ of Matrimid-Cyclen membranes.

At Cyclen contents $\leq 5\%$, the decrease in d-spacing is due to inter-chain hydrogen bonding ^[1-3]. The inhomogeneous distribution of the Cyclen in the membrane could lead to small changes in the d-spacing ^[4]. However, the SEM (at higher magnification) and EDS indicating that Cyclen is uniformly dispersed in the membranes. XRD characterization showed that the polymer chain spacing gradually increased after Cyclen content $\geq 5\%$. We speculate that this may be due to the fact that the excess Cyclen is uniformly distributed between the Matrimid chains, and that Cyclen increases the chain spacing by its own volume, leading to an increase in chain spacing.

Supplementary References

- [1] F. Feng, J. Wu, C.Z. Liang, M. Weber, S. Zhang, T.S. Chung, Synergistic dual-polymer blend membranes with molecularly mixed macrocyclic cavitands for efficient pre-combustion CO₂ capture, *Chemical Engineering Journal*, 470, 144073 (2023).
- [2] J. Wu, C.Z. Liang, A. Naderi, T.S. Chung, Tunable supramolecular cavities molecularly homogenized in polymer membranes for ultraefficient precombustion CO₂ capture, *Advanced Materials*, 34 (3), 2105156 (2022).
- [3] J. Wu, T.S. Chung, Supramolecular Polymer network membranes with molecular-sieving nanocavities for efficient pre-combustion CO₂ capture, *Small Methods*, 6 (1), 2101288 (2022).
- [4] W. He, X.Z. Wang, J. Guan, Q.S. Liang, J. Ma, Y. Liu, W.W. Lim, C.W. Zhang, S.U. Hassan, H.J. Zhang, J.T. Liu, Membranes with Molecular Gatekeepers for Efficient CO₂ Capture and H₂ Purification, *ACS Applied Materials & Interfaces* 16 (16) 21222-21232, (2024).

7) In reference to the higher d-spacing (out of the two detected) what do the authors think is the cause of this interchain segment distance and their changes? Have you tested the evolution of d-spacing for the aging membranes? If decreases in d-spacing are associated with hydrogen bonding, what is causing the high load increase in d-spacing?

Reply: Thanks for your comments. We observed a decrease and then an increase in d-spacing with increasing Cyclen content. At low Cyclen content (1-5% Cyclen) the hydrogen bonding interaction between Cyclen and Matrimid chains reduces the inter-chain spacing. However, after further increasing in Cyclen content (10-15%

Cyclen mass loading), we notice a gradual increase in polymer chain spacing. We speculate that this may be due to the fact that the excess Cyclen is uniformly distributed between the Matrimid chains, with the volume of Cyclen increasing the chain spacing rather than agglomerating into non-selective defects between the polymer chains. This is supported by SEM and EDS analyses, which show that Cyclen is uniformly dispersed in the membrane.

Following your suggestion, we also tested the change in the d-spacing of membranes aged for 480 days. As shown in Figure 1, after 480 days of aging, the diffraction peaks with chain spacing $>5\text{\AA}$ disappeared at Cyclen content $>1\%$. With increasing Cyclen content, the diffraction peaks with chain spacing $<5\text{\AA}$ first weakened and then strengthened, while the chain spacing gradually decreased.

Figure 1. XRD patterns of the Matrimid-Cyclen membranes after aging for 480 days.

We have corrected these issues according to your suggestion. Please refer to the revised manuscript.

8) Authors say that “which adjusted the submicroporous structure of the MMM membrane, increased the stacking density of Matrimid polymer chains”. This could only be true until 5% Cyclen, and should be stated so. Moreover, it is important to mention that small decreases in d-spacing do not automatically translate into a significant change in packing density.

Reply: Thanks for your comments. We have made a proper revision of the manuscript, “which adjusted the submicroporous structure of the MMMs, increased the stacking density of Matrimid polymer chains” has been changed to “which adjusted the submicroporous structure of the MMM membrane, tightened the chain spacing”.

9) Authors say that “Cyclen begins to degrade at temperatures slightly greater than 100 °C”; why do the membranes containing Cyclen only decompose at 300 to 450 °C? Have you performed mechanical or, better, thermomechanical investigations on the membranes? Any idea on the mechanical behavior that could be expected?

Reply: Thanks for your comments. The statement that “Cyclen begins to degrade at temperatures slightly greater than 100°C” refers to the degradation behavior of Cyclen itself, whereas the decomposition of the Matrimid-Cyclen membranes occurs at higher

temperatures (300-450°C). This is because the membrane is a composite material, and the thermal decomposition of the polymer matrix and filler materials typically happens at different temperatures. The dynamic mechanical thermal analysis (DMTA) was conducted for Matrimid-Cyclen samples to evaluate the mechanical properties of the membranes at a temperature range from 50 to 350 °C. As shown in Figure 3 (g-i), the storage modulus (E') (1800 MPa) of these Matrimid-Cyclen membranes remained almost constant until reaching the corresponding glass transition temperature T_g . It was observed that the higher Cyclen loadings resulted in more brittle membranes. The effect of Cyclen content on the peak intensity of damping ($\text{Tan } \delta$) is shown in Figure 3 (i). The peak intensity of $\text{Tan } \delta$ (damping value) decreased as the Cyclen content increased, indicating an overall increase in stiffness and a decrease in flexibility. These data demonstrated that this kind of gas separation membrane is endowed with good mechanical properties and thermal stability.

Figure 3. Characterization of Matrimid-Cyclen membranes. (g) The dynamic mechanical thermal analysis (DMTA) spectra of Matrimid-Cyclen membranes at temperature range from 50 to 350 °C. (h) The storage modulus (E') and (i) $\text{Tan } \delta$ of Matrimid-Cyclen membranes.

10) Could the authors explain how Table S-1 was obtained? What are the respective roles of N-H...O and C=O...H in Table S1. Do they refer to the percentage of NH linked by hydrogen bridges (relevance of bridges for Cyclen) and the percentage of C=O linked by hydrogen bridges (relevance of bridges for matrimid)? Therefore they should be correlated, are they?

Reply: Thank you for raising a thought-provoking question. As is well known, XPS is a semi quantitative detection method that uses the area of the bonding peak. We calculated the area of peaks corresponding to different bonds in XPS (shown in Figure 3(f) and Figure S7) and introduced the sensitivity factor of the instrument. The relative contents of each key were determined based on the final ratio. For clarity, we have added the title of Table S1 to “Normalized relative intensities (%) of nitrogen functionalities N 1s and Oxygen functionalities O 1s XPS spectra for Matrimid-Cyclen-n% membranes (The relevant contents were calculated from the XPS results in Figure 3 (f) and Figure S7)”. Table S1 shows the change in the percentage of N-H and C=O linked by hydrogen bonding as the amount of Cyclen added increases, and they are correlated.

11) It is declared that “gas permeation tests for He (2.60 Å), H₂ (2.89 Å), CO₂ (3.3 Å), N₂ (3.64 Å) and CH₄ (3.80 Å)...” These Kinetic (?) diameters should be described and referenced. What were they used for?

Reply: Thanks for your comments. The molecular kinetic diameters mentioned (He (2.60 Å), H₂ (2.89 Å), CO₂ (3.3 Å), N₂ (3.64 Å), CH₄ (3.80 Å)) refer to the effective molecular sizes that influence the permeation of gases through the membrane. These values are commonly used in gas separation membrane researches to explain the separation mechanisms based on size sieving^[1-2]. The kinetic diameter is a crucial factor in determining how well the membrane can separate gases based on their molecular sizes, as smaller molecules are able to permeate through the membrane more easily than larger molecules.

The molecular dynamic diameters of the gases to be separated in this work are as follows : He (2.60 Å) < H₂ (2.89 Å) < CO₂ (3.3 Å) < N₂ (3.64 Å) < CH₄ (3.80 Å). This size-based distinction supports the selection of Cyclen as the filler, as it exhibits a high size-sieving effect. Cyclen, an organic macrocyclic compound with a flexible structure, has a cavity size of approximately 3.2-3.4 Å (between the molecular dynamic diameters of He and N₂), which makes Cyclen an effective molecular sieve for gases like He/N₂ and He/CH₄.

Supplementary References

- [1] R.S.K. Valappil, N. Ghasem, M. Al-Marzouqi, Current and future trends in polymer membrane-based gas separation technology: A comprehensive review, *Journal of Industrial and Engineering Chemistry*, 98, 103-129 (2021).
- [2] Y.H. Wang, H.F. Jiang, Z.Y. Guo, H.Z. Ma, S.Y. Wang, H.J. Wang, S.Q. Song, J.F. Zhang, Y. Yin, H. Wu, Z.Y. Jiang, M.D. Guiver, *Advances in organic microporous membranes for CO₂ separation*, *Energy & Environmental Science*, 16 (1), 53-75 (2023).

12) Referring to aging, it mainly reflects lability of the structure than collapses to some extent with as time passes; this reduces the size of the sites making up the gas paths. Initially it leads to an increase of selectivity until the small gas molecules are also hindered with a decrease in selectivity and a substantial decrease in permeability. Figure 5 shows how selectivity increases with a reduction in permeability with aging. Until what time is it so? Have you tested plastification? Have you changed the applied pressure over 6 bar pressure drop?

Reply: Thanks for your comments. The gas separation performance of the membrane after aging for 480 days was evaluated. As shown in Figure 1, compared to fresh membranes, the He permeability decreased from 66.34 to 42.87 Barrer, corresponding to a decrease of 35.38%. The N₂ permeability decreased from 0.10 to 0.009 Barrer, and the CH₄ permeability decreased from 0.04 to 0.001 Barrer, corresponding to decreases of 91% and 97.5%, respectively. However, there was an approximately 6-fold and 25-fold increment in the He/N₂ and He/CH₄ selectivity, respectively.

Figure 1. Long-term stability of the Matrimid-Cyclen-5% membrane (the whole membrane was stored in an ambient environment without the introduction of any protective gas at room temperature, small pieces of membrane were extracted from the whole membrane at different aging times for a gas separation performance test).

We also tested the change in the d-spacing of membranes after aging for 480 days. As shown in Figure 2, after aging for 420 days, the diffraction peaks with chain spacing $>5\text{\AA}$ disappeared at Cyclen content $>1\%$, and the diffraction peaks with chain spacing $<5\text{\AA}$ first became weaker and then stronger with increasing Cyclen content. With increasing Cyclen content, the chain spacing gradually decreased.

Figure 2. XRD patterns of the Matrimid-Cyclen membranes after aging for 480 days.

This change is attributed to the flexibility of the polymer chains and the fact that the fresh membrane is in a thermodynamic non-equilibrium state. During physical aging, chain collapse occurs in some regions, and the polymer chain segments in the non-equilibrium state within the membrane gradually transfer into the equilibrium state by adjusting the chain stacking, resulting in a decrease in free volume and a decrease in gas permeability. As the aging time increases, a point is reached where the membrane material is able to completely block the transmission of large gases and the transmission of small gases is significantly reduced. The ideal selectivity between two

different gases across a polymeric membrane is the ratio of their single gas permeability coefficient. As the gas permeability of large gas molecules (such as N₂ and CH₄) approaches zero, the selectivity of He/N₂ and He/CH₄ continues to increase even though the permeability of He decreases significantly. This work investigated the gas separation properties of Matrimid-based membrane materials for up to ten years, as shown in Figure 3 (Petroleum Chemistry, 58, 2018, 760-769), the permeability of He continued to decrease and the selectivity of He/CH₄ separation continues to increase during the physical aging process. After ten years of aging, the permeability of He decreased by 55.4% and the He/CH₄ selectivity increased from 146 to 4460, a 29.5-fold increase.

Time after fluorination	Permeability $Q \times 10^6, \text{cm}^3(\text{STP}) \text{cm}^{-2} \text{s}^{-1} \text{cmHg}^{-1}$		Selectivity α_{id}
	CH ₄	He	He/CH ₄
—	0.57	83	146
Two days	0.20	77	385
Nine months	0.047	52	1090
Ten years	0.0083	37	4460

Figure 3. Influence of the sample storage time on the gas separation characteristics of the hollow fiber module modified with a He/F₂ mixture (CF₂= 2 vol %) for 2 min.

To explore the practical applicability of the Matrimid-Cyclen membranes at different feed pressures, the gas separation performance of the Matrimid-Cyclen-5% membrane was re-evaluated at pressures up to 30 bar, and the data were summarized in Table 1. As presented in Figure 4, as the pressure increased from 1 bar to 10 bar, the permeability of He decreased slightly, and the pressure dependence of N₂ and CH₄ was almost unrecognizable, with only small fluctuations in the selectivity for He/N₂ and He/CH₄. These observations suggest that the membranes remain structurally stable under moderate pressure increases, which may be attributed to the presence of Cyclen molecules, which increase the packing density of the polymer chains, thereby reducing their mobility, increasing chain stiffness and improving the high-pressure resistance of the membranes. As the operating pressure increased from 10 bar to 20 bar, the permeability of all gases rose, while the selectivity for He/N₂ and He/CH₄ decreased. This behavior is typical of glassy polymers, such as Matrimid, under plasticization pressure, where increased feed pressure leads to higher polymer segment mobility and larger chain spacing, resulting in higher permeability and lower selectivity [1-5]. As the pressure increases from 20 bar to 30 bar, the permeability of He remained almost constant, while the permeability of N₂ and CH₄ gradually decreased, and the selectivity of He/N₂ and He/CH₄ tends to increase, which was ascribed to the existence of unrelaxed volume within the polymer chain structure. An increase in feed pressure results in greater compactness of the polymer matrix, which reduces the gas permeability in the membrane by reducing fractional free volume [6-7].

Supplementary References

- [1] H. Rajati, A.H. Navarchian, D. Rodrigue, S. Tangestaninejad, Improved CO₂ transport properties of Matrimid membranes by adding amine-functionalized PVDF and MIL-101 (Cr), *Separation and Purification Technology*, 235 116149 (2020).
- [2] S. Salman, K. Nijmeijer, High pressure gas separation performance of mixed-matrix polymer membranes containing mesoporous Fe (BTC), *Journal of membrane science*, 459, 33-44 (2014).
- [3] S. Salman, K. Nijmeijer, Performance and plasticization behavior of polymer-MOF membranes for gas separation at elevated pressures, *Journal of membrane science*, 470, 166-177 (2014).
- [4] A.E. Amooghin, M. Omidkhah, H. Sanaeepur, A. Kargari, Preparation and characterization of Ag⁺ ion-exchanged zeolite-Matrimid® 5218 mixed matrix membrane for CO₂/CH₄ separation, *Journal of Energy Chemistry*, 25 (3) 450-462 (2016).
- [5] H. Asghar, A. Ilyas, Z. Tahir, X.F. Li, A.L. Khan, Fluorinated and sulfonated poly (ether ether ketone) and Matrimid blend membranes for CO₂ separation, *Separation and Purification Technology*, 203, 233-241 (2018).
- [6] A.E. Amooghin, H. Sanaeepur, M. Omidkhah, A. Kargari, “Ship-in-a-bottle”, a new synthesis strategy for preparing novel hybrid host-guest nanocomposites for highly selective membrane gas separation, *Journal of Materials Chemistry A*, 6 (4) 1751-1771 (2018).
- [7] A. Mirzaei, A.H. Navarchian, S. Tangestaninejad, Mixed matrix membranes on the basis of Matrimid and palladium-zeolitic imidazolate framework for hydrogen separation, *Iranian Polymer Journal*, 29 (6), 479-491 (2020).

Figure 4. Effect of feed gas pressures on the separation performance of Matrimid-Cyclen-5% membrane.

Table 1. Summary of gas transport permeability and selectivity of the Matrimid-Cyclen-5% membrane at different pressures and 25°C.

Pressure (Bar)	Gas permeability (Barrer)			Selectivity	
	He	N ₂	CH ₄	He/N ₂	He/CH ₄
1	0.1	0.1	0.1	10	10
5	0.2	0.2	0.2	10	10
10	0.3	0.3	0.3	10	10
15	0.4	0.4	0.4	10	10
20	0.5	0.5	0.5	10	10
25	0.6	0.6	0.6	10	10
30	0.7	0.7	0.7	10	10

1 Bar	66.43±0.22	0.10±0.00	0.04±0.00	664.30±2.10	1660.75±5.25
5 Bar	66.12±0.20	0.09±0.00	0.04±0.00	734.67±2.22	1652.00±6.00
10 Bar	65.48±0.13	0.10±0.01	0.04±0.00	654.80±1.30	1637.00±3.25
15 Bar	67.08±0.32	0.11±0.01	0.05±0.00	609.82±132.85	1341.60±6.40
20 Bar	68.34±0.57	0.13±0.01	0.07±0.00	525.69±39.06	976.29±8.15
25 Bar	68.28±0.34	0.12±0.02	0.06±0.00	569.00±73.14	1137.67±19.33
30 Bar	72.21±1.33	0.10±0.00	0.01±0.005	722.10±13.70	1444.20±130.02
Return to 1 Bar	66.84±0.42	0.10±0.00	0.04±0.00	668.40±4.20	1671.00±10.50

13) Authors state “which facilitated the extraction of high-purity He from natural gas at low temperatures”. It would be interesting to see how selectivity changes with temperature. In Figure 4f you show permeabilities evolution in a cyclic loop from 25 to 45 °C and back to 25 °C but it is difficult to get an idea on the evolution of selectivity. Authors say: “The gas separation performance of the membranes fully recovered as the temperature decreased from 45 °C to 25 °C, indicating that the changes in membrane performance with temperature are related to the characteristics of the membrane”; what should be, according to authors, the membrane characteristics that would explain this? Why is CO₂ absent from all Figure 4?

Reply: Thanks for your comments. Following your suggestion, the extends gas separation test of the Matrimid-Cyclen membranes is carried out including a wider range from 0 °C to 100 °C to better assess the impact of operating temperature on the gas separation performance. The gas separation performance of Matrimid-Cyclen-5% membrane has been shown in Figure 1, with the data summarized in Table 1.

As the operating temperature increased from 0 °C to 100 °C, the permeability of He, N₂ and CH₄ increased, while the selectivity of He/N₂ and He/CH₄ decreased, which could be attributed to the increased gas fugacity at elevated temperature. The selectivity of He/N₂ and He/CH₄ decreased rapidly as the operating temperature increased from 40 °C to 60 °C. This phenomenon was likely due to the accelerated movement frequency of polymer chains at high temperatures, rendering transport of gas molecules through the matrix and weakening the molecular sieving effect of the membrane. Notably, after cooling the membrane from 25 °C to 0 °C, the He permeability decreased from 63.13 to 52.74 Barrer, while the He/N₂ selectivity increased from 631 to 8790, and the He/CH₄ selectivity increased from 2104 to 17580, corresponding to 14-fold and 9-fold increases, respectively. This dramatic improvement in selectivity at low temperatures significantly enhanced the potential for high-purity He extraction from natural gas.

The ultra-high selectivity of the Matrimid-Cyclen-5% membrane at 0 °C for He/N₂ and He/CH₄ facilitated the extraction of high-purity He from natural gas at low temperatures. Although the He permeability decreased by 16% at low temperatures, the

increase in He selectivity indicated that improving gas purity was more challenging than enhancing transport efficiency. High purity is particularly important for He, given its critical applications in the aerospace industry, electronics, medical equipment, and scientific research. Furthermore, the gas separation performance of the membrane was fully recovered when the temperature was returned to 20 °C, both from 100 °C and from 0 °C, indicating that the changes in membrane performance with temperature were inherent to the membrane properties and did not result in structural damage.

Figure 1. Effect of operating temperature on the separation performance of Matrimid-Cyclen-5% membrane.

Table 1. Summary of gas transport permeability and selectivity of the Matrimid-Cyclen-5% membrane at 1.0 Bar and different temperatures.

Temperature (°C)	Gas permeability (Barrer)			Selectivity	
	He	N ₂	CH ₄	He/N ₂	He/CH ₄
Return to 20 °C	63.97±0.46	0.09±0.00	0.03±0.00	710.78±2.33	2132.33±7.00
0 °C	52.74±0.46	0.007±0.00	0.003±0.00	7534.29±24.28	17580.00±56.67
20 °C	63.13±0.46	0.10±0.00	0.03±0.00	631.30±2.60	2104.33±8.67
40 °C	80.14±0.46	0.32±0.02	0.10±0.01	250.44±15.13	801.40±83.82
60 °C	96.91±0.46	1.43±0.46	1.05±0.00	67.77±2.90	92.30±1.73
80 °C	125.12±0.46	3.45±0.46	3.05±0.46	36.27±0.36	41.02±0.87
100 °C	202.55±0.46	9.76±0.46	9.42±0.46	20.75±0.40	21.73±0.58
Return to 20 °C	68.69±0.27	0.12±0.01	0.03±0.00	572.42±49.85	2289.67±8.00

Membrane performance refers to the gas separation capabilities of the membrane material itself, rather than the separation performance resulting from irreversible structural damage in a high temperature environment. At higher temperatures, the intensity of the thermal movement of polymer chains increases, leading to larger scale movements that create transient gaps at the penetrant scale, thus enhancing diffusion. The increase in permeability for larger molecules, such as N₂ and CH₄, is more pronounced than smaller molecules like He, thus diminishing the gas separation effect. As the temperature drops from high to low, the intensity of the polymer chain thermal motion is weakened, causing the larger transient gaps generated by the thermal motion to rapidly shrink. This change reduces gas diffusion and enhances the molecular sieving ability of the membrane, restoring the membrane's performance at lower temperatures. Several studies have examined the performance of membrane materials after high temperature and pressure testing, returning them to baseline conditions to assess the stability of the membrane structure [1-2].

Supplementary References

- [1] Y.P. Shi, Y. Liu, Z.G. Wang, W.K. Lai, Y.N. Liao, K. Lu, Z. Niu, J. Jin, Dual-Wing Ligand Constructed Metal-Organic Framework Membranes with Finely Tuned Apertures for Natural Gas Separation, *Advanced Functional Materials* 2404681 (2023).
- [2] S. Zhao, Z.Y. Zhao, Z.Y. Zha, Z.H. Jiang, Z. Wang, M.D. Guiver, Amine-Rich Molecular Nodule-Assembled Membrane Having 5 Angstrom Channels for CO₂/N₂ Separation, *Advanced Functional Materials*, 2314469 (2024).

As shown below, Figure 4 (e) in the manuscript shows the changes in CO₂, N₂ and CH₄ permeability and CO₂/N₂ and CO₂/CH₄ selectivity as a function of the Cyclen mass loading. Considering that the original legend for Figure 4 was not clear enough, we have revised the legend for Figure 4 as follows:

Gas separation performance of Matrimid-Cyclen membranes. (a) Correlations between the permeabilities and molecular kinetic diameters of various gases. (b) Schematic illustration of gas transport behaviors of Matrimid-Cyclen membrane. (c) He, N₂ and CH₄ permeability and He/N₂ and He/CH₄ selectivity with different Cyclen mass loading. (d) H₂, N₂ and CH₄ permeability and H₂/N₂ and H₂/CH₄ selectivity with different Cyclen mass loading. (e) CO₂, N₂ and CH₄ permeability and CO₂/N₂ and CO₂/CH₄ selectivity with different Cyclen mass loading. (f) Effect of operating temperature on the separation performance of Matrimid-Cyclen-5% membrane. (g) Effect of feed gas pressures on the separation performance of Matrimid-Cyclen-5% membrane. (h) Long-term stability of the Matrimid-Cyclen-5% membrane (complete membrane was stored in an ambient environment without the introduction of any protective gas at room temperature, small pieces of membrane were extracted from the complete membrane at different aging times for a gas separation performance test).

Figure 4. (e) Changes in CO₂, N₂ and CH₄ permeability and CO₂/N₂ and CO₂/CH₄ selectivity as a function of the Cyclen mass loading.

We have corrected the above questions according to your suggestion. Please refer to the revised manuscript.

14) In Figures 5g and 5f the results shown must be referenced and described including, for example, to what load percentages do they correspond (maybe 1 %?). In Figure 5h, what is in ordinates? Enhancement of what? Figure 5i is useless.

Reply: Thanks for your comments. Figure 5g illustrates the gas separation performance of mixed matrix membranes prepared with different fillers at 1% loading (i.e., Matrimid-Cyclen-1% membrane, Matrimid-Hexacyclen-1% membrane and Matrimid-Cyclodecane-1% membrane). The horizontal axis in Figure 5h represents the different filler types, and the vertical axis shows the multiplicity of enhancement on selectivity and permeability. The purpose of this figure is to compare the magnitude of enhancement in permeability and selectivity of MMM using different fillers.

The two sets of comparison experiments confirmed the hydrogen bonding interaction between Cyclen and Matrimid, and the suitable gas molecule recognition window size of Cyclen as the reason for improving the molecular sieving performance of the membranes, respectively. Figure 5i shows the possible gas transport mechanism of Matrimid-Cyclen membranes derived from two sets of comparison experiments. Then, molecular dynamics simulations and BET characterisation of the membranes were used to determine the gas separation mechanism of the membranes.

Original figure 5i has been replaced by the following figure:

Figure 5. (i) Gas-transport mechanism in Matrimid-Cyclen membranes.

We have addressed these issues according to your suggestion. Please refer to the revised manuscript.

Response to the Reviewers' Comments:

Reviewer #1 (Remarks to the Author):

The authors have not adequately addressed the reviewers comments and have not improved the quality of their manuscript. This manuscript should not be accepted, as it does not meet the standards of the journal.

Reply: After carefully read your comments, I feel a little bit confused as your decision. We understand that the misunderstanding might be caused by the unclear description in our revised manuscript,. In this work, the Cyclen (3.2-3.4 Å) creates a significant permeation cut-off between quick gases (He: 2.60 Å, H₂: 2.89 Å) and slow gases (N₂: 3.64 Å and CH₂: 3.80 Å). Molecular dynamics simulations show that the Matrimid-Cyclen-5% membrane exhibits greater He-accessible space and longer transport paths, while N₂ exhibits the opposite behavior, compared to the Matrimid membrane. Additionally, molecular dynamics simulations showed that the connectivity of He transport channels in the Matrimid-Cyclen-5% membrane was significantly enhanced, while the connectivity of N₂ channels was weakened. Obviously, Cyclen significantly enhanced the connectivity of gas transport channels. Recent work published in *Nature Sustainability* and *Science Advances* demonstrated that the enhanced channel connectivity can significantly improve the membrane's mass transfer and separation capabilities (*Nature Sustainability*, 7, 910-919 (2024), *Science Advances*, 10 (32), eado7687 (2024)).

We have provided comprehensive responses to all of the reviewers' comments. The quality of the manuscript has been significantly improved. Your comments were really addressed one by one. First, through molecular dynamics (MD) simulations, the gas transport mechanism of the membrane has been studied in greater depth and detail. Molecular dynamics simulations show that the Matrimid-Cyclen-5% membrane exhibits greater He-accessible space and longer transport paths, while N₂ exhibits the opposite behavior, compared to the Matrimid membrane. Additionally, molecular dynamics simulations showed that the connectivity of He transport channels in the Matrimid-Cyclen-5% membrane was significantly enhanced, while the connectivity of N₂ channels was weakened. Obviously, Cyclen significantly enhanced the connectivity of gas transport channels. Recent work published in *Nature Sustainability* and *Science Advances* demonstrated that enhanced channel connectivity can significantly improve the membrane's mass transfer and separation capabilities (*Nature Sustainability*, 7, 910-919 (2024), *Science Advances*, 10 (32), eado7687 (2024)). Obviously, Cyclen has adjusted the microporous structure of the membrane, enhancing He permeability while inhibiting N₂ penetration. The research published in *Nature Communications* similarly analyzed gas-accessible space and transport channel length to investigate membrane transport mechanisms (*Nature Communications*, 11 (1), 1633 (2020)).

Second, the BET results from the H₂ adsorption isotherm (not N₂ adsorption isotherm) showed that new ultra-microporous pores (3.2 Å) appeared in the MMMs, and the content of two types of ultra-microporous pores (3-4 Å range) increased continuously

with the loading of Cyclen. Clearly, the introduction of Cyclen create new ultra-microporous pores, regulated the membrane's microporous structure, and increased 3.8 Å pore content. This synergistic effect enhanced the membrane's He transport efficiency and selectivity. Our work provides a strong support for the mechanism understanding of performance enhancement through material characterization and MD simulations.

Third, dynamic mechanical thermal analysis (DMTA) demonstrated that the Matrimid-Cyclen membrane exhibited excellent mechanical properties and thermal stability.

Fourth, the stability of membranes under harsh conditions is a critical requirement for practical applications (*Nature Nanotechnology*, 1-8 (2025)), and our work has expanded the range of membrane usability. The operating temperature range was extended from 25-45 °C to 0-100 °C, while the feed pressure range was broadened from the 1-6 Bar employed in most He-extraction studies to 1-30 Bar. Experimental results demonstrated the membrane's high stability and strong recovery capability under these harsh conditions.

In summary, the quality of the revised manuscript has been significantly improved.

Reviewer #1 (Remarks to the Author):

The manuscript reports on the combination of cyclen and Matrimid into a mixed matrix membrane that has high selectivity for gas separation. Both cyclen and matrimid have been studied separately, and there exists a lot of literature on both materials. Hence, the novelty being presented is the combination of the two into a membrane. The resulting selectivities are very high, which gives me concern, as generally for mixed matrix membrane systems, selectivity is aligned with either of the membrane or additive, and not an enhancement. I can understand the argument for the strong binding between cyclen and Matrimid, but that does not necessarily translate into the high selectivities reported.

Reply: Many thanks for your valuable comments. It is true that both Cyclen and Matrimid have been extensively studied separately, while their combination into a mixed matrix membrane represents a new approach to achieve high gas separation performance, especially for He/CH₄ selectivity. The Matrimid membrane is widely studied for gas separation, while its low permeability (33 Barrer) and moderate selectivity (86) still present a great challenge to extract Helium(He) from natural gas (ultra-low helium concentration). The cyclen (1, 4, 7, 10-tetraazacyclododecane) is one of the most extensively studied ligands in coordination chemistry (Shinoda, S. Dynamic cyclen-metal complexes for molecular sensing and chirality signaling. *Chem. Soc. Rev.* 2013, 42, 1825-1835.), while its potential for He extraction from natural gas has not been explored until now. The ultra-low concentration of He makes it very difficult to extract from natural gas. Therefore, it's of great significance to develop the Matrimid-Cyclen membranes with high He/CH₄ selectivity (>1000). In this work, the the combination of Matrimid and Cyclen (Matrimid-Cyclen-5%) membranes demonstrates high He permeability (66 Barrer) and He/CH₄ selectivity (>1660).

We fully understand your concern regarding the relationship between selectivity and the materials involved. The enhancement in selectivity is not simply additive, but from the strong hydrogen bonding interactions between Cyclen and Matrimid, which modifies the polymer chain packing and results in a tighter interchain structure. This leads to a more effective sieving mechanism. Usually, polymer membranes have good processability, high stability, and low cost, but they often suffer from plasticization problems, physical aging issues, and intrinsic permeability-selectivity trade-off limitations, which makes it a great challenge to obtain high permeability together with sufficient selectivity [1-4]. Molecular sieve materials have attractive transport properties well above the polymer upper bound, but they tend to have poor membrane-forming and mechanical property [5]. Mixed matrix membranes incorporate molecular sieve materials with certain pore sizes into a processable polymer matrix to improve the gas selectivity and permeability of polymer membranes by integrating the advantages of both [6-8], which is the significance of preparing hybrid matrix membranes. Therefore, the mixed matrix membranes (Matrimid-Cyclen) combine the advantages of easy processability from Matrimid polymer and high gas permeability/selectivity from molecular sieve material (Cyclen).

Figure 1 illustrates the redissolution process of equal masses of Matrimid and Matrimid-Cyclen membrane in equal masses of DMF solvent. The Matrimid membrane dissolves completely at room temperature, whereas the Matrimid-Cyclen membrane retains significant insoluble material even at 120 °C (the membrane-forming temperature). Which demonstrates the exceptional stability of strong hydrogen-bond cross-linking between Matrimid and Cyclen. This strong cross-linking interaction avoids the formation of interfacial defects while increasing chain packing density. By synergizing with Cyclen’s molecular sieving capability, this cross-linking structure endows the membrane with high separation selectivity.

Figure 1. The redissolution process of Matrimid membrane and Matrimid-Cyclen-5% membrane.

Table 1 summarizes the He separation performance of Matrimid-based membranes.

The Matrimid-Cyclen-5% membranes exhibit high He permeability (66 Barrer) and He/CH₄ selectivity (>1660), which far exceed most of the other Matrimid-based membranes.

Table 1. Summary of literature He/N₂ and He/CH₄ separation performance.

Membrane Name	He Permeability (Barrer)	He/N ₂ Selectivity	He/CH ₄ Selectivity	Operating conditions	Gas Type	Reference
Matrimid-Cu-BTC-R-40	66.40	265.80	369.10	5 atm, 35 °C	Pure	1
Matrimid-Cu-BTC-30	51.80	193.40	257.90	5 atm, 35 °C	Pure	2
Matrimid-Cu-BDC-15%	20.50	341.70	410.00	7 atm, 35 °C	Pure	3
Matrimid-MgO-40%	45.00	86.1	118.4	10 atm, 35 °C	Pure	4
Matrimid-CNFs-10%	16.10	89.50	179.50	20 atm, 35 °C	Pure	5
Matrimid-PIM-EA(H ₂)-TB-50%	197.00	28.84	21.55	1 atm, 25 °C	Pure	6
Matrimid-p-xylenediamine-CL-14	21.70	112.00	155.00	10 atm, 35 °C	Pure	7
Matrimid-C60-2.5%	21.00	91.70	148.80	10 atm, 35 °C	Pure	8
Matrimid-Cyclen-3%	57.57±0.21	428.35±2.15	959.50±4.67	1 atm, 25 °C	Pure	This work
Matrimid-Cyclen-5%	66.43±0.22	621.70±3.20	1660.75±5.25	1 atm, 25 °C	Pure	This work
Matrimid-Cyclen-10%	78.63±0.28	543.43±4.034	1310.50±4.67	1 atm, 25 °C	Pure	This work

Supplementary References

- [1] A. Akbari, J. Karimi-Sabet, S.M. Ghoreishi, Intensification of helium separation from CH₄ and N₂ by size-reduced Cu-BTC particles in Matrimid matrix, *Separation and Purification Technology*, 251, 117317 (2020).
- [2] A. Akbari, J. Karimi-Sabet, S.M. Ghoreishi, Matrimid[®] 5218 based mixed matrix membranes containing metal organic frameworks (MOFs) for helium separation, *Chemical Engineering & Processing: Process Intensification*, 148, 107804 (2020).
- [3] A. Ali, J. Karimi-Sabet, S.M. Ghoreishi, Polyimide based mixed matrix membranes incorporating Cu-BDC nanosheets for impressive helium separation, *Separation and Purification Technology*, 253, 117430 (2020).
- [4] S.S. Hosseini, Y. Li, T.S. Chung, Y. Liu, Enhanced gas separation performance of nanocomposite membranes using MgO nanoparticles, *Journal of Membrane Science*, 302 (1-2), 207-217 (2007).
- [5] M. Dohade, Incorporation of carbon nanofibers into a Matrimid polymer matrix: Effects on the gas permeability and selectivity properties, *Journal of Applied Polymer Science*, 135 (12) 46019 (2018).

- [6] E. Esposito, I. Mazzei, M. Monteleone, A. Fuoco, M. Carta, N.B. Makeown, R. Malpass-Evans, J.C. Jansen, Highly permeable Matrimid®/PIM-EA (H₂)-TB blend membrane for gas separation, *Polymers*, 11 (1), 46 (2018).
- [7] P.S. Tin, T.S. Chung, Y. Liu, R. Wang, S.L. Liu, K.P. Pramoda, Effects of cross-linking modification on gas separation performance of Matrimid membranes, *Journal of Membrane Science*, 225(1-2), 77-90 (2003).
- [8] T.S. Chung, S.S. Chan, R. Wang, Z.H. Lu, C.B. He, Characterization of permeability and sorption in Matrimid/C60 mixed matrix membranes, *Journal of Membrane Science*, 211 (1), 91-99 (2003).

There is no reported error for the experimental results and no discussion on reproducibility, so this raises questions about the provided data.

Reply: Many thanks for your comments. Repeated experiments were conducted and the error bars have been included in the revised manuscript. The gas separation performance and fabrication of mixed matrix membrane with gas molecule recognition window (cyclen) was found to be reproducible. The specific data has been listed as follows.

Table 1. Summary of Gas transport properties of the Matrimid and Matrimid-Cyclen-n% membranes at 1.0 atm and 25 °C (n = 1-15% indicated the degree of Cyclen mass loading).

Membrane	Gas permeability (Barrer)		
	He	H ₂	CO ₂
Matrimid	33.59±0.41	27.45±0.08	15.03±0.06
Matrimid-Cyclen-1%	47.87±0.24	43.04±0.16	16.90±0.06
Matrimid-Cyclen-3%	57.57±0.21	55.69±0.23	17.66±0.12
Matrimid-Cyclen-5%	66.43±0.22	62.17±0.32	20.07±0.08
Matrimid-Cyclen-10%	78.63±0.28	76.08±0.12	22.17±0.15
Matrimid-Cyclen-15%	85.93±0.80	80.85±0.30	24.76±0.07

Membrane	Idea Selectivity					
	He/N ₂	He/CH ₄	H ₂ /N ₂	H ₂ /CH ₄	CO ₂ /N ₂	CO ₂ /CH ₄
Matrimid	79.97±6.05	86.13±2.25	65.00±4.82	70.38±1.98	35.79±2.88	38.54±0.99
Matrimid-Cyclen-1%	145.06±1.029	368.23±2.8.10	143.47±9.71	331.08±26.34	56.33±3.81	130.00±1.033
Matrimid-Cyclen-3%	575.70±2.15	959.50±4.67	428.35±2.15	928.17±4.67	135.85±0.92	294.33±2.00

Matrimid-Cyclen-5%	664.30±2 .10	1660.75± 5.25	621.70± 3.20	1554.25 ±8.00	200.70± 0.80	501.75±3 .75
Matrimid-Cyclen-10%	561.64±4 1.36	1310.50± 4.67	543.43± 40.34	1268.00 ±3.17	158.36± 11.10	369.50±2 .33
Matrimid-Cyclen-15%	358.04±1 2.28	661.00±4 8.42	336.88± 11.59	621.92± 49.33	103.17± 3.64	190.46±1 4.71

Table 2. Summary of gas transport permeability and selectivity of the Matrimid-Cyclen-5% membrane at 1.0 atm and different temperatures.

Temperature (°C)	Gas permeability (Barrer)			Selectivity	
	He	N ₂	CH ₄	He/N ₂	He/CH ₄
Return to 20 °C	63.97±0.46	0.09±0.00	0.03±0.00	710.78±2.33	2132.33±7.00
0 °C	52.74±0.46	0.007±0.00	0.003±0.00	7534.29±24.28	17580.00±56.67
20 °C	63.13±0.46	0.10±0.00	0.03±0.00	631.30±2.60	2104.33±8.67
40 °C	80.14±0.46	0.32±0.02	0.10±0.01	250.44±15.13	801.40±83.82
60 °C	96.91±0.46	1.43±0.46	1.05±0.00	67.77±2.90	92.30±1.73
80 °C	125.12±0.46	3.45±0.46	3.05±0.46	36.27±0.36	41.02±0.87
100 °C	202.55±0.46	9.76±0.46	9.42±0.46	20.75±0.40	21.73±0.58
Return to 20 °C	68.69±0.27	0.12±0.01	0.03±0.00	572.42±49.85	2289.67±8.00

Table 3. Summary of gas transport permeability and selectivity of the Matrimid-Cyclen-5% membrane at different pressures and 25°C.

Pressure (Bar)	Gas permeability (Barrer)			Selectivity	
	He	N ₂	CH ₄	He/N ₂	He/CH ₄
1 Bar	66.43±0.22	0.10±0.00	0.04±0.00	664.30±2.10	1660.75±5.25
5 Bar	66.12±0.20	0.09±0.00	0.04±0.00	734.67±2.22	1652.00±6.00
10 Bar	65.48±0.13	0.10±0.01	0.04±0.00	654.80±1.30	1637.00±3.25
15 Bar	67.08±0.32	0.11±0.01	0.05±0.00	609.82±132.85	1341.60±6.40
20 Bar	68.34±0.57	0.13±0.01	0.07±0.00	525.69±39.06	976.29±8.15
25 Bar	68.28±0.34	0.12±0.02	0.06±0.00	569.00±73.14	1137.67±19.33
30 Bar	72.21±1.33	0.10±0.00	0.01±0.005	722.10±13.70	1444.20±130.02

Return to 1
Bar 66.84±0.42 0.10±0.00 0.04±0.00 668.40±4.20 1671.00±10.50

Table 4. Summary of gas transport properties of the Matrimid and Matrimid-Cyclen-5% membrane at 1.0 atm and 25 °C during the aging process.

Membrane	Gas permeability (Barrer)		
	He	H ₂	CO ₂
Fresh	66.43±0.22	62.17±0.32	20.07±0.08
Aging for 7 days	64.22±.34	59.90±0.23	18.30±0.07
Aging for 20 days	59.96±0.53	54.68±0.14	16.67±0.17
Aging for 40 days	59.73±0.09	53.39±0.33	15.96±0.09
Aging for 110 days	54.31±0.10	50.76±0.10	14.12±0.03

Membrane	Selectivity					
	He/N ₂	He/CH ₄	H ₂ /N ₂	H ₂ /CH ₄	CO ₂ /N ₂	CO ₂ /CH ₄
Fresh	664.30±2.1 0	1660.75± 5.25	621.70± 3.20	1554.25± 8.00	200.70 ±0.80	501.75±3 .75
Aging for 7 days	713.56±67. 960	1605.50± 8.50	665.56± 68.04	1497.50± 21.22	203.33 ±21.79	457.50±1 .42
Aging for 20 days	856.57±57. 74	2998.00± 26.50	781.14± 54.20	2734.00± 6.76	238.14 ±18.02	833.50±9 .66
Aging for 40 days	995.50±1.5 0	5973.00± 9.00	889.83± 5.07	5339.00± 30.45	266.00 ±1.33	1596.00± 7.94
Aging for 110 days	1086.20±2. 00	6788.80± 12.45	1015.20 ±1.74	6345.00± 10.90	282.40 ±0.53	1765.00± 3.31

Figure 1 shows the preparation process of the Matrimid-Cyclen membrane, the dispersion of cyclen in Matrimid solutions is homogeneous. There is no Tyndall phenomenon in the clear and transparent Matrimid/Cyclen solutions, indicating that Cyclen is uniformly dispersed in the mixed solution. The reproducibility of a mixed matrix membrane depends on the dispersibility of the filler, the higher the solubility of the filler and the better the dispersion in the mixed solution, the smaller the error in the performance of the membrane material.

Figure 1. Preparation process of the Matrimid-Cyclen membranes. (a) Cyclen powder (1.0, 3.0, 5.0, 10.0 or 15.0 mg) were added to 6.0 g DMF solvent to obtain clear and transparent solutions. (b) The clean Matrimid dope solution was added to the Cyclen/DMF solution to obtain homogeneous Matrimid/Cyclen solutions. (c) The Matrimid/Cyclen solutions was injected into clean petri dishes and transferred to a vacuum oven at an ambient temperature of 120 °C for 8-hour solvent evaporation to obtain Matrimid-Cyclen membranes.

I am also concerned about the high cyclen loading in the membrane, up to 15%. There is no discussion about mechanical properties in the manuscript, but the literature on Matrimid is very clear that high levels of additives result in a very brittle membrane. I see no reason why this is not the case here. This has issues with the ability to translate the performance into a viable membrane.

Reply: Thanks for your comments. We understand your concern about the potential brittleness of the Matrimid mixed matrix membranes with higher Cyclen loadings. To address this, the dynamic mechanical thermal analysis (DMTA) was conducted on Matrimid-Cyclen samples to assess the mechanical properties across a temperature range of 50 to 350 °C. As shown in Figure 3 (g-i), the storage modulus (E') of the Matrimid-Cyclen membranes (1800 MPa) was almost completely maintained before reaching the corresponding T_g . This indicated that the membrane retains its mechanical integrity even with higher Cyclen content. Furthermore, we observed that higher Cyclen loadings did increase in the brittleness of the membranes, as seen in the decreasing peak intensity of damping ($\tan \delta$) with increasing Cyclen content (Figure 3i). These data demonstrated that this kind of gas separation membrane was endowed with good mechanical property and thermal stability, even with the higher Cyclen loading.

Figure 3. Characterization of Matrimid-Cyclen membranes. (g) The dynamic mechanical thermal analysis (DMTA) spectra of Matrimid-Cyclen membranes at temperature range from 50 to 350 °C. (h) The storage modulus (E') and (i) $\tan \delta$ of Matrimid-Cyclen membranes.

The operating temperature, feed pressure and aging behaviors follows the expected trend of mixed matrix membranes, with the presented discussion valid. The application for helium extraction is important, and it is good that the authors considered both separation from methane and nitrogen. I recommend that the manuscript be rejected, and the authors submit to a more appropriate journal.

Reply: Thank you very much for your professional comments and suggestions. We sincerely apologize for some inadequate in data analysis and descriptions in the original manuscript. As you pointed out, several issues required further attention, and we have made substantial revisions to address the concerns. Below is a detailed list of our responses to your comments. In our work, we designed the mixed matrix membranes based on several key considerations derived from numerous studies.

Firstly, Cyclen was selected as filler due to its size-sieving effect, which is beneficial for He separation. Based on the molecular dynamic diameters of He (2.60 Å), H₂ (2.89 Å), CO₂ (3.30 Å), N₂ (3.64 Å) and CH₄ (3.80 Å), the Cyclen with cavity size (3.2-3.4 Å) between the molecular dynamic diameters of He and N₂ was selected, which makes it an ideal candidate for He separation, such as He/N₂ and He/CH₄. Furthermore, the Cyclen with a certain cavity size creates a special transport channel for small gases, like He, which increases their transport paths in the membrane, counteracting the negative effect of the increased chain stacking density and enhancing the permeability of small gas molecules.

Secondly, the chosen filler must have good solubility and dispersion in solvents. To ensure uniform dispersion of the filler in the membrane and to minimize the formation of non-selective voids caused by filler agglomeration, we selected Cyclen for its high solubility and good dispersion in organic solvents. Cyclen was fully dissolved in DMF, and then the clean Matrimid dope solution was added to the Cyclen/DMF solution. The resulting homogeneous Matrimid/Cyclen solution, which was stirred for 24 hours and sonicated for 1 hour, remained clear and transparent, even with up to 15% Cyclen content, as shown in Figure 1.

Figure 1. (a) Cyclen powder (1.0, 3.0, 5.0, 10.0 or 15.0 mg) were added to 6.0 g DMF solvent to obtain clear and transparent solutions. (b) Matrimid dope solution. (c) The clean Matrimid dope solution was added to the Cyclen/DMF solution to obtain clear and transparent mixed Matrimid/Cyclen solutions.

Thirdly, compatibility between filler and polymer matrix. It is crucial to create mixed matrix membranes with fair matching between filler and polymer matrix to gain attractive performance. This is why we chose Cyclen as a filler. Hydrogen bonding between Cyclen and the Matrimid polymer chains immobilizes Cyclen molecules in the interchain gaps of the Matrimid membrane, resulting in homogeneously mixed matrix membranes. The SEM image (Figure 2) shows the surface and interface of the membrane, confirming the good interfacial compatibility between Cyclen and Matrimid.

Figure 2. SEM images of the cross section of the Matrimid-Cyclen-n% membranes up to $\times 50,000$ magnification (n% is the Cyclen mass loading and was varied from 1 to 15 wt%, n = 0, 1, 3, 5, 10, 15).

Fourthly, Helium (He) purification requirements. Helium is an extremely scarce, non-renewable gas with a strong safety profile that is an essential resource for the development of many high-tech industries, such as the aerospace industry, electronics, medical equipment, and scientific research. On Earth, He occurs in very low concentrations, mainly in the atmosphere and in natural gas. Helium is commonly extracted industrially from high-pressure natural gas. However, the initial concentration of He in most natural gas reservoirs is still as low as 0.3%, and to extract helium from natural gas with ultra-low He content, the gas separation membranes must possess He/CH₄ selectivity greater than 1000. Highly permeable polymers rarely provide sufficient selectivity for high-purity He extraction, and Matrimid was chosen as the polymer matrix due to its dense stacking of polymer chains. XRD characterization revealed that the strong hydrogen bonding between Cyclen and Matrimid modulated the chain stacking of the mixed matrix membranes, which reduce the pore size, narrow the pore size distribution and enhances the selectivity of the membrane by modulating the chain spacing.

We have verified the success of the hybrid matrix membrane design with ultra-high helium selectivity to achieve excellent separation performance.

Reviewer #4 (Remarks to the Author):

This study reports the preparation of mixed matrix membranes by incorporating Cyclen molecules into Matrimid polyimide, and explores their application in gas separation. The approach of membrane development is certainly innovative. I have carefully reviewed the revised manuscript and the authors' responses to the previous reviewers' comments. The authors have provided a comprehensive set of characterization data and presented a plausible explanation for the polymer-filler interactions—primarily through hydrogen bonding—which may play a critical role in enhancing gas selectivity. Most of the previous concerns have been addressed appropriately. In my view, the revised manuscript could be considered for publication in Nature Communications, but the authors must address the remaining concerns regarding data accuracy and reproducibility, especially the membranes showed record-high gas separation selectivity.

Reply: Thanks for your valuable comment. A detailed list of responses to the detailed comments is provided below.

1. A key concern raised by previous reviewers pertains to the unusually record-high gas separation selectivity of the Matrimid-Cyclen mixed matrix membranes. I share Reviewer 1's skepticism about the record-high selectivities reported. I strongly recommend that the authors re-evaluate their results and, if possible, validate the membrane performance through collaboration with other membrane research groups, either in China or internationally, to rule out any inadvertent errors.

Reply: Thanks for your comments. The gas separation performance were re-evaluated for both pure and mixed gases in National University of Singapore. It should be noted that the Matrimid-Cyclen membranes have high gas separation performance but not the world record-high membranes as shown in Table S6. Comparing the gas separation performance of our membranes in this work with that reported in other literature (Figure 1 and Table 1) [1-7], although the Matrimid-Cyclen membrane exhibits higher He/CH₄ selectivity than some membranes, its permeability is slightly lower than that of others. Furthermore, the He/CH₄ selectivity of some fresh membranes even surpasses that of Matrimid-Cyclen membrane. One study published in Angewandte Chemie (July 2025) also employed Matrimid as the polymer matrix [8]. Under optimal conditions, the fresh membrane exhibited a He/CH₄ separation selectivity of 1790. These comparative data demonstrate that Matrimid-based membranes can achieve high selectivity for He separation.

Figure 1. Gas separation performance of Matrimid-Cyclen membranes compared with other gas separation membranes reported for He separation. (a) He/N₂ and (b) He/CH₄.

Table 1. Summary of literature He/N₂ and He/CH₄ separation performance.

Membrane Name	He Permeability (Barrer)	He/N ₂ Selectivity	He/CH ₄ Selectivity	Operating conditions	Gas Type	Reference
TR-M1	388.90	62.30	78.40	1.0 atm, 25 °C	Pure	21
TR-M2	921.50	46.00	62.00	1.0 atm, 25 °C	Pure	21
TB-700	210.20	519.70	639.20	2.0 atm, 25 °C	Pure	22
TB-750	160.30	1112.70	2524.40	2.0 atm, 25 °C	Pure	22
TB/PSS-550	400.20	176.60	327.30	2.0 atm, 25 °C	Pure	22
PIM-1	1336.00	7.00	4.00	--, 35 °C	Pure	23
FPIM-1	824.00	420.00	1005.00	--, 35 °C	Pure	23
FPIM-5	754.00	857.00	3770.00	--, 35 °C	Pure	23
FPIM-10	826.00	346.00	1271.00	--, 35 °C	Pure	23
AO-PIM-1	412.00	26.80	27.60	2.0 atm, 35 °C	Pure	24
PIM-EA-TB	2570.00	4.90	3.70	1.0 atm, 25 °C	Pure	25
PIM-MP-TB	1310.00	6.60	5.00	1.0 atm, 25 °C	Pure	26

PIM-TMN-Trip	2300	5.80	3.20	1.0 atm, 25 °C	Pure	27
Matrimid	33.59	79.97	86.13	1.0 atm, 25 °C	Pure	This work
Matrimid-Cyclen-1%	47.87	145.06	368.23	1.0 atm, 25 °C	Pure	This work
Matrimid-Cyclen-3%	57.57	575.70	959.50	1.0 atm, 25 °C	Pure	This work
Matrimid-Cyclen-5%	66.43	664.30	1660.75	1.0 atm, 25 °C	Pure	This work
Matrimid-Cyclen-10%	78.63	561.64	1310.50	1.0 atm, 25 °C	Pure	This work
Matrimid-Cyclen-15%	85.93	358.04	661.00	1.0 atm, 25 °C	Pure	This work

Supplementary References

[1] L. Wang, Y. Li, P. Zhang, X.F. Chen, P. Nian, Y.B. Wei, H.S. Lu, X.H. Gu, X.R. Wang, Thermally rearranged poly (benzoxazole-co-imide) composite membranes on α -Al₂O₃ support for helium extraction from natural gas, *Journal of Membrane Science*, 657, 120614 (2022).

[2] H.F. Guo, J. Wei, Y.L. Ma, Z.K. Qin, X.H. Ma, R. Selyanchyn, B.D. Wang, X.Z. He, B. Tang, L. Yang, L. Yao, W.J. Jiang, Y.F. Zhuang, D.G. Yin, X. Li, Z.D. Dai, Carbon molecular sieve membranes fabricated at low carbonization temperatures with novel polymeric acid porogen for light gas separation, *Separation and Purification Technology*, 317, 123883 (2023).

[3] X.H. Ma, K.H. Li, Z.Y. Zhu, H. Dong, J. Lv, Y.G. Wang, I. Pinnau, J.X. Li, B.W. Chen, Y. Han, High-performance polymer molecular sieve membranes prepared by direct fluorination for efficient helium enrichment, *Journal of Materials Chemistry A*, 9 (34), 18313-18322 (2021).

[4] R. Swaidan, B.S. Ghanem, E. Litwiller, I. Pinnau, Pure-and mixed-gas CO₂/CH₄ separation properties of PIM-1 and an amidoxime-functionalized PIM-1, *Journal of membrane science*, 457, 95-102 (2014).

[5] M. Carta, R. Malpass-Evans, M. Croad, Y. Rogan, J.C. Jansen, P. Bernardo, F. Bazzarelli, N.B. Mckeown, An efficient polymer molecular sieve for membrane gas separations, *Science*, 339 (6117), 303-307 (2013).

[6] R. Williams, L.A. Burt, E. Esposito, J.C. Jansen, E. Tocci, C. Rizzuto, M. Lanč, M. Carta, N.B. Mckeown, A highly rigid and gas selective methanopentacene-based polymer of intrinsic microporosity derived from Tröger's base polymerization, *Journal of Materials Chemistry A*, 6 (14), 5661-5667 (2018).

[7] I. Rose, C.G. Bezzu, M. Carta, B. Comesaña-Gándara, E. Lasseguette, M.C. Ferrari, P. Bernardo, G. Clarizia, A. Fuoco, J.C. Jansen, K.E. Hart, T.P. Liyana-Arachchi, C.M. Colina, N.B. McKeown, Polymer ultrapermeability from the inefficient packing of 2D chains, *Nature Materials*, 16 (9), 932-937 (2017).

[8] C. Wang, X.B. Chen, X. Liu, Z.Y. Li, R.X. Liu, S.J. Luo, S.J. Zhang, Plasma-Engineered Sub-10 nm Surface Fluorination Enables Ultrasensitive Hollow Fiber Membranes. *Angewandte Chemie International Edition*, e202512119 (2025).

We re-evaluated the performance of the membrane based on the separation results of mixed gases. The mixed-gas permeability and selectivity of Matrimid-Cyclen-5% membrane were lower than ideal-gas permeability and selectivity (Figure 2) because of the competitive sorption of CO₂ in the membrane. For the Matrimid-Cyclen-5% membrane, the CO₂ permeability decreased from 20.07 to 17.66 Barrer, the CO₂/N₂ ideal-gas selectivity was ~200.7 (Table 2), whereas the CO₂/N₂ mixed-gas selectivity reached ~185.3 (CO₂/N₂=15vol.%/85vol.%). The mixed-gas separation performance verifies the reliable gas separation selectivity of the membranes developed in this work.

Figure 2. Pure-gas and mixed-gas separation performance of Matrimid-Cyclen-5% membrane.

Table 2. Summary of gas transport permeability and selectivity of the Matrimid-Cyclen-5% membrane at 1.0 atm and 25°C.

Gas Type	CO ₂ permeability (Barrer)	CO ₂ /N ₂ Selectivity
Pure Gas	20.07±0.08	200.7±0.80
Mixed-15:85	17.66±0.12	185.3±0.91

Following your suggestion, we have added a comparative figure incorporating data from more studies on polymer-based membranes for He separation to the revised manuscript (lines 413-416, 959-964 and 970-971).

Revised section:

The selectivities of Matrimid-Cyclen membrane even exceeded those of some carbon molecular sieve (CMS) membranes, which is one of the highest values reported for He gas separation membrane (Supplementary Figure S10).

Figure S10. Gas separation performance of Matrimid-Cyclen membranes compared with other gas separation membranes reported for He separation. (a) He/N₂ and (b) He/CH₄. (The red symbols and blue symbols are gas data of this work, while the others are gas data of literature)

Table S6. Summary of literature He/N₂ and He/CH₄ separation performance.

Membrane Name	He Permeability (Barrer)	He/N ₂ Selectivity	He/CH ₄ Selectivity	Operating conditions	Gas Type	Reference
Matrimid-Cu-BTC-R-40	66.40	265.80	369.10	5.0 atm, 35 °C	Pure	1
Matrimid-Cu-BTC-30	51.80	193.40	369.10	5.0 atm, 35 °C	Pure	2
Matrimid-Cu-BDC-15%	20.50	341.70	410.00	7.0 atm, 35 °C	Pure	3
Matrimid-MgO-40%	45.00	86.1	118.4	10.0 atm, 35 °C	Pure	4
Matrimid-CNFs-10%	16.10	89.50	179.50	20.0 atm, 35 °C	Pure	5
Matrimid-PIM-EA(H ₂)-TB-50%	197.00	28.84	21.55	1.0 atm, 25 °C	Pure	6
Matrimid-p-xylenedia mine-CL-14	21.70	112.00	155.00	10.0 atm, 35 °C	Pure	7
Matrimid-C60-2.5%	21.00	91.70	148.80	10.0 atm, 35 °C	Pure	8
HPEI	42.10	91.52	323.85	--	Pure	9
6FDA-APAF-TR1h	101.10	53.50	69.72	--	Pure	9
R-TR-PBOIb-1h	130.00	72.20	139.79	1.0 atm, 35 °C	Pure	10
APAF5-ODA5	93.20	43.14	81.04	--, 35 °C	Pure	11

ODA10	43.80	69.52	125.14	--, 35 °C	Pure	11
6F-APAF-Ac	33.00	84.00	106.00	3.0 atm, 30 °C	Pure	12
Nafion	37.00	154.17	445.78	2.0 atm, 35 °C	Pure	13
Nafion-H+	29.10	161.67	363.75	4.0 atm, 30 °C	Pure	14
PI-1.00	59.00	65.56	134.09	2.0 atm, 35 °C	Pure	15
IPI-0.50	75.00	50.00	76.00	2.0 atm, 35 °C	Pure	15
6FDA/PMDA (25/75)-TAB	35.70	360.00	1600.0 0	10.0 atm, 35 °C	Pure	16
TR-M1	388.90	62.30	78.40	0.10 atm, 25 °C	Pure	17
TB-600	450.30	145.70	263.30	2.0 atm, 25 °C	Pure	18
TB-700	210.20	519.70	639.20	2.0 atm, 25 °C	Pure	18
AO-PIM-1	412.00	26.80	27.60	2.0 atm, 35 °C	Pure	19
TPBI-(H3PO4)0.33	18.30	248.00	870.00	6.8 atm, 25 °C	Pure	20
TR-M1	388.90	62.30	78.40	1.0 atm, 25 °C	Pure	21
TR-M2	921.50	46.00	62.00	1.0 atm, 25 °C	Pure	21
TB-700	210.20	519.70	639.20	2.0 atm, 25 °C	Pure	22
TB-750	160.30	1112.70	2524.4 0	2.0 atm, 25 °C	Pure	22
TB/PSS-550	400.20	176.60	327.30	2.0 atm, 25 °C	Pure	22
PIM-1	1336.00	7.00	4.00	--, 35 °C	Pure	23
FPIM-1	824.00	420.00	1005.0 0	--, 35 °C	Pure	23
FPIM-5	754.00	857.00	3770.0 0	--, 35 °C	Pure	23
FPIM-10	826.00	346.00	1271.0 0	--, 35 °C	Pure	23
AO-PIM-1	412.00	26.80	27.60	2.0 atm, 35 °C	Pure	24

PIM-EA-TB	2570.00	4.90	3.70	1.0 atm, 25 °C	Pure	25
PIM-MP-TB	1310.00	6.60	5.00	1.0 atm, 25 °C	Pure	26
PIM-TMN-Trip	2300	5.80	3.20	1.0 atm, 25 °C	Pure	27
Matrimid	33.59	79.97	86.13	1.0 atm, 25 °C	Pure	This work
Matrimid-Cyclen-1%	47.87	145.06	368.23	1.0 atm, 25 °C	Pure	This work
Matrimid-Cyclen-3%	57.57	575.70	959.50	1.0 atm, 25 °C	Pure	This work
Matrimid-Cyclen-5%	66.43	664.30	1660.7 5	1.0 atm, 25 °C	Pure	This work
Matrimid-Cyclen-10%	78.63	561.64	1310.5 0	1.0 atm, 25 °C	Pure	This work
Matrimid-Cyclen-15%	85.93	358.04	661.00	1.0 atm, 25 °C	Pure	This work

2. The authors should carefully re-examine their raw data and the time-lag profiles. Based on my understanding, Matrimid typically exhibits low gas permeability for CH₄ and N₂ (<0.5 Barrer). In this study, the reported CH₄ permeability is below 0.1 Barrer, for instance, the Matrimid-Cyclen-5% membrane shows a CH₄ permeability of only 0.04 Barrer, which further declines to 0.001 Barrer after aging. At such low permeability values, the pressure increase on the permeate side would be extremely slow. In my experience, accurate measurement in this regime is highly sensitive to experimental conditions, particularly potential gas leakage. Notably, the reported vacuum pressure of 0.006579 Torr (equivalent to ~0.877 Pa) is relatively high for reliable time-lag measurements. The addition of nanofillers can increase gas sorption, prolonging the time needed to reach steady-state permeation. This can lead to underestimation of the slope and thus an artificially low gas permeability. Furthermore, leakage from the atmosphere could distort measurements. Did the authors account for and subtract gas leakage? This could be a critical source of error. The authors should revisit their raw data and provide representative time-lag profiles, including slope fittings, for key samples in the Supplementary Information. The authors should also provide more experimental details of their time-lag rig, including the accuracy of the pressure sensor.

Reply: Thank you very much for your professional suggestions. The gas permeation properties of the membranes were analyzed using the constant volume-variable pressure permeation cell. Figure 1 shows the schematic of a variable-pressure constant-volume gas permeation testing cell. After multi-layer sealing, the entire device exhibits excellent hermetic integrity, with a leakage rate of 2.68×10^{-6} Torr/s. The pressure sensor exhibited an accuracy of $\pm 0.25\%$ FS.

Figure 1. Schematic diagram of variable pressure constant-volume gas permeation cell. Abbreviations: mass flow controller (MFC), pressure regulator (PR), gas chromatography (GC).

At lower permeability values, the permeate-side pressure increases very slowly, the test duration needs to be extended to obtain more accurate permeability values. As shown in Figure 2 (*Science Advances* 11 (23), 2025, eadt7512), the CO₂ permeability was even below 0.001 Barrer, with the longest permeate-side pressure collection time being 130,000 seconds [1].

Figure 2. Effect of permeation temperature on H₂ permeability and H₂/CO₂ selectivity.

We taking the data from CH₄ samples tested using the Matrimid-Cyclen-5% membrane aged for approximately 600 days as an example, the data collection time for this group exceeded 210,000 seconds, and the data fitting results are presented in Figure 3. After excluding interference from instrument leakage, the CH₄ permeability

obtained from this data set was 0.0007 Barrer. After averaging results from multiple data sets, the CH₄ permeability of the membrane reported in the manuscript is 0.001 Barrer. The dataset obtained through prolonged testing minimizes the influence of environmental factors, ensuring reliable gas permeability measurements.

Figure 3. Single-gas CH₄ permeation plots of the Matrimid-Cyclen membrane showing time lags at 25 °C and 1 Bar.

Supplementary References

[1] G.M. Iyer, C.E. Ku, C. Zhang. Hyperselective carbon membranes for precise high-temperature H₂ and CO₂ separation, *Science Advances*, 11 (23), eadt7512 (2025).

Following your suggestion, schematic diagram of the variable-pressure constant-volume gas permeation cell and membrane cell structure were added to the revised manuscript (lines 270-271 and 946-949).

Revised section:

To investigate the gas transport properties of Matrimid and Matrimid-Cyclen membranes, gas permeation tests for He (2.60 Å), H₂ (2.89 Å), CO₂ (3.3 Å), N₂ (3.64 Å) and CH₄ (3.80 Å) were performed using a constant-volume, variable-pressure system at 25 °C under a feed pressure of 1.0 bar (Supplementary Figure S7).

Figure S7. Schematic diagram of variable pressure constant-volume gas permeation cell. Abbreviations: mass flow controller (MFC), pressure regulator (PR), gas chromatography (GC).

3. Given the reported high performance, I strongly recommend that the authors fabricate thin-film composite (TFC) membranes and measure gas permeance and selectivity for both single gases and gas mixtures, ideally with gas chromatography. The Matrimid-Cyclen solution should be suitable for casting onto porous polymeric or AAO supports. With selective layer thicknesses below 1 micron, or ideally down to ~100 nm, the gas permeance should be significantly higher, and the influence of leakage on time-lag measurements substantially reduced. Such tests would help confirm whether the TFC membranes maintain high gas permeance and selectivity (e.g., for He/CH₄ or CO₂/CH₄ mixtures). This is crucial, as the low permeability of thick films may introduce significant measurement error.

Reply: Thanks for your recognition and comments. We prepared a series of thin-film composite (TFC) membranes from Matrimid-Cyclen-5% solution on PAN, PES and AAO-1 (Pore size ranging from 100 to 500 nm) and AAO-2 (Pore size ranging from 40 to 70 nm) supports under identical conditions (Figure 1a). Performance evaluation under the same test conditions revealed that the He/CH₄ separation performance was highly dependent on the substrate, the AAO-2-supported TFC membranes demonstrated the highest selectivity of 689 (Figure 1b). Subsequently, we fabricated membranes with a thickness of approximately 10 μm, which exhibited a He permeability of 62.65 Barrer and He/CH₄ selectivity of 1510, slightly lower than that of 30 μm-thick membranes (Figure 2). This set of data excludes the influence of measurement errors, demonstrating the reliability of the data in this work.

Figure 1. Optical photograph (a) and Gas separation performance (b) of TFC membranes.

Figure 2. Optical photograph (a) and Gas separation performance (b) of Matrimid-Cyclen-5% membrane.

4. Building on the points above, the authors are encouraged to test their approach using other polymer matrices with higher intrinsic gas permeability. This would help determine whether similar enhancements can be achieved and reduce the susceptibility to measurement error.

Reply: Thanks for your comments. Common polymer matrices for mixed-matrix membranes (MMMs) in gas separation applications include polybenzimidazole (PBI), Matrimid, Pebax, and microporous polymers of intrinsic microporosity (PIM-1). PBI and Pebax membranes demonstrate significantly lower helium permeability compared to Matrimid membrane. Notably, PIM-1 membrane exhibit superior gas permeability among these materials. For membrane fabrication, PIM-1 is typically dissolved in dichloromethane (DCM). The highly polar nature of Cyclen (1,4,7,10-tetraazacyclododecane) results in extremely poor solubility in dichloromethane, as clearly demonstrated by the visible particle agglomeration shown in Figure 1. Filler agglomeration can create non-selective interfacial voids within the membrane matrix, dramatically compromising gas separation selectivity. Consequently, it is not possible to determine whether performance improvements can be achieved based on PIM-based MMMs.

Figure 1. Optical photograph of Cyclen (5.0 mg) dispersed in DCM solution (6.0 g).

5. A central hypothesis is that the Cyclen macrocycle facilitates gas transport through its cavity and narrows the polymer interchain spacing via hydrogen bonding. The authors performed control experiments with Cyclododecane and Hexacyclen. However, it would strengthen their case to include additional controls using amine-containing molecules without macrocyclic cavities, such as piperazine.

Reply: Thanks for your valuable comments. The Matrimid-Piperazine-1% membrane was prepared under identical experimental conditions, and its performance was presented in Figure 1. The He permeability of the Matrimid-piperazine-1% membrane remains comparable to that of the Matrimid membrane, while its He/CH₄ selectivity is higher than that of the Matrimid membrane. This indicates that the piperazine without macrocyclic cavities also can enhance He/CH₄ selectivity by narrowing the polymer interchain spacing via hydrogen bonding.

Thanks for your valuable comments. The Matrimid-Piperazine-1% membrane was prepared under identical experimental conditions, and its performance is presented in Figure 1. The He permeability of the Matrimid-piperazine-1% membrane remains comparable to that of the Matrimid membrane, while its He/CH₄ selectivity is higher than that of the Matrimid membrane. This indicates that the filler with cavity can effectively enhance the gas transport performance of MMMs.

Figure 1. Gas separation performance of mixed matrix membranes prepared with different fillers (Cyclen, Cyclododecane, Piperazine and Hexacyclen) at 1% loading.

We have updated the data for the Matrimid-piperazine-1% membrane in Figures 5g-h

and Table 9 of the manuscript, and made the revisions to the manuscript from line 423 to line 434.

Revised section:

Figure 5g-h show the improvement in both He permeability and He/CH₄ selectivity of MMMs formed in Matrimid with **four** different fillers, the data are summarized in Supplementary Table S9. The increase in He/CH₄ selectivity of MMMs prepared with Cyclen and Cyclododecane, which have the same intra-ring dimensions as fillers is 427% and -18%, respectively, as compared to the pristine Matrimid membrane. **This demonstrates that tightening the polymer chain stacking by hydrogen bonding can effectively enhance the separation performance of MMMs.** Compared to Piperazine without an annular cavity, the He permeability of MMMs relative to the Matrimid membrane increased by 143% and 108%, respectively. **This indicates that the piperazine without macrocyclic cavities also can enhance He/CH₄ selectivity by narrowing the polymer interchain spacing via hydrogen bonding.** Compared to Hexacyclen with a larger cavity size, the increase in He/CH₄ selectivity of MMMs over the Matrimid membrane was 428% and 210%, respectively. This indicates that the filler with suitable cavity size can improve the **gas transport efficiency and separation performance of MMMs.** The **three** sets of comparison experiments confirmed the hydrogen bonding interaction between Cyclen and Matrimid, and the suitable gas molecule recognition window size of Cyclen as the reason for improving the molecular sieving performance of the membranes, respectively.

Figure 5. Gas separation performance of Matrimid-Cyclen membranes compared with other polymeric membranes. (g-h) Gas separation performance of mixed matrix membranes (Matrimid-Cyclen-1% membrane, Matrimid-Cyclododecane-1% membrane, **Matrimid-Piperazine-1% membrane** and Matrimid-Hexacyclen-1% membrane) prepared with different fillers (Cyclen, Cyclododecane, **Piperazine** and Hexacyclen) at 1% loading.

Table S9. Summary of gas transport permeability and selectivity of the MMMs with different fillers.

Membrane	Gas permeability (Barrer)	Selectivity
----------	---------------------------	-------------

	He	Enhancement (%)	He/CH ₄	Enhancement (%)
Matrimid-Cyclen-1%	47.87	142.51	368.23	427.53
Matrimid-Cyclododecane-1%	89.27	265.76	70.84	-17.75
Matrimid-Piperazine-1%	36.23	107.86%	157.52	182.89
Matrimid-Hexacyclen-1%	99.44	296.04	180.80	209.92

6. The three-dimensional plots in Figure 4 are difficult to interpret. A clearer presentation would be to convert them to two-dimensional plots.

Reply: Thanks for your suggestions. We modified the data presentation format, converted the three-dimensional images to two-dimensional representations, and made the revisions to the manuscript from line 253.

Revised section:

Figure 4. Gas separation performance of Matrimid-Cyclen membranes. (f) Effect of operating temperature on the separation performance of Matrimid-Cyclen-5% membrane. (g) Effect of feed gas pressures on the separation performance of Matrimid-Cyclen-5% membrane. (h) Long-term stability of the Matrimid-Cyclen-5% membrane (membrane was stored in an ambient environment without the introduction of any protective gas at room temperature, small pieces of membrane were extracted from the complete membrane at different aging times for a gas separation performance test).

Reviewer #5 (Remarks to the Author):

The authors have carefully addressed the comments raised from the previous review round—the manuscript now has met the standards of Nature Communications and it can be considered for publication.

Reply: Thank the reviewer for the highly positive remarks and support of publication of this work.

Response to the Reviewers' Comments:

Reviewer #4 (Remarks to the Author):

The authors have made commendable efforts and have addressed most of the previous comments satisfactorily. However, some results still require deeper analysis. In particular, the thin-film composite (TFC) membrane data are of high significance and should be presented both in the main manuscript and in the Supporting Information. The experimental design could also be strengthened by preparing additional control samples, such as a TFC membrane made from Matrimid without the filler. The membranes should be carefully analyzed, as the TFC results are critical in determining whether the proposed membrane fabrication strategy can be scaled up for industrial applications.

Many studies on gas separation membranes report gas permeability and selectivity; however, they rarely fabricate membranes in the form of thin-film composites, and even fewer evaluate mixed-gas separation performance. In my opinion, this represents a key bottleneck that has limited the progress and industrial relevance of gas separation membranes. Since the authors have now developed highly selective materials, they should prepare TFC versions and thoroughly assess their performance.

Reply: We sincerely appreciate your comments and professional suggestions. Following your suggestions, the manuscript has been thoroughly revised as detailed below:

Comprehensive presentation and assessment of TFC membrane gas performance data. First, we have thoroughly addressed the need for a comprehensive assessment of gas performance data in TFC membranes. To this end, the revised manuscript presents expanded performance data and discussion specifically on the Matrimid-Cyclen TFC membranes, covering their performance across different thicknesses, stability, and comparisons of these properties against those of other membranes reported in the literature (see revised manuscript in Figures 7c-d, Supplementary Figure S14, Supplementary Tables 11-12).

Second, we conducted additional experiments with control samples to further elucidate the role of the filler. Pure Matrimid TFC membranes (without filler) were fabricated under identical processing conditions. The data and discussion for Matrimid and Matrimid-Cyclen TFC membranes have been added to the revised manuscript (see revised manuscript in Supplementary Table 13).

In addition, we have also performed a detailed morphological analysis of the TFC membranes. The top surface and cross-sectional images, along with their analysis, have been added to the revised manuscript (see Figures 7a-b).

Furthermore, we evaluated both the pure and mixed-gas (CO₂/N₂, 50/50 vol%) performance of the Matrimid-Cyclen-1.8 μm TFC membrane. Under these mixed-gas conditions, competitive adsorption between gas molecules leads to a decrease in both CO₂ permeance and CO₂/N₂ selectivity compared to single-gas permeation. Notably, the membrane maintains a CO₂/N₂ separation selectivity above 80, demonstrating its outstanding molecular sieving capability under industrially

relevant conditions.

In summary, all suggested revisions have been carefully addressed and we have integrated control experiments, morphological characterization, and mixed-gas performance evaluation into the revised manuscript to enhance the integrity of the work.

Added:

Figure 7. SEM images and gas separation performance of Matrimid-Cyclen-5% TFC membrane. (a) Cross section and (b) surface SEM images of the Matrimid-Cyclen TFC membrane with different thicknesses. (c) Gas separation performances of the Matrimid-Cyclen TFC membrane with different thickness. (d) Comparison of He/CH₄ separation performance of Matrimid-Cyclen TFC membranes with available literature data.

Thin-film composite (TFC) membranes, composed of a highly selective thin layer supported by a mechanically robust porous support, have been employed for practical gas separation [61-62]. Matrimid-Cyclen TFC membranes were fabricated on AAO supports using a scalable bar-coating method [63]. Scanning electron microscopy (SEM) images (Figures 7a-b) showed no significant aggregation or particulate formation. As the membrane thickness increased, the membrane structure evolved to become denser and smoother (Figure 7a). For the calculation of gas permeance, the thickness of the Matrimid-Cyclen TFC membrane's selective layer was determined from cross-sectional SEM images, with values of approximately 110 nm, 1.8 μm, and 5.6 μm. Figure 7c illustrates the gas permeance and selectivity of the Matrimid-Cyclen TFC membranes with average thicknesses ranging from 110 nm to 5.6 μm (Supplementary Table S11). The He permeance decreased and the He/CH₄

selectivity increased as the membrane thickness increased. Reproducibility was confirmed by testing six randomly selected Matrimid-Cyclen TFC membranes with an average thickness of 1.8 μm , which exhibited consistent He permeance and He/CH₄ selectivity (Supplementary Figure S14). The average He permeance was 102 \pm 6 GPU, with a corresponding He/CH₄ selectivity of 728 \pm 59. Figure 7d illustrates the He/CH₄ selectivity and He permeance for Matrimid-Cyclen TFC membranes and other types of membranes reported in the literature (please see the detailed comparison in Supplementary Table S12). Although the membrane's gas permeance is not as high as that of some polymer membranes, Matrimid-Cyclen-1.8 μm exhibits higher He/CH₄ selectivity than most reported membranes. Moreover, its performance still exceeds the 2008 Robeson upper bound. In addition, the gas permeance of the AAO support, pure Matrimid TFC membranes (without filler), and Matrimid-Cyclen TFC membranes (with filler) is summarized in Supplementary Table S13. Notably, the Matrimid-Cyclen TFC membrane exhibits significantly higher He/CH₄ selectivity than the pure Matrimid TFC membrane, the critical role of the cyclen filler in enhancing membrane selectivity.

Figure S14. He separation performance of Matrimid-Cyclen TFC membranes (numbers 1 to 6 mean six different batches of membranes).

Table S11. Summary of gas transport properties for membranes of varying average thicknesses on AAO supports at 1.0 atm and 25 °C.

Membrane	Gas permeance (GPU)		
	He	H ₂	CO ₂
AAO supports	7747.92 \pm 115.9	6656.22 \pm 41.38	3895.34 \pm 81.29
Matrimid-Cyclen-110nm	1385.95 \pm 16.75	1111.54 \pm 25.62	458.02 \pm 45.12
Matrimid-Cyclen-1.8 μm	101.93 \pm 6.41	68.32 \pm 4.16	22.58 \pm 3.11
Matrimid-Cyclen-5.6 μm	18.25 \pm 0.36	15.37 \pm 0.25	5.15 \pm 0.16

Membrane	ideal Selectivity					
	He/N ₂	He/CH ₄	H ₂ /N ₂	H ₂ /CH ₄	CO ₂ /N ₂	CO ₂ /CH ₄
AAO supports	1.88	2.56	1.62	2.20	0.96	1.29
Matrimid-Cyclen-110nm	58.41	66.09	46.84	53.01	19.30	21.84
Matrimid-Cyclen-1.8μm	550.95	728.04	369.30	488.00	122.05	161.29
Matrimid-Cyclen-5.6μm	829.55	1073.53	698.64	904.12	234.10	302.94

Table S12. Summary of literature He/N₂ and He/CH₄ separation performance.

Membrane Name	He permeance (GPU)	He/N ₂ Selectivity	He/CH ₄ Selectivity	Operating conditions	Gas Type	Reference
40nm PIM-1	23795.70	--	2.80	0.0 atm, --	Mixed-50:50	88
80nm PIM-1	21087.00	--	7.00	0.0 atm, --	Mixed-50:50	88
PIM-1	29.70	6.60	4.50	7.0 atm, 35 °C	Pure	24
PVDC-PVC	38.01	--	27.81	--, 25 °C	Pure	89
Metal oxide/PIM-PI-0	324.00	11.83	11.74	--, 21 °C	Pure	90
Metal oxide/PIM-PI-5	107.00	66.88	191.07	--, 21 °C	Pure	90
TFC-10%	24.68	--	22.34	2.0 atm, 25 °C	Pure	91
TFC-10%-450	232.10	--	47.70	2.0 atm, 25 °C	Pure	91
TFC-10%-500	916.04	--	65.24	2.0 atm, 25 °C	Pure	91
TFC-10%-550	1248.48	--	149.04	2.0 atm, 25 °C	Pure	91
TFC-2%-550	3299.49	--	33.03	2.0 atm, 25 °C	Pure	91
Ultem 1000	65.40	158.90	201.00	1.5 atm, 35 °C	Pure	92
Cellulose acetate	106.00	97.00	97.00	--, 30 °C	Pure	93
Aromatic polyamide	7.65	85.00	43.81	20.0 atm, 25 °C	Pure	94

Torlon	40.00	--	345.00	80.0 atm, 35 °C	Pure	95
P84	1.17	23.40	16.71	0.001 atm, 25 °C	Pure	96
PIM-PI-AAO-72nm	15.70	39.25	157.00	4.0 atm, 35 °C	Pure	97
PIM-PI-AAO-72nm-Age	7.70	128.33	85.56	4.0 atm, 35 °C	Pure	97
Matrimid-900	3.10	--	16700.00	7.0 atm, 35 °C	Pure	98
Matrimid-1	195.00	--	>12.00	5.0 atm, 35 °C	Pure	99
Matrimid-2	157.00	--	>109.00	5.0 atm, 35 °C	Pure	99
Matrimid-3	3313.00	--	>1.90	5.0 atm, 35 °C	Pure	99
BTPDA	643.00	--	31.40	3.0 atm, 35 °C	Pure	100
E/A-16-35°C	1577.80	--	101.80	3.0 atm, 35 °C	Pure	100
BTESE	9038.80	--	1.50	3.0 atm, 35 °C	Pure	100
[Cu ₂ (bza) ₄ (pyz)] _n	7.90	3.91	7.29	0.5 atm, 20 °C	Pure	101
IRMOF-3	3046.00	--	1.61	1.0 atm, --	Pure	102
MMOF	41.00	--	2.80	1.0 atm, --	Pure	103
ZIF-62	51.60	17.40	13.90	1.0 atm, 25 °C	Pure	104
SAPO-34	681.70	--	13.80	1.4 atm, --	Mixed- 50:50	105
ZIF-8	565.00	4.28	4.60	1.0 atm, 24 °C	Pure	106
Zr-MOF (fumarate)	154.70	--	21.20	2.0 atm, 35 °C	Pure	107
{001 }-oriented Zr-MOF(fumarate)	1493.30	--	42.80	2.0 atm, 35 °C	Pure	107
Teflon AF 2700	10500.00	3.89	4.60	3.5 atm, 22 °C	Pure	108
Hyflon AD 60	2600.00	14.44	35.00	3.5 atm, 22 °C	Pure	108
Cytop	790.00	43.89	130.00	3.5 atm, 22 °C	Pure	108

Copolymer A	1400.00	48.28	260.00	3.5 atm, 22 °C	Pure	108
Copolymer B3	770.00	78.57	480.00	3.5 atm, 22 °C	Pure	108
Poly(PFMMD)	2160.00	31.77	108.00	3.5 atm, 22 °C	Pure	109
PFMMD-co-PFMD 2	2320.00	57.28	332.00	3.5 atm, 22 °C	Pure	109
PFMMD-co-PFMD 3	2970.00	67.50	405.00	3.5 atm, 22 °C	Pure	109
PFMMD-co-CTFE 2	1120.00	94.12	473.00	3.5 atm, 22 °C	Pure	109
PFMMD-co-CTFE 3	804.00	164.08	900.00	3.5 atm, 22 °C	Pure	109
PBDI	45	--	1380	1.0 atm, 100 °C	Mixed- 50:50	110
PF-SPF	65.00	866.67	2166.67	1.0 atm, 35 °C	Pure	111
Matrimid-Cyclen-110nm	1385.95	58.41	66.09	1.0 atm, 25 °C	Pure	This work
Matrimid-Cyclen-1.8µm	101.93	550.95	728.04	1.0 atm, 25 °C	Pure	This work
Matrimid-Cyclen-5.6µm	18.25	829.55	1073.53	1.0 atm, 25 °C	Pure	This work

Table S13. Summary of gas transport properties of the AAO supports (Pore size < 20 nm), and the TFC membranes made from Matrimid and from Matrimid-Cyclen at 1.0 atm and 25 °C (Cyclen mass loading is 5%).

Membrane	Gas permeance (GPU)		
	He	H ₂	CO ₂
AAO supports	7747.92±115.9	6656.22±41.38	3895.34±81.29
Matrimid-1.2µm	109.50±6.94	92.22±4.75	30.90±1.52
Matrimid-Cyclen-1.8µm	101.93±6.41	68.32±4.16	22.58±3.11

Membrane	ideal Selectivity					
	He/N ₂	He/CH ₄	H ₂ /N ₂	H ₂ /CH ₄	CO ₂ /N ₂	CO ₂ /CH ₄
AAO supports	1.88	2.56	1.62	2.20	0.96	1.29
Matrimid-1.2µm	44.51	55.30	37.49	46.58	12.56	15.61

1. The authors prepared TFC membranes by coating a Matrimid/Cyclen-5% solution onto PAN and PES supports. These membranes show very high gas permeance (>10,000 GPU) (e.g. 30,000 GPU on PAN) and high He/CH₄ selectivity (\approx 100). However, these values appear too good to be true, as the reported permeance is unusually high for this type of system. The authors should carefully verify these measurements and rule out possible experimental or data interpretation errors. Furthermore, they should evaluate the permeance and selectivity for other gases (e.g., H₂, CO₂, N₂) to provide a more comprehensive understanding of the membrane's gas separation performance.

Reply: Thank you very much for your professional suggestions, which prompt us to re-examine our data and experimental methods. Upon re-examining all test results and data, we identified that the issue originated from the determination of membrane thickness.

Gas permeation tests of the membranes were conducted on a variable-pressure constant-volume gas permeation cell. The gas permeability coefficient through the membrane was calculated according to the steady state pressure increment.

Due to the low concentration of the casting solution, it penetrated into the substrate, causing significant solution retention within the support layer. Our initial method for calculating membrane thickness failed to account for this permeation into the support and the partial dissolution of the support by the casting solution. This led to a substantial overestimation of the true selective layer thickness, leading to an overestimation of permeability by several orders of magnitude.

Gas permeation performance was reported using two common units: GPU (1 GPU=1 \times 10⁻⁶ cm³ (STP) \cdot cm⁻² \cdot s⁻¹ \cdot cmHg⁻¹) and Barrer (1 Barrer=1 \times 10⁻¹⁰cm³ (STP) \cdot cm \cdot cm⁻² \cdot s⁻¹ \cdot cmHg⁻¹). We initially made a calculation error when converting gas permeability (in Barrer) to permeance (in GPU) by incorrectly multiplying the permeability value by 10, which was not the correct conversion between these two units. This error was identified during a comprehensive review of our calculations.

The revised calculation method involves first directly obtaining gas permeability and then deriving the permeance of the TFC membrane according to equation (1) [1-2].

$$J = \frac{P}{l} \quad (1)$$

Where J denotes the gas permeance of the TFC membrane in GPU (1 GPU=1 \times 10⁻⁶ cm³ (STP) \cdot cm⁻² \cdot s⁻¹ \cdot cmHg⁻¹), P denotes the gas permeance coefficient in Barrer (1 Barrer=1 \times 10⁻¹⁰cm³ (STP) \cdot cm \cdot cm⁻² \cdot s⁻¹ \cdot cmHg⁻¹), l represents the thickness of the membrane selective-layer (μ m).

To prevent solution permeation from interfering with the accurate measurement of the effective membrane thickness and the corresponding gas permeability calculations, anodic aluminum oxide (AAO) with pore sizes smaller than 20 nm was selected as the support. Cross-sectional SEM images reveal the minimal casting solution within substrate (Figure 1). The gas permeance of the thin-film composite (TFC) membrane was shown in Figure 2. Six randomly selected samples of the Matrimid-Cyclen TFC

membrane (average thickness: 1.8 μ m) exhibited consistent He permeance and He/CH₄ selectivity. The average He permeance was 101.93 \pm 6.41 GPU, with a corresponding He/CH₄ selectivity of 728.04 \pm 58.83.

Figure 1. Contact surface between the AAO support and the selective layer.

Figure 2. He separation performance of TFC membranes (numbers 1 to 6 mean six different batches of Matrimid-Cyclen TFC membranes)

Supplementary References

[1] W.J. Fu, L. Zhang, J.C. Liu, T. Yang, M.X. Sun, X.H. Ma, Y.P. Zhao, L. Chen, Ceramic-based composite membranes decorated by incorporating ZIF-8 and PDMS for highly efficient CO₂/N₂ separation, *Separation and Purification Technology*, 352, 128142 (2025).

[2] C. Feng, Y.L. Ma, J. Wei, M. Deng, Z.K. Qin, J.Y. Liu, B. Tang, X.H. Ma, J. Liu, W.J. Jiang, L. Yang, L. Yao, Z. Changwu, Z.D. Dai, 2D zeolite-based thin film nanocomposite membranes for efficient CO₂ separation, *Industrial & Engineering Chemistry Research*, 63 (25), 11134-11144. (2024).

We have also evaluated the permeance and selectivity for other gases, and compared these properties with those of other types of membranes reported in the literature. The results and discussion have been added to the revised manuscript.

Added:

Figure 7. SEM images and gas separation performance of Matrimid-Cyclen-5% TFC membrane. (c) Gas separation performances of the Matrimid-Cyclen TFC membrane with different thicknesses. (d) Comparison of He/CH₄ separation performance of Matrimid-Cyclen TFC membranes with available literature data.

Figure 7c illustrates the gas permeance and selectivity of the Matrimid-Cyclen TFC membranes with average thicknesses ranging from 110 nm to 5.6 μm (Supplementary Table S11). The He permeance decreased and the He/CH₄ selectivity increased as the membrane thickness increased. Reproducibility was confirmed by testing six randomly selected Matrimid-Cyclen TFC membranes with an average thickness of 1.8 μm, which exhibited consistent He permeance and He/CH₄ selectivity (Supplementary Figure S14). The average He permeance was 102±6 GPU, with a corresponding He/CH₄ selectivity of 728±59. Figure 7d illustrates the He/CH₄ selectivity and He permeance for Matrimid-Cyclen TFC membranes and other types of membranes reported in the literature (please see the detailed comparison in Supplementary Table S12). Although the membrane's gas permeance is not as high as that of some polymer membranes, Matrimid-Cyclen-1.8 μm exhibits higher He/CH₄ selectivity than most reported membranes. Moreover, its performance still exceeds the 2008 Robeson upper bound.

Figure S14. He separation performance of Matrimid-Cyclen TFC membranes (numbers 1 to 6 mean six different batches of membranes).

Table S11. Summary of gas transport properties for membranes of varying average thicknesses on AAO supports at 1.0 atm and 25 °C.

Membrane	Gas permeance (GPU)		
	He	H ₂	CO ₂
AAO supports	7747.92±115.9	6656.22±41.38	3895.34±81.29
Matrimid-Cyclen-110nm	1385.95±16.75	1111.54±25.62	458.02±45.12
Matrimid-Cyclen-1.8μm	101.93±6.41	68.32±4.16	22.58±3.11
Matrimid-Cyclen-5.6μm	18.25±0.36	15.37±0.25	5.15±0.16

Membrane	ideal Selectivity					
	He/N ₂	He/CH ₄	H ₂ /N ₂	H ₂ /CH ₄	CO ₂ /N ₂	CO ₂ /CH ₄
AAO supports	1.88	2.56	1.62	2.20	0.96	1.29
Matrimid-Cyclen-110nm	58.41	66.09	46.84	53.01	19.30	21.84
Matrimid-Cyclen-1.8μm	550.95	728.04	369.30	488.00	122.05	161.29
Matrimid-Cyclen-5.6μm	829.55	1073.53	698.64	904.12	234.10	302.94

Table S12. Summary of literature He/N₂ and He/CH₄ separation performance.

Membrane Name	He permeance (GPU)	He/N ₂ Selectivity	He/CH ₄ Selectivity	Operating conditions	Gas Type	Reference
40nm PIM-1	23795.70	--	2.80	0.0 atm, --	Mixed-50:50	88
80nm PIM-1	21087.00	--	7.00	0.0 atm, --	Mixed-50:50	88
PIM-1	29.70	6.60	4.50	7.0 atm, 35 °C	Pure	24
PVDC-PVC	38.01	--	27.81	--, 25 °C	Pure	89
Metal oxide/PIM-PI-0	324.00	11.83	11.74	--, 21 °C	Pure	90
Metal oxide/PIM-PI-5	107.00	66.88	191.07	--, 21 °C	Pure	90
TFC-10%	24.68	--	22.34	2.0 atm, 25 °C	Pure	91

TFC-10%-450	232.10	--	47.70	2.0 atm, 25 °C	Pure	91
TFC-10%-500	916.04	--	65.24	2.0 atm, 25 °C	Pure	91
TFC-10%-550	1248.48	--	149.04	2.0 atm, 25 °C	Pure	91
TFC-2%-550	3299.49	--	33.03	2.0 atm, 25 °C	Pure	91
Ultem 1000	65.40	158.90	201.00	1.5 atm, 35 °C	Pure	92
Cellulose acetate	106.00	97.00	97.00	--, 30 °C	Pure	93
Aromatic polyamide	7.65	85.00	43.81	20.0 atm, 25 °C	Pure	94
Torlon	40.00	--	345.00	80.0 atm, 35 °C	Pure	95
P84	1.17	23.40	16.71	0.001 atm, 25 °C	Pure	96
PIM-PI-AAO-72nm	15.70	39.25	157.00	4.0 atm, 35 °C	Pure	97
PIM-PI-AAO-72nm-Ag e	7.70	128.33	85.56	4.0 atm, 35 °C	Pure	97
Matrimid-900	3.10	--	16700.0 0	7.0 atm, 35 °C	Pure	98
Matrimid-1	195.00	--	>12.00	5.0 atm, 35 °C	Pure	99
Matrimid-2	157.00	--	>109.00	5.0 atm, 35 °C	Pure	99
Matrimid-3	3313.00	--	>1.90	5.0 atm, 35 °C	Pure	99
BTPDA	643.00	--	31.40	3.0 atm, 35 °C	Pure	100
E/A-16-35°C	1577.80	--	101.80	3.0 atm, 35 °C	Pure	100
BTESE	9038.80	--	1.50	3.0 atm, 35 °C	Pure	100
[Cu ₂ (bza) ₄ (pyz)] _n	7.90	3.91	7.29	0.5 atm, 20 °C	Pure	101
IRMOF-3	3046.00	--	1.61	1.0 atm, --	Pure	102
MMOF	41.00	--	2.80	1.0 atm, --	Pure	103
ZIF-62	51.60	17.40	13.90	1.0 atm, 25 °C	Pure	104

SAPO-34	681.70	--	13.80	1.4 atm, --	Mixed-50:50	105
ZIF-8	565.00	4.28	4.60	1.0 atm, 24 °C	Pure	106
Zr-MOF (fumarate)	154.70	--	21.20	2.0 atm, 35 °C	Pure	107
{001}-oriented Zr-MOF(fumarate)	1493.30	--	42.80	2.0 atm, 35 °C	Pure	107
Teflon AF 2700	10500.00	3.89	4.60	3.5 atm, 22 °C	Pure	108
Hyflon AD 60	2600.00	14.44	35.00	3.5 atm, 22 °C	Pure	108
Cytop	790.00	43.89	130.00	3.5 atm, 22 °C	Pure	108
Copolymer A	1400.00	48.28	260.00	3.5 atm, 22 °C	Pure	108
Copolymer B3	770.00	78.57	480.00	3.5 atm, 22 °C	Pure	108
Poly(PFMMD)	2160.00	31.77	108.00	3.5 atm, 22 °C	Pure	109
PFMMD-co-PFMD 2	2320.00	57.28	332.00	3.5 atm, 22 °C	Pure	109
PFMMD-co-PFMD 3	2970.00	67.50	405.00	3.5 atm, 22 °C	Pure	109
PFMMD-co-CTFE 2	1120.00	94.12	473.00	3.5 atm, 22 °C	Pure	109
PFMMD-co-CTFE 3	804.00	164.08	900.00	3.5 atm, 22 °C	Pure	109
PBDI	45	--	1380	1.0 atm, 100 °C	Mixed-50:50	110
PF-SPF	65.00	866.67	2166.67	1.0 atm, 35 °C	Pure	111
Matrimid-Cyclen-110nm	1385.95	58.41	66.09	1.0 atm, 25 °C	Pure	This work
Matrimid-Cyclen-1.8µm	101.93	550.95	728.04	1.0 atm, 25 °C	Pure	This work
Matrimid-Cyclen-5.6µm	18.25	829.55	1073.53	1.0 atm, 25 °C	Pure	This work

2. Such high permeance values are typically observed for ultrafiltration supports coated with a PDMS gutter layer. Did the authors apply a PDMS coating on the polymeric supports? This may allow the authors to prepare good quality TFC membranes.

Reply: Thanks for your comments. PDMS coating was not applied because the

polymer-based dense membrane did not require additional PDMS coating.

3. The gas permeance of the bare supports, as well as that of the TFC membranes made from pure Matrimid (without filler) and from Matrimid/Cyclen (with filler), should be reported for comparison.

Reply: Thanks for your comments. The gas permeance of the supports, pure Matrimid TFC membranes (without filler), and Matrimid-Cyclen TFC membranes (with filler) has been presented in Table 1. The Matrimid-Cyclen-TFC membrane exhibited significantly higher He/CH₄ selectivity than the pure Matrimid-TFC membrane, demonstrating that the introduction of the cyclen filler is essential for enhancing membrane selectivity.

Table 1. Summary of gas transport properties of the AAO supports (Pore size < 20 nm), and the TFC membranes made from Matrimid and from Matrimid-Cyclen at 1.0 atm and 25 °C (Cyclen mass loading is 5%).

Membrane	Gas permeance (GPU)		
	He	H ₂	CO ₂
AAO supports	7747.92±115.9	6656.22±41.38	3895.34±81.29
Matrimid-1.2µm	109.50±6.94	92.22±4.75	30.90±1.52
Matrimid-Cyclen-1.8µm	101.93±6.41	68.32±4.16	22.58±3.11

Membrane	ideal Selectivity					
	He/N ₂	He/CH ₄	H ₂ /N ₂	H ₂ /CH ₄	CO ₂ /N ₂	CO ₂ /CH ₄
AAO supports	1.88	2.56	1.62	2.20	0.96	1.29
Matrimid-1.2µm	44.51	55.30	37.49	46.58	12.56	15.61
Matrimid-Cyclen-1.8µm	550.95	728.04	369.30	488.00	122.05	161.29

To elucidate the role of the filler, we evaluated the permeance and selectivity of pure Matrimid TFC membranes (without filler) and Matrimid-Cyclen TFC membranes. The results and discussion have been supplemented in the revised manuscript.

Added:

In addition, the gas permeance of the AAO support, pure Matrimid TFC membranes (without filler), and Matrimid-Cyclen TFC membranes (with filler) is summarized in Supplementary Table S13. Notably, the Matrimid-Cyclen TFC membrane exhibits significantly higher He/CH₄ selectivity than the pure Matrimid TFC membrane, the critical role of the cyclen filler in enhancing gas selectivity.

Table S13. Summary of gas transport properties of the AAO supports (Pore size < 20

nm), and the TFC membranes made from Matrimid and from Matrimid-Cyclen at 1.0 atm and 25 °C (Cyclen mass loading is 5%).

Membrane	Gas permeance (GPU)		
	He	H ₂	CO ₂
AAO supports	7747.92±115.9	6656.22±41.38	3895.34±81.29
Matrimid-1.2µm	109.50±6.94	92.22±4.75	30.90±1.52
Matrimid-Cyclen-1.8µm	101.93±6.41	68.32±4.16	22.58±3.11

Membrane	ideal Selectivity					
	He/N ₂	He/CH ₄	H ₂ /N ₂	H ₂ /CH ₄	CO ₂ /N ₂	CO ₂ /CH ₄
AAO supports	1.88	2.56	1.62	2.20	0.96	1.29
Matrimid-1.2µm	44.51	55.30	37.49	46.58	12.56	15.61
Matrimid-Cyclen-1.8µm	550.95	728.04	369.30	488.00	122.05	161.29

4. The authors should clarify why the membranes on polymeric supports exhibit a selectivity of approximately 100, whereas the membrane coated on AAO-2 shows much lower permeance but much higher selectivity (≈ 689). How many samples were tested to obtain these results, and what are the associated uncertainties? Although the precise permeance values are not given, they appear from the plots to be around 600 GPU, with a selective layer thickness of about 1 µm, which is relatively thick. If the other TFC membranes have similar selective-layer thicknesses, their corresponding gas permeabilities would exceed 10,000 Barrer while maintaining a selectivity of 100. This seems inconsistent with the low pure-gas permeability values (He permeability <100 Barrer) of the dense films, and therefore the authors should double-check the results.

Reply: Thanks for your comments. Indeed, significant performance variations among thin-film composite (TFC) membranes are expected when fabricated on different substrates.

Previous studies demonstrated the influence of various supports on the gas transport properties of Matrimid membranes [1]. As shown in Table 1, the minimum and maximum permeance values for the same gas (hydrogen) were 4.8 GPU and 1245 GPU, respectively, representing a nearly 300-fold variation. The minimum and maximum selectivity values for H₂/N₂ were 3.2 and 77.9, showing roughly a 25-fold difference. These findings from literature indicate that the gas performance was significantly influenced by various supports.

Table 1. Summary of literature H₂ permeance and H₂/N₂ separation performance.

Support Type Polymer	H ₂ Permeance (GPU)	H ₂ /N ₂ Selectivity
Sulzer PAN UF	4.8	75.2
Synder PAN 30 kDa	8.2	77.9
Vladipor Fluoroplast 50	13.3	4.9
Nanostone Water PV 350	1245	3.2

Therefore, we performed morphological characterization of both supports and TFC membranes. SEM images revealed the variations of pore size in different supports (Figure 1). Furthermore, the distinct morphological differences in the selective layer on different supports were observed in thicker membranes (fabricated to prevent membranes from collapsing rapidly under electron beam bombardment), as shown in Figure 2. Our results demonstrate that differences in support type and the resulting variations in membrane structure are responsible for the significant divergence in properties among membranes fabricated on different supports.

For each membrane type, at least three samples were prepared and tested to ensure reproducibility. The average deviation in gas permeance among replicates was less than 10%.

Gas permeation tests of the membranes were conducted on a variable-pressure constant-volume gas permeation cell. The gas permeability coefficient through the membrane was calculated according to the steady state pressure increment. For TFC membranes fabricated on polymeric supports, our initial thickness calculation method failed to account for penetration of the casting solution into the support and the partial dissolution of the support by the casting solution. This led to a substantial overestimation of the true selective layer thickness, leading to an overestimation of permeance by orders of magnitude. To avoid the influence of solution penetration, which renders the effective membrane thickness unmeasurable and affects performance calculations, the polymeric supports were no longer selected as the support material for the TFC membrane.

Figure 1. SEM images of the upper surface of supports for TFC preparation. The images were acquired after Au metallization at a magnification of 100,000 \times .

Figure 2. SEM images of the TFC membranes on different supports.

Supplementary References

[1] M. Longo, M. Monteleone, E. Esposito, A. Fuoco, E. Tocci, M.C. Ferrari, B. Comesaña-Gándara, R. Malpass-Evans, N.B. McKeown, J.C. Jansen, Thin film composite membranes based on the polymer of intrinsic microporosity PIM-EA (Me₂)-TB blended with Matrimid® 5218, *Membranes*, 12 (9), 881 (2022).

5. If the selective-layer thickness were reduced to around 100 nm, would it be possible to achieve higher permeance while maintaining high selectivity?

Reply: Thanks for your comments. When the selective layer thickness is reduced to around 100 nm, the gas permeance increases while the selectivity decreases significantly (Figure 1 and Table 1). This trade-off relationship is consistent with established patterns reported by the previous literature for polymeric TFC membranes [1-3]. Therefore, it remains a great challenge to achieve both higher permeance and high selectivity at this thickness based on our experimental data. For thicker membranes, the polymer chains are more densely packed, which significantly hinders gas permeation. Moreover, thicker membranes provide longer diffusion paths and more interaction sites, thereby exhibiting lower permeance and higher selectivity. In contrast, for the thinner membrane (Matrimid-Cyclen-110nm), the looser packing of polymer chains causes the rapid increase in permeance. Due to the shorter diffusion paths, the membrane exhibits a reduced discrimination between small and large molecule gases, resulting in lower selectivity.

Figure 1. Effect of membrane's thickness of the selective layer on gas separation performances.

Table 1. Summary of gas transport properties for membranes of varying average thicknesses on AAO supports at 1.0 atm and 25 °C.

Membrane	Gas permeance (GPU)		
	He	H ₂	CO ₂
AAO supports	7747.92±115.9	6656.22±41.38	3895.34±81.29
Matrimid-Cyclen-110nm	1385.95±16.75	1111.54±25.62	458.02±45.12
Matrimid-Cyclen-1.8μm	101.93±6.41	68.32±4.16	22.58±3.11
Matrimid-Cyclen-5.6μm	18.25±0.36	15.37±0.25	5.15±0.16

Membrane	ideal Selectivity					
	He/N ₂	He/CH ₄	H ₂ /N ₂	H ₂ /CH ₄	CO ₂ /N ₂	CO ₂ /CH ₄
AAO supports	1.88	2.56	1.62	2.20	0.96	1.29
Matrimid-Cyclen-110nm	58.41	66.09	46.84	53.01	19.30	21.84
Matrimid-Cyclen-1.8μm	550.95	728.04	369.30	488.00	122.05	161.29
Matrimid-Cyclen-5.6μm	829.55	1073.53	698.64	904.12	234.10	302.94

Based on this, the cross-sectional imaging analysis was performed on TFC membranes with various thicknesses. SEM images revealed that the selective layer structure of thin TFC membranes was relatively loose with distinct striations, whereas thick TFC membranes exhibited a denser, more homogeneous structure (Figure 1).

Figure 1. Cross section SEM images of the TFC membranes with different thicknesses on AAO supports.

Supplementary References

[1] M. Yavari, T. Le, H.Q. Lin, Physical aging of glassy perfluoropolymers in thin film composite membranes. Part I. Gas transport properties, *Journal of Membrane Science*, 525, 387-398 (2017).

[2] J.L. Wang, Y.H. Ding, M. He, X.D. Ding, X. Liu, W.Q. Shi, Direct Preparation of Ultrathin Polymer membranes on porous substrates for the separation of Helium from methane, *Small*, 21 (4), 2406440 (2025).

6. The authors are also encouraged to evaluate the mixed-gas performance of the TFC membranes, which would provide more realistic insight into their industrial potential.

Reply: Thanks for your comments. Following your suggestions, the He and CH₄ mixed gases cannot be prepared in our lab or bought from company, therefore, we evaluated both the pure and mixed-gas (CO₂/N₂, 50/50 vol%) performance of the Matrimid-Cyclen-1.8 μm TFC membrane (Figure 1), and the results were summarized in Table 1. Compared to single-gas permeation, both the CO₂ permeance and CO₂/N₂ selectivity are lower under equivolume mixed-gas conditions due to competitive adsorption among gas molecules.

Figure 1. Effect of feed gas composition on the separation performance of

Matrimid-Cyclen-1.8 μ m TFC membrane.

Table 1. Summary of single-gas and mixed-gas (CO₂/N₂ = 50 vol./50 vol.%) separation performance of Matrimid-Cyclen TFC membrane (average thickness: 1.8 μ m) at 1.0 atm and 25 °C.

Gas type	CO ₂ permeance (GPU)	CO ₂ /N ₂ Selectivity
Pure gas	22.58 \pm 3.11	122.05
Mixed-50:50	18.34 \pm 1.35	87.88

7. More detailed characterization of the TFC membranes is strongly recommended, including both surface and cross-sectional imaging, along with accurate measurement of the selective-layer thickness. These data should be used to calculate the gas permeability, enabling direct comparison between permeance and permeability. The corresponding experimental methods and analysis procedures should also be clearly described.

Reply: Thanks for your comments. Matrimid-Cyclen TFC membranes were fabricated on AAO supports using a scalable bar-coating method [1]. Scanning electron microscopy (SEM) images (Figures 1a-b) showed no significant aggregation or particulate formation. As the membrane thickness increased, the membrane structure evolved to become denser and smoother (Figure 1a). For the calculation of gas permeance, the thickness of the Matrimid-Cyclen TFC membrane's selective layer was determined from cross-sectional SEM images, with values of approximately 110 nm, 1.8 μ m, and 5.6 μ m.

Figure 1. SEM images and gas separation performance of Matrimid-Cyclen-5% TFC membrane. (a) Cross section and (b) surface SEM images of the Matrimid-Cyclen TFC membrane with different thickness.

For TFC membrane, the gas separation parameters, including gas permeance (J) and separation selectivity, were calculated according to the following equation (1) [2-3]:

$$J = \frac{P}{l} \quad (1)$$

Where J denotes the gas permeance of the TFC membrane in GPU (1 GPU=1 \times 10⁻⁶

$\text{cm}^3 \text{ (STP)} \cdot \text{cm}^{-2} \cdot \text{s}^{-1} \cdot \text{cmHg}^{-1}$), P denotes the gas permeance coefficient in Barrer ($1 \text{ Barrer} = 1 \times 10^{-10} \text{ cm}^3 \text{ (STP)} \cdot \text{cm} \cdot \text{cm}^{-2} \cdot \text{s}^{-1} \cdot \text{cmHg}^{-1}$), l represents the thickness of the membrane selective-layer (μm).

The gas permeation ideal selectivity ($\alpha_{A/B}$) is obtained by the following equation (2):

$$\alpha_{A/B} = \frac{J_A}{J_B} \quad (2)$$

where J_A and J_B refer to the permeance of gases A and B, respectively.

Supplementary References

[3] J. Guan, J.C. Du, Q. Sun, W. He, J. Ma, S.U. Hassan, J. Wu, H.J. Zhang, S. Zhang, J.T. Liu, Metal-organic cages improving microporosity in polymeric membrane for superior CO_2 capture, *Science Advances*, 11 (4), eads0583 (2025).

[4] W.J. Fu, L. Zhang, J.C. Liu, T. Yang, M.X. Sun, X.H. Ma, Y.P. Zhao, L. Chen, Ceramic-based composite membranes decorated by incorporating ZIF-8 and PDMS for highly efficient CO_2/N_2 separation, *Separation and Purification Technology*, 352, 128142 (2025).

[5] C. Feng, Y.L. Ma, J. Wei, M. Deng, Z.K. Qin, J.Y. Liu, B. Tang, X.H. Ma, J. Liu, W.J. Jiang, L. Yang, L. Yao, Z. Changwu, Z.D. Dai, 2D zeolite-based thin film nanocomposite membranes for efficient CO_2 separation, *Industrial & Engineering Chemistry Research*, 63 (25), 11134-11144. (2024).

Following your suggestions, detailed experimental methods and data analysis procedures, surface and cross-sectional imaging of the TFC membranes, and the corresponding performance analysis have been added to the revised manuscript.

Added:

Figure 7. SEM images and gas separation performance of Matrimid-Cyclen-5% TFC membrane. (a) Cross section and (b) surface SEM images of the Matrimid-Cyclen TFC membrane with different thicknesses.

Thin-film composite (TFC) membranes, composed of a highly selective thin layer supported by a mechanically robust porous support, have been employed for practical gas separation [61-62]. Matrimid-Cyclen TFC membranes were fabricated on AAO supports using a scalable bar-coating method [63]. Scanning electron microscopy

(SEM) images (Figures 7a-b) showed no significant aggregation or particulate formation. As the membrane thickness increased, the membrane structure evolved to become denser and smoother (Figure 7a). For the calculation of gas permeance, the thickness of the Matrimid-Cyclen TFC membrane's selective layer was determined from cross-sectional SEM images, with values of approximately 110 nm, 1.8 μm , and 5.6 μm .

Fabrication of TFC membranes A raw Matrimid dope solution was prepared by dissolving 0.4 g pure Matrimid powder in 16.0 g DMF solvent stirred for 10 hours and sonicated for 2 hours. The clear Matrimid dope solution was obtained by filtering the raw Matrimid dope solution through a 1.2 μm PTFE syringe. Cyclen powder (20.0 mg) were added to 24.0 g DMF solvent, and the mixture was stirred for 10 hours and sonicated for 2 hours to obtain clear and transparent solutions. The clear Matrimid dope solution was added to the Cyclen/DMF solution, stirred for 24 hours and sonicated for 1 hour to obtain homogeneous Matrimid-Cyclen solutions. The solution was then cast onto the anodic aluminum oxide (AAO) support using a bar-coater with a uniform speed and then put into a vacuum oven at 120 $^{\circ}\text{C}$ for 2-hr solvent evaporation. The as-fabricated samples were named as Matrimid-Cyclen-n (where n represents the membrane thickness). The membrane thickness was measured using cross-sectional SEM imaging.

For TFC membrane, the gas separation parameters, including gas permeance (J) and separation selectivity, were calculated according to the following equation (3) :

$$J = \frac{P}{l} \quad (3)$$

Where J denotes the gas permeance of the TFC membrane in GPU (1 GPU= $1 \times 10^{-6} \text{ cm}^3 \text{ (STP)} \cdot \text{cm}^{-2} \cdot \text{s}^{-1} \cdot \text{cmHg}^{-1}$), P denotes the gas permeance coefficient in Barrer (1 Barrer= $1 \times 10^{-10} \text{ cm}^3 \text{ (STP)} \cdot \text{cm} \cdot \text{cm}^{-2} \cdot \text{s}^{-1} \cdot \text{cmHg}^{-1}$), l represents the thickness of the membrane selective-layer (μm).

The gas permeation ideal selectivity ($\alpha_{A/B}$) is obtained by the following equation (4):

$$\alpha_{A/B} = \frac{J_A}{J_B} \quad (4)$$

where J_A and J_B refer to the permeance of gases A and B, respectively.

Thank you very much for your attention and time. Look forward to hearing from you.

Response to the Reviewers' Comments:

Reviewer #4 (Remarks to the Author):

The authors have made considerable efforts to address the reviewer' comments, and most of the concerns have been satisfactorily resolved. I am particularly pleased to see that the authors prepared thin-film composite membranes with varied thickness and systematically measured gas permeance and selectivity. The results are reasonable. These additional experiments significantly strengthen the manuscript and greatly improve the overall quality of the work.

The new results confirm that my previous concerns were valid and, importantly, they have been clearly and convincingly explained in the revised manuscript. Overall, the revisions have substantially enhanced the rigor, clarity, and impact of the study. The manuscript is now much improved and suitable for publication.

Minor comment: The title of the manuscript could be further improved for clarity and specificity. For example, "Mixed-matrix membranes with molecular recognition windows for selective helium extraction from natural gas" would more accurately reflect the content and focus of the work.

Reply: Thank the reviewer for the highly positive remarks and support of publication of this work. The title of the manuscript has been changed to "Mixed-matrix membranes with molecular recognition windows for selective helium extraction from natural gas".

Helium extraction from natural gas via gas molecule recognition window

Wen He, Xiangzeng Wang, Jian Guan, Quansheng Liang, Ji Ma, Ying Liu, Hongjun Zhang, Chunwei Zhang, Jiangtao Liu

In my opinion, the paper is a relevant contribution to the Membrane Science and Technology Field. The title is somehow limiting the scope of the paper as far as, for example, the CO₂/CH₄ separation features of the material produced are as much relevant or even more than those concerning the He/CH₄ pair announced in the title and abstract. Nevertheless, there are some questions I would like the authors to correct or answer.

- 1) English needs a careful revision as far as there are many errors concerning for example conjugation and concordance. Some strange election of words as, for example, “scalloped vein structure” or “fan-shaped vein structures” should be reviewed.
- 2) Figure 1 is nice and stunning but useless. Remove it.
- 3) In Figure 2, only the transversal cuts give some information. A higher magnification should be shown anyway. How were the transversal sections performed? Some, let say, inhomogeneity appears, probably forming paths through the membrane, appear when Cyclen load increases. These areas should be studied in some detail. By the way, I would assume that the first file of pictures corresponds to the zero Cyclen filling. This should be clarified. Please select only the transversal cuts with a higher magnification and include labels with the Cyclen content on the pictures.
- 4) How do you get the very same thickness with or without Cyclen? Authors mention “injected into a clean Petri-dish by a syringe” How was this controlled as to get errors of only 25% in thickness? Were there variations depending on the Cyclen load? How were these errors evaluated?
- 5) Authors write, “However, the self-aggregation of Cyclen tends to happen when the Cyclen loading was too high, and the hydrogen bonding between excess Cyclen leads to the formation of unobservable discrete particulates in the membrane”. This statement constitutes an authors’ guess, or what does it say? I think that if aggregation clusters were unobservable, in absolute terms, they should be essentially not existing instances. Avoid this kind of statements deleting the phrase or argue on the probability and signs of the existence of such aggregates that cannot be seen at the magnifications or with the procedures of analysis used.
- 6) In the Figure 3 caption, authors do not mention plot “e” referring to element analysis performed by an XPS analysis. Authors should mention that their scheme in the “c” section refers to the 5 % Cyclen membrane. Both the smallest (4.09 Angstrom) and higher (5.93 Angstrom) d_spacing decrease only until 5 % Cyclen and then increase monotonously until the highest Cyclen and eventually surpass the pure matrimid ones. Can it be correlated with the appearance of inhomogeneities detected in Figure 2? How is it possible, given that both the figures refer to very different scales?
- 7) In reference to the higher d-spacing (out of the two detected) what do the authors think is the cause of this interchain segment distance and their changes? Have you tested the evolution of d-spacing for the aging membranes? If decreases in d_spacing are associated with hydrogen bonding, what is causing the high load increase in d-spacing?
- 8) Authors say that “which adjusted the submicroporous structure of the MMM membrane, increased the stacking density of Matrimid polymer chains”. This could only be true until 5% Cyclen, and should be stated so. Moreover, it is important to mention that small

decreases in d-spacing do not automatically translate into a significant change in packing density.

- 9) Authors say that “Cyclen begins to degrade at temperatures slightly greater than 100 °C”; why do the membranes containing Cyclen only decompose at 300 to 450 °C? Have you performed mechanical or, better, thermomechanical investigations on the membranes? Any idea on the mechanical behavior that could be expected?
- 10) Could the authors explain how Table S-1 was obtained? What are the respective roles of N-H...O and C=O...H in Table S1. Do they refer to the percentage of NH linked by hydrogen bridges (relevance of bridges for Cyclen) and the percentage of C=O linked by hydrogen bridges (relevance of bridges for matrimid)? Therefore they should be correlated, are they?
- 11) It is declared that “gas permeation tests for He (2.60 Å), H₂ (2.89 Å), CO₂ (3.3 Å), N₂ (3.64 Å) and CH₄ (3.80 Å)...” These Kinetic (?) diameters should be described and referenced. What were they used for?
- 12) Referring to aging, it mainly reflects lability of the structure than collapses to some extent with as time passes; this reduces the size of the sites making up the gas paths. Initially it leads to an increase of selectivity until the small gas molecules are also hindered with a decrease in selectivity and a substantial decrease in permeability. Figure 5 shows how selectivity increases with a reduction in permeability with aging. Until what time is it so? Have you tested plastification? Have you changed the applied pressure over 6 bar pressure drop?
- 13) Authors state “which facilitated the extraction of high-purity He from natural gas at low temperatures”. It would be interesting to see how selectivity changes with temperature. In Figure 4f you show permeabilities evolution in a cyclic loop from 25 to 45 °C and back to 25 °C but it is difficult to get an idea on the evolution of selectivity. Authors say: “The gas separation performance of the membranes fully recovered as the temperature decreased from 45 °C to 25 °C, indicating that the changes in membrane performance with temperature are related to the characteristics of the membrane”; what should be, according to authors, the membrane characteristics that would explain this? Why is CO₂ absent from all Figure 4?
- 14) In Figures 5g and 5f the results shown must be referenced and described including, for example, to what load percentages do they correspond (maybe 1 %?). In Figure 5h, what is in ordinates? Enhancement of what? Figure 5i is useless.